# Reinforcement Learning with Feedback Graphs

**Christoph Dann**
Google Research
cdann@cdann.net

**Yishay Mansour**
Tel Aviv University
and Google Research
mansour.yishay@gmail.com

**Mehryar Mohri**
Google Research and
Courant Institute of Math. Sciences
mohri@google.com

**Ayush Sekhari**
Cornell University
as3663@cornell.edu

**Karthik Sridharan**
Cornell University
ks999@cornell.edu

## Abstract

We study RL in the tabular MDP setting where the agent receives additional observations per step in the form of transitions samples. Such additional observations can be provided in many tasks by auxiliary sensors or by leveraging prior knowledge about the environment (e.g., when certain actions yield similar outcome). We formalize this setting using a feedback graph over state-action pairs and show that model-based algorithms can incorporate additional observations for more sample-efficient learning. We give a regret bound that predominantly depends on the size of the maximum acyclic subgraph of the feedback graph, in contrast with a polynomial dependency on the number of states and actions in the absence of side observations. Finally, we highlight fundamental challenges for leveraging a small dominating set of the feedback graph, as compared to the well-studied bandit setting, and propose a new algorithm that can use such a dominating set to learn a near-optimal policy faster.

## 1 Introduction

For many real-world applications, the sample complexity of RL is still prohibitively high, making it vital to simplify the learning task by incorporating domain knowledge. An effective approach to do so is through imitation learning [1] but there are many applications where even an expert may not know how to provide demonstrations of near-optimal policies (e.g. in drug discovery or tutoring systems). In such applications, an expert may still be able to provide insights into the structure of the task, for example, that certain actions yield similar behavior in certain states. These insights could, in principle, be baked into a model or value-function class, but RL with complex function classes is still very challenging, both in theory and practice [2, 3, 4, 5].

A more convenient approach to incorporate structure from domain knowledge is to directly provide additional observations to the RL algorithm. In that case, an online RL algorithm not only gets to see the outcome (reward and next state) of executing the current action in the current state, but also an outcome of executing other actions, possibly even in other states. While there is often a trivial way to include such observations of hypothetical transitions in existing methods, little is theoretically known about the benefits of doing so. This raises the question:

*How do additional observations of hypothetical transitions affect the sample-efficiency of online RL algorithms, and how to best incorporate such observations?*

To study this question in full generality, we assume that the additional observations come from some (black-box) oracle that provides the algorithm with a set of transition samples, in addition to the current reward and next state $(r_h, s_{h+1})$ from the environment (Figure 1 left). To study how helpful

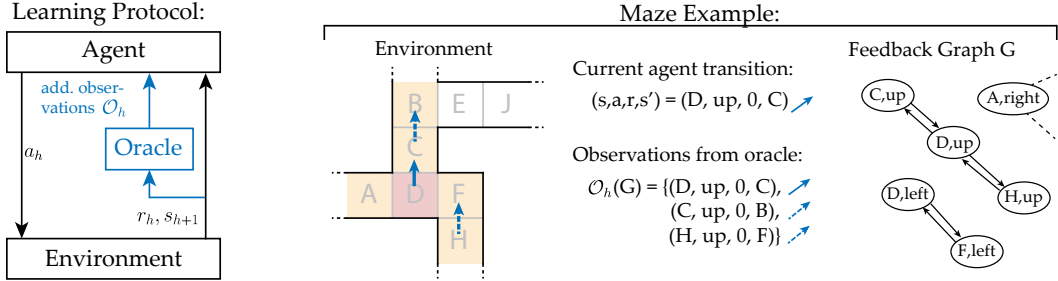

Figure 1: *Left:* RL loop with additional observations from the (black-box) oracle; *Right:* grid world example; Through additional sensors, the robot in state $D$ also sees nearby states (yellow) and when taking the action *up*, the oracle provides the actual transition (solid arrow) as well as hypothetical transitions (dashed arrow) from nearby states. This is formalized by a *feedback graph $G$* over state-action pairs shown on the right (snippet). Since $(D, up)$ has an edge to $(C, up)$ and $(H, up)$ in the feedback graph, the agent receives a (hypothetical) transition for both.

the oracle is, we formalize which observations it provides by a *feedback graph* [6] over state-action pairs: an edge in the feedback graph from state-action pair $(s, a)$ to state-action pair $(\bar{s}, \bar{a})$ indicates that, when the agent takes action $a$ at state $s$, the oracle additionally provides a reward and next-state sample $(\bar{r}', \bar{s}')$ that the agent would have seen, had it taken action $\bar{a}$ in state $\bar{s}$. Thus, at each time step, the agents gets to see the outcome of executing the current $(s, a)$ as well as an outcome for each state-action pair that has an edge from $(s, a)$ in the feedback graph.

To illustrate this setting, consider a robot moving in a grid world (Figure 1, right). Through auxiliary sensors, it can sense positions in its line of sight. When the robot takes an action to move in a certain direction, it can also predict what would have happened for the same action in other positions in the line of sight. The oracle formalizes this prediction ability and provides the RL algorithm with observations of (hypothetical) movements for that action in the nearby states. Here, the feedback graph connects state-action pairs with matching actions and states in the line of sight.

Our definition of feedback graphs also captures the common practice of image augmentation, that has been empirically demonstrated to improve performance [7, 8, 9]. In this case, the states are images and for any state-action pair $(s, a)$, the feedback graph contains an edge to all $(\bar{s}, a)$ such that $\bar{s}$ is a random transformation of $s$. Another real-world motivating example is recommender systems, where, the goal – to maximize the long-term satisfaction of each user, by choosing which items to recommend given the user history – is a natural setting for RL. Here, RL with feedback graphs can improve the generalization across users, which is still a major challenge for RL [10]. There exist other techniques, such as collaborative filtering, that can predict that certain users would react similarly to certain recommendations, but they have other drawbacks. Incorporating these techniques into the oracle can leverage the complementary strengths of these methods and RL.

We present an extensive study of RL in the tabular MDP setting with feedback graphs defining the availability of additional observations. Our main contributions, summarized in Table 1, are:

- We prove that, by incorporating the additional observations into the model estimation step, existing model-based RL algorithms [11, 12] can achieve regret and sample-complexity bounds that scale with the mas-number $\mu$ of the feedback graph – as opposed to potentially much larger number of states and actions $SA$ (Section 3). This result cannot be readily derived by application of standard bandit analyses and required a series of novel results on real-valued self-normalizing sequences over graph vertices (Appendix E).

- We give a lower bound on the regret (Appendix B) that scales with the independence number $\alpha$ of the feedback graph. For undirected feedback graphs, the equality $\mu = \alpha$ holds. This suggests that the regret guarantee of our algorithm cannot be improved further for such feedback graphs.

- We also prove a lower sample complexity bound, which shows that (1) an improvement to scale with the domination number $\gamma$ may be possible and (2) we cannot completely remove the dependence on the independence number $\alpha$ in the lower order terms. The latter is in stark contrast to the bandit setting [e.g. 6, 13, 14, 15, 16], where the sample complexity can scale with $\gamma$ only. Thus, leveraging a small domination set of the feedback graph is fundamentally harder in the MDP setting, since the agent first has to learn how to reach the state-action pairs in the dominating set.

| | | **Worst-Case Regret** | **Sample Complexity** |
|---|---|---|---|
| without feedback graph | ORLC [17] | $\widetilde{O}(\sqrt{SAH^2T} + SA\hat{S}H^2)$ | $\widetilde{O}\left(\frac{SAH^2}{\epsilon^2} + \frac{SA\hat{S}H^2}{\epsilon}\right)$ |
| | Lower bounds [18, 19] | $\widetilde{\Omega}(\sqrt{SAH^2T})$ | $\widetilde{\Omega}\left(\frac{SAH^2}{\epsilon^2}\right)$ |
| with feedback graph | ORLC [Thm. 1, Cor. 1] | $\widetilde{O}(\sqrt{\mu H^2T} + \mu\hat{S}H^2)$ | $\widetilde{O}\left(\frac{\mu H^2}{\epsilon^2} + \frac{\mu\hat{S}H^2}{\epsilon}\right)$ |
| | Algorithm 3 [Thm. 3] | at least $O(\gamma^{1/3}T^{2/3})$ | $\widetilde{O}\left(\frac{\gamma H^3}{p_0\epsilon^2} + \frac{\gamma\hat{S}H^2}{p_0\epsilon} + \frac{\mu\hat{S}H^2}{p_0}\right)$ |
| | Lower bounds [Thm. 4, 5] | $\widetilde{\Omega}(\sqrt{\alpha H^2T})$ | $\widetilde{\Omega}\left(\frac{\gamma H^2}{p_0\epsilon^2} + \frac{\alpha}{p_0} \wedge \frac{\alpha H^2}{\epsilon^2}\right)$ |

Table 1: Comparison of our main results. The symbols $\mu$, $\alpha$ and $\gamma$ denote the mas-, independence and domination number of the feedback graph respectively, with $\gamma \le \alpha \le \mu \le SA$. The symbol $T$ denotes the total number of episodes, $\epsilon$ denotes the optimality gap of the returned policy, $H$ denotes the episode length and $p_0$ is a parameter for how easy the dominating set can be reached.

- We present an algorithm that overcomes the above challenges for the MDP setting and achieves a sample complexity bound that scales with the more favorable domination number $\gamma$ in the leading $\frac{1}{\epsilon^2}$ term (Section 5). A key insight for obtaining this result is a new formulation of multi-task RL as an extended MDP (Section 4).

## 2 Problem setup

### 2.1 Episodic RL in tabular MDP

The agent interacts with an MDP in episodes indexed by $k$. Each episode is a sequence $(s_1, a_1, r_1, \dots, s_H, a_H, r_H)$ of $H$ states $s_h \in \mathcal{S}$, actions $a_h \in \mathcal{A}$ and scalar rewards $r_h \in [0, 1]$. The initial state $s_1$ can be chosen arbitrarily, possibly adversarially. Actions are taken as prescribed by the agent's policy $\pi_k$ which are deterministic and time-dependent mappings from states to actions, i.e., $a_h = \pi_k(s_h, h)$ for all time steps $h \in [H] := \{1, 2, \dots H\}$. The successor states and rewards are sampled from the MDP as $s_{h+1} \sim P(s_h, a_h)$ and $r_h \sim P_R(s_h, a_h)$. We denote by $\mathcal{X} = \mathcal{S} \times \mathcal{A}$ the space of all state-action pairs $(s, a)$ that the agent can encounter, i.e., visit $s$ and take $a$. For a pair $x \in \mathcal{X}$, we denote by $s(x)$ and $a(x)$ its state and action respectively. We restrict ourselves to tabular MDPs with finite $\mathcal{X}$. The agent knows the horizon $H$ and the set $\mathcal{X}$, but does not have access to $P$ and $P_R$.

The Q-value of a policy is defined as the reward to go, given the current state and action when the agent follows $\pi$ afterwards $Q_h^\pi(s, a) := \mathbb{E}[\sum_{t=h}^H r_t \mid a_h = a, s_h = s, a_{h+1:H} \sim \pi]$, and the state-values of $\pi$ are $V_h^\pi(s) := Q_h^\pi(s, \pi_h(s))$. The expected return of a policy in episode $k$ is simply the value $V_1^\pi(s_{k,1})$ of the initial state $s_{k,1}$. Any policy that achieves optimal reward to go, i.e., $\pi(s, h) \in \text{argmax}_a Q_h^\pi(s, a)$ is called optimal. We use the superscript $^\star$ to denote any optimal policy and its related quantities. The sample-efficiency of an algorithm can either be measured by its *sample-complexity* or its *regret*. The regret $R(T) = \sum_{k=1}^T (V_1^\star(s_{k,1}) - V_1^{\pi_k}(s_{k,1}))$ is the cumulative difference of achieved and optimal return after $T$ episodes. Sample complexity $N(\epsilon)$ is the number of episodes after which the algorithm can identify an $\epsilon$-optimal policy $\pi$ (with $V_1^\pi(s_1) \ge V_1^\star(s_1) - \epsilon$).

### 2.2 Feedback graphs

In a typical RL setting, when the agent (learner) takes an action $a_h$ at state $s_h$ at time $h$, it receives a sample of the reward $r_h$ and the next-state $s_{h+1}$ observed after taking a step in the MDP. However, in our setting, besides the transition $((s_h, a_h), r_h, s_{h+1})$, an oracle provides the agent with additional observations, about transitions at other states and actions. We denote the set of observations provided to the agent at time step $h$ by $\mathcal{O}_h$ (see Figure 1, left). While being an interesting research direction, the goal of our work in this paper is not to study the design or implementation of these oracles (that provide $\mathcal{O}_h$) but rather how RL algorithms can benefit from the additional observations. To that end, we formalize the additional observations available to the agent by a directed graph $G = (\mathcal{X}, \mathcal{E})$ over state-action pairs called a *feedback graph*, where the vertex set $\mathcal{X}$ comprises of all feasible state-action pairs in the MDP. An edge $x \xrightarrow{G} \bar{x}$ in $G$ (short for $(x, \bar{x}) \in \mathcal{E}$), from a state-action pair $x$ to another state-action pair $\bar{x}$, indicates that when the agent takes action $a(x)$ at state $s(x)$, along with a transition for that step, it also observes a sample of the reward and the next-state $(r', s')$ it would have observed if the agent had been at state $s(\bar{x})$ and had executed action $a(\bar{x})$. Thus, the set

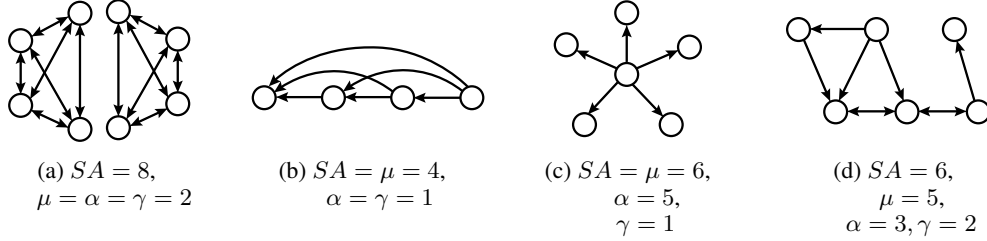

Figure 2: Examples of feedback graphs with different vertex number $SA$, mas-number $\mu$, independence number $\alpha$ and domination number $\gamma$.

(a) $SA = 8$, $\mu = \alpha = \gamma = 2$

(b) $SA = \mu = 4$, $\alpha = \gamma = 1$

(c) $SA = \mu = 6$, $\alpha = 5$, $\gamma = 1$

(d) $SA = 6$, $\mu = 5$, $\alpha = 3, \gamma = 2$

of observations $\mathcal{O}_h$ received by the agent at time step $h$ is

$$\mathcal{O}_h(G) := \{(x_h, r_h, s_{h+1})\} \cup \{(\bar{x}, r', s') \mid x_h \xrightarrow{G} \bar{x}\},$$

where $x_h := (s_h, a_h)$, and the observations $(\bar{x}, \bar{r}, \bar{s})$ are sampled independently with $s' \sim P(x)$ and $r' \sim P_R(x)$, given all previous observations. [1]

We emphasize that even though the agent receives additional observations $\mathcal{O}_h(G)$ at each time step, the next state is still $s_{h+1}$, as determined by the original transition. Further, for the ease of analysis, throughout the paper, we will assume that self-loops (edges of the form $x \xrightarrow{G} x$ in $G$) are implicit and will therefore not include them in edge set $\mathcal{E}$.

**Important graph properties.** The analysis of regret and sample-complexity in this setting uses the following graph-theoretic properties [13], each capturing a notion of connectivity of the graph:

- **Mas-number ($\mu$):** A set of vertices $\mathcal{V} \subseteq \mathcal{X}$ forms an acyclic subgraph of $G$ if the subgraph $\widetilde{G}(\mathcal{V}, \{(v, w) \subseteq \mathcal{V} \times \mathcal{V} : v \xrightarrow{G} w\})$, induced by restricting $G$ to $\mathcal{V}$, does not contain a cycle. The size of the maximum acyclic subgraph denotes the *mas-number* $\mu$ of $G$.

- **Independence number ($\alpha$):** A set of vertices $\mathcal{V} \subseteq \mathcal{X}$ is an independent set if there is no edge between any two nodes of $\mathcal{V}$, i.e. for all $v, w \in \mathcal{V}, v \xrightarrow{G} \!\!\!\!\!/\;\; w$. The size of the largest independent set is called the *independence number* $\alpha$ of $G$.

- **Domination number ($\gamma$):** A set of vertices $\mathcal{V} \subset \mathcal{X}$ forms a dominating set if there is an edge from a vertex in $\mathcal{V}$ to any vertex in $G$, i.e. for all $x \in \mathcal{X}$, there exists a $v \in \mathcal{V}$ such that $v \xrightarrow{G} x$. The size of the smallest dominating set is called the *domination number* $\gamma$.

In the context of feedback graphs, mas- and independence number can be interpreted as a worst-case notion of connectivity measuring how many different vertices any algorithm can visit before observing any vertex twice. In contrast, the domination number is a best-case notion answering the question of how many vertices an algorithm that takes the graph structure into account has to visit in order to receive an observation from every vertex. For any graph, we have $\gamma \leq \alpha \leq \mu \leq |\mathcal{X}|$, where each inequality can be $\Theta(|\mathcal{X}|)$ apart. Independence- and mas-number coincide, $\alpha = \mu$, for undirected graphs where for every edge there is an edge pointing backward. See Figure 2 for examples of feedback graphs and their graph properties and Appendix A.2 for a more extensive discussion. Here, we only give three relevant examples where feedback graph properties can be much smaller than $SA$:

a) *State aggregation* ([20]) can be considered a special case where the feedback graph consists of disjoint cliques, each consisting of the state-action pairs whose state belongs to a an aggregated state. Here $\mu = \alpha = \gamma = AB$ where $B$ is the number of aggregated states and $A = |\mathcal{A}|$.

b) *Reinforcement learning with multiple tasks* is discussed in Section 4.

c) *Four-rooms environment.* For a concrete example of how small graph properties can be in practice, we consider a version of the classic four-room environment [21] where the agent receives side observations for $(s, a)$-pairs with $s$ in its field of view and $a$ matching the current action. Here, $|\mathcal{X}| = SA = 376$, $\mu \leq 146$, $\alpha = 44$ and $\gamma = 16$ (more details in Appendix A.2).

| **Algorithm 1:** Optimistic model-based RL | **Algorithm 2:** `SampleEpisode`$(\pi, s_1, D)$ |
|---|---|
| 1  $D_1 \leftarrow$ Initialize model statistics  $/\!\!/ \widehat{r}, \widehat{r^2}, \widehat{P}, n$ | 1  **for** $h = 1, \ldots, H$ **do** |
| 2  **for** *episode* $k = 1, 2, 3, \ldots$ **do** | 2  $\quad$ Take action $a_h = \pi(s_h, h)$ and |
| 3  $\quad \pi_k, \widetilde{V}_{k,1}, \underset{\sim}{V}_{k,1} \leftarrow$ `OptimistPlan`$(D_k)$ | $\quad\quad$ transition to $s_{h+1}$ with reward $r_h$; |
| 4  $\quad$ Observe initial state $s_{k,1}$ | 3  $\quad$ Receive observations $\mathcal{O}_h(G)$; |
| 5  $\quad D_{k+1} \leftarrow$ `SampleEpisode`$(\pi_k, s_{k,1}, D_k)$ | 4  $\quad$ **for** *transition* $(x, r, s') \in \mathcal{O}_h(G)$ **do** |
| | 5  $\quad\quad D \leftarrow$ `UpdateModel`$(D, x, r, s')$; |
| | **return :** $D$ |

## 3  Mas-number based regret bounds

As a first contribution, we show the benefits of a feedback graph in achieving more favorable learning guarantees for optimistic model-based algorithms. Algorithms in this family, such as UCBVI [22], ORLC [12] or (STRONG-)EULER [11, 23], maintain an estimate $D$ of the MDP model and alternate between (1) computing a policy $\pi_k$ by optimistic planning with $D$ and (2) updating $D$ using the observations from executing $\pi_k$ for one episode (see Algorithm 1). Model-based algorithms can naturally incorporate side observations by updating the model estimate $D$ with all observations $\mathcal{O}_h(G)$ provided by the oracle as stipulated by the feedback graph. As highlighted in lines 3–4 of Algorithm 2, the `SampleEpisode` subroutine calls `UpdateModel` at each time step *for all transition observations in $\mathcal{O}_h(G)$* as opposed to only the performed transition $((s_h, a_h), r_h, s_{h+1})$.

Different model-based algorithms deviate in their implementation of `UpdateModel` and `OptimistPlan`. For concreteness, we will analyze a version of the EULER or ORLC algorithm [12, 11] with `UpdateModel` and `OptimistPlan` provided in Algorithms 4 and 5 in Appendix C.1. We expect our results to directly extend to other model-based RL algorithms, such as UCBVI [22]. Here, the model estimate $D$ consists of first and second moments of the immediate reward i.e. $\widehat{r}(x)$ and $\widehat{r^2}(x)$ respectively, transition frequencies $\widehat{P}(x)$ and the number of observations $n(x)$ for each state-action pair $x \in \mathcal{X}$. `OptimistPlan` is a version of value iteration with reward bonuses. It returns an upper-confidence bound $\widetilde{V}_{k,1}$ on the optimal value function $V_1^\star$ as well as a lower-confidence bound $\underset{\sim}{V}_{k,1}$ on the value function of the returned policy $V_1^{\pi_k}$, and can be can be viewed as an extension of the UCB policy from the bandit to the MDP setting. We prove the following regret bound:

**Theorem 1** (Regret bound). *For any tabular episodic MDP with episode length $H$, state-action space $\mathcal{X} \subseteq \mathcal{S} \times \mathcal{A}$ and directed feedback graph $G$, the regret of Algorithm 1 with subroutines in Alg. 2, 5 and 4 and 5 is with probability at least $1 - \delta$ for all number of episodes $T$*

$$R(T) = \widetilde{O}\big(\sqrt{\mu H^2 T} + \mu \widehat{S} H^2\big), \tag{1}$$

*where $\mu$ is the size of the maximum acyclic subgraph of $G$ and algorithm parameter $\widehat{S} \leq S$ is a bound on the number of possible successor states of each $x \in \mathcal{X}$.*

**Remark 2** (IPOC and sample-complexity bound). *Equation (1) also bounds the cumulative size $\sum_{k=1}^T (\widetilde{V}_{k,1}(s_{k,1}) - \underset{\sim}{V}_{k,1}(s_{k,1}))$ of policy certificates [12] which implies that the algorithm also satisfies a sample-complexity bound $N(\epsilon) = \widetilde{O}\big(\frac{\mu H^2}{\epsilon^2} + \frac{\widehat{S} M H^2}{\epsilon}\big)$.*

Theorem 1 replaces a factor of $SA$ in the best known regret bounds for RL, without side observations [12, 11], with the mas-number $\mu$ (see also Table 1). This is a substantial improvement, since, in many feedback graphs $SA$ may be very large while $\mu$ is a constant. The only remaining polynomial dependency on $\widehat{S} \leq S$ is in the lower-order term, and is standard for model-based RL algorithms. Even in the tabular MDP setting without the feedback graph, removing this term without incurring a significant penalty in $H$ is an interesting open problem. On the lower bound side, we show in Appendix B that the worst-case regret of any algorithm is $\widetilde{\Omega}(\sqrt{\alpha H^2 T})$, where $\alpha$ denotes the independence number of $G$. While $\mu$ and $\alpha$ can differ for general graphs, they are equal for undirected feedback graphs. In that case, the regret bound in Theorem 1 is optimal up to lower-order terms and log-factors. We will show in the following sections that the sample-complexity, however, can be improved further to depend on the domination number $\gamma$, instead of $\alpha$ or $\mu$ as above.

**Technical challenges and proof technique.** See Appendix C.7 for a brief discussion of how our analysis differs from existing ones. In essence, at the core of UCB analyses with feedback graphs

in bandits [24] is a discrete pigeon-hole argument on the vertices of the graph. Applying such an argument in the MDP setting yields an undesirable regret term of $\Omega(\sqrt{H^3T})$, due to an additional concentration argument to account for the stochasticity in the visitation of various state-action pairs by the corresponding policies. We avoid this concentration argument by generalizing the bandit analysis and developing a series of novel technical results on self-normalizing sequences of weight-functions on vertices of a graph. These might be of independent interest (see Appendix E).

## 4  Example: Multi-task RL

In Section 5, we show that the sample complexity of RL can be improved further when the feedback graph has a small dominating set. As a preparation for that discussion, we first show here that feedback graphs can be used to efficiently learn in certain multi-task RL problems. This may be of independent interest. The specific setting we consider is:

There are $m$ tasks that share the same underlying transition dynamics $P$. Each task admits different reward distributions, denoted by $P_R^{(i)}$ for the $i$-th task, where $i \in [m]$. We assume that the initial state is fixed and that reward distributions of all but one task are known to the agent.[2] The goal is to learn a policy that, given the task identity $i$, performs $\epsilon$-optimally. This is equivalent to learning an $\epsilon$-optimal policy for each task.

The naive approach is to use $m$ instances of any PAC-RL algorithm to learn each task separately. With Algorithm 1 for example, this would require in total

$$\widetilde{O}\Big(\frac{m(1+\epsilon\widehat{S})\mu H^2}{\epsilon^2}\Big)$$

episodes. If the number of tasks $m$ is large, this can be significantly more costly than learning a single task. We will now show that this dependency on $m$ can be removed with the help of feedback graphs.

**Extended MDP.** We can learn the $m$ tasks jointly by running Algorithm 1 in an extended MDP $\bar{\mathcal{M}}$. In this extended MDP, the state is augmented with a task index, that is, $\bar{\mathcal{S}} = \mathcal{S} \times [m]$. In states with index $i$, the rewards are drawn from $P_R^{(i)}$ and the dynamics according to $P$ with successor states having the same task $i$. Formally, the dynamics $\bar{P}$ and expected rewards $\bar{r}$ of the extended MDP are

$$\bar{P}((s',j)|(s,i),a) = \mathbf{1}\{i=j\}P(s'|s,a), \quad \text{and,} \quad \bar{r}((s,i),a) = r_i(s,a)$$

for all $s \in \mathcal{S}$, $a \in \mathcal{A}$, $i, j \in [m]$ where $r_i(s,a) = \mathbb{E}_{r \sim P_R^{(i)}(s,a)}[r]$ are the expected rewards of task $i$. Essentially, the extended MDP consists of $m$ disjoint copies of the original MDP, each with the rewards of the respective task.

The key for learning all tasks jointly is to define the feedback graph $\bar{G}$ of the extended MDP so that it connects all copies of state-action pairs that are connected in the feedback graph $G$ of the original MDP. That is, for all $s, s' \in \mathcal{S}, a, a' \in \mathcal{A}, i, j \in [m]$, $((s,i),a) \overset{\bar{G}}{\to} ((s',j),a') \Leftrightarrow (s,a) \overset{G}{\to} (s',a')$. Note that we can simulate an episode of $\bar{\mathcal{M}}$ by running the same policy in the original MDP because we assumed that the immediate rewards of all but one task are known. Therefore, to run Algorithm 1 in the extended MDP, it is only left to determine the task index $i_k$ of each episode $k$. To ensure learning all tasks equally fast and not wasting resources on a single task, we choose the task for which the value confidence bounds are farthest apart, i.e., $i_k \in \mathrm{argmax}_{i \in [m]} \ \widetilde{V}_{k,1}((s_{k,1},i)) - \underaccent{\sim}{V}_{k,1}((s_{k,1},i))$. This choice implies that if this difference is at most $\epsilon$ for the chosen task, then the same holds for all other tasks. Thus, when the algorithm encounters $\widetilde{V}_{k,1}((s_{k,1},i_k)) - \underaccent{\sim}{V}_{k,1}((s_{k,1},i_k)) \leq \epsilon$, the current policy is $\epsilon$-optimal for all tasks as $\widetilde{V}_{k,1} \leq V_1^{\pi_k} \leq V_1^{\star} \leq \widetilde{V}_{k,1}$. By Remark 2 above, this happens for Algorithm 1 in

$$\widetilde{O}\Big(\frac{(1+\epsilon\widehat{S})\mu H^2}{\epsilon^2}\Big)$$

episodes. Note that we used the mas-number $\mu$ and maximum number of successor states $\widehat{S}$ of the original MDP, as these quantities are identical in the extended MDP. By learning tasks jointly through feedback graphs, *the total number of episodes needed to learn a good policy for all tasks*

**Algorithm 3:** RL using given dominating set $\mathcal{X}_D$ and parameters $\delta \in (0, 1]$ and $\widehat{S} \leq S$

---

1   Initialize model statistics $D$ and active task set $\mathcal{I} \leftarrow \{1, \ldots, \gamma\}$;
    */* First phase: find policy to reach each vertex in given dominating set */*

2   **while** $\mathcal{I} \neq \varnothing$ **do**

3      $\pi, \widetilde{V}_h, \underset{\sim}{V}_h \leftarrow \texttt{OptimistPlan}(D)$ ;              *// Alg. 5, with probability parameter $\delta/2$*

4      $j \leftarrow \operatorname{argmax}_{i \in \mathcal{I}} \widetilde{V}_1((s_1, i)) - \underset{\sim}{V}_1((s_1, i))$;

5      **for** $i \in \mathcal{I}$ **do**

6          **if** $\widetilde{V}_1((s_1, i)) \leq 2\underset{\sim}{V}_1((s_1, i))$ **then**

7              $\pi^{(i)}((s, 0), h) \leftarrow \pi((s, i), h)$ for all $s \in \mathcal{S}$ and $h \in [H]$ ;     *// map policy to task 0*

8              $\mathcal{I} \leftarrow \mathcal{I} \setminus \{i\}, \; \widehat{p}^{(i)} \leftarrow \underset{\sim}{V}_1((s_1, i))$

9      $D \leftarrow \texttt{SampleEpisode}(\pi, (s_1, j), D)$ ;              *// Alg. 2, apply to extended MDP $\bar{\mathcal{M}}$*

    */* Second phase: play learned policies to uniformly sample from dominating set */*

10   **while** $\widetilde{V}_1((s_1, 0)) - \underset{\sim}{V}_1((s_1, 0)) > \epsilon$ **do**

11      $j \leftarrow (j \mod \gamma) + 1$ ;              *// Choose policy in circular order*

12      $D \leftarrow \texttt{SampleEpisode}(\pi^{(j)}, (s_1, 0), D)$ ;         *// Alg. 2, apply to extended MDP $\bar{\mathcal{M}}$*

13      $\pi, \widetilde{V}_h, \underset{\sim}{V}_h \leftarrow \texttt{OptimistPlan}(D)$ ;         *// Alg. 5, with probability parameter $\delta/2$*

14   $\hat{\pi}(s, h) \leftarrow \pi((s, 0), h)$ for all $s \in \mathcal{S}$ and $h \in [H]$;      *// map policy back to original MDP*
     **return :** $\hat{\pi}$

---

*does not grow (polynomially) with the number of tasks* and we save a factor of $m$ compared to the naive approach without feedback graphs. This might seem to be too good to be true but it is possible because the rewards of all but one task are known and the dynamics is identical across tasks. While one could derive and analyze a specialized algorithm without feedback graphs for this setting, this would likely be much more tedious.

## 5   Domination number based sample-complexity bounds

Algorithm 1 uses additional observations efficiently, despite being agnostic to the feedback graph structure. Yet, sometimes, an alternative approach can be further beneficial. In some problems, there are state-action pairs which are highly informative, that is, they have a large out-degree in the feedback graph. Consider for example a ladder in the middle of a maze. Going to this ladder and climbing it may be time-consuming (low reward) but it reveals the entire structure of the maze, thereby making a subsequent escaping much easier. In this case the domination number $\gamma \ll \alpha \leq \mu$ is much smaller than other feedback graph properties. Explicitly exploiting such state-action pairs may not be advantageous to improve over the regret (as such pairs could have low reward giving worst case regret $\Omega(T^{2/3})$), but may be useful when the goal is to eventually learn a good policy without caring about the performance during the learning process. We therefore study the sample-complexity of RL in the MDP setting given a small dominating set $\mathcal{X}_D = \{X_1, \ldots, X_\gamma\}$ of the feedback graph. We discuss in Appendix F.1 how to extend Algorithm 3 when the dominating set is not known.

We propose a simple algorithm that aims to explore the MDP by uniformly visiting state-action pairs in the dominating set. This works because the dominating set admits outgoing edges to every vertex, that is $\forall x \in \mathcal{X}, \exists x' \in \mathcal{X}_D : x' \overset{G}{\to} x$. However, compared to the bandit setting [13] with immediate access to all vertices, there are additional challenges for such an approach in the MDP setting:

1. **Unknown policy for visiting the dominating set**: While we assume to know the identity of the state-action pairs in a dominating set, we do not know how to reach those pairs.

2. **Low probability of visiting the dominating set**: Some or all state-action pairs in the dominating set might be hard to reach under any policy.

The lower bound in Theorem 5 in Appendix B shows that these challenges are fundamental. To address them, Algorithm 3 proceeds in two phases. In the first phase (lines 2–9), we learn policies $\pi^{(i)}$ that visit each element $X_i \in \mathcal{X}_D$ in the dominating set with probability at least $\frac{p^{(i)}}{2}$. Here,

$p^{(i)} = \max_\pi \mathbb{E}[\sum_{h=1}^{H} \mathbf{1}\{(s_h, a_h) = X^{(i)}\} \mid \pi]$ is the highest expected number of visits to $X_i$ per episode possible. The first phase leverages the construction for multi-task learning from Section 4. We define an extended MDP for a set of tasks $0, 1, \ldots, \gamma$. Task 0 is to maximize the original reward and tasks $1, \ldots, \gamma$ aim to maximize the number of visits to each element of the dominating set. To this end, we define the rewards for each task of the extended MDP as

$$\bar{r}((s,0),a) = r(s,a) \qquad \bar{r}((s,k),a) = \mathbf{1}\{(s,a) = X_k\}, \quad \forall k \in [\gamma], s \in \mathcal{S}, a \in \mathcal{A}.$$

The only difference with Section 4 is that we use a subset of the tasks and stop playing a task once we have identified a good policy for it. The stopping condition in Line 6 ensures that policy $\pi^{(i)}$ visits $X_i$ in expectation at least $\widehat{p}^{(i)} \geq \frac{p^{(i)}}{2}$ times. In the second phase (lines 10–14), each policy $\pi^{(i)}$ is played until there are enough samples per state-action pair to identify an $\epsilon$-optimal policy.

**Theorem 3** (Sample complexity of Algorithm 3). *For any tabular episodic MDP with state-actions $\mathcal{X}$, horizon $H$, feedback graph with mas-number $\mu$ and given dominating set $\mathcal{X}_D$ with $|\mathcal{X}_D| = \gamma$ and accuracy $\epsilon > 0$, Algorithm 3 returns with probability at least $1 - \delta$ an $\epsilon$-optimal policy after*

$$\widetilde{O}\Big( \frac{\gamma H^3}{p_0 \epsilon^2} + \frac{\gamma \widehat{S} H^3}{p_0 \epsilon} + \frac{\mu \widehat{S} H^2}{p_0} \Big) \tag{2}$$

*episodes. Here, $p_0 = \min_{i \in [\gamma]} p^{(i)}$ is possible expected number of visits to the node in the dominating set that is hardest to reach.*

The last term $\frac{\mu \widehat{S} H^2}{p_0}$ is spent in the first phase on learning how to reach the dominating set. If the algorithm did not use the extended MDP for multi-task learning as sketched in Section 4, the sample-complexity for the first phase would be $\frac{\gamma \mu \widehat{S} H^2}{p_0}$ (since we then would pay an additional linear factor in the number of state-action pairs we want to learn to reach).

The first two terms in (2) come from visiting the dominating set uniformly in the second phase. Comparing that to the sample-complexity of Algorithm 1 in Table 1, $\mu$ is replaced by $\frac{\gamma H}{p_0}$ in poly$(\epsilon^{-1})$ terms. This can yield substantial savings when there is a small and easily accessible dominating set, e.g., when $\gamma \ll \frac{\mu p_0}{H}$ and $\epsilon \ll p_0$. There is a gap between the bound above and the sample-complexity lower bound in Table 1 (see also Theorem 5 in Appendix B) , but one can show that a slightly specialized version of the algorithm reduces this gap to a single factor of $H$ in the class of MDPs of the lower bound (by using that $p_0 \leq 1, \hat{S} = 2$ in this class, see Appendix F for details).

**Technical challenges and proof technique.** By building on the analysis of Algorithm 1 and the arguments sketched in Section 4, we first show an intermediate sample complexity bound of $\widetilde{O}\big(\frac{\gamma H^3}{p_0 \epsilon^2} + \frac{\gamma \widehat{S} H^3}{p_0 \epsilon} + \frac{\mu H^2}{p_0^2} + \frac{\mu \widehat{S} H^2}{p_0}\big)$. To remove the undesirable $p_0^{-2}$ term, we adapt the analysis of EULER [11] from regret to sample-complexity which gives that $\mu H^2 p_0^{-2}$ can be replaced by $\mu H v^\star p_0^{-2}$ where $v^\star$ is the average optimal return in the first phase. While one can easily bound $v^\star \leq \max_i p_i$, this does not cancel a factor of $p_0 = \min_i p_i$ in the denominator. Dealing with this max vs. min mismatch is the main technical challenge in our proof and requires a novel induction argument over subset of tasks or episodes (see Appendix F).

## 6 Related work

To the best of our knowledge, we are the first to study RL with feedback graphs in the MDP setting. In the bandit setting, many works on feedback graphs exist, going back to Mannor and Shamir [6]. For stochastic bandits, Caron et al. [25] provided the first regret bound for UCB in terms of clique covering number which was later improved to mas-number [24].[3] Both are gap-dependent bounds as is common in bandits. Recently, the first gap-dependent bounds in the MDP setting were proved for a version of EULER [23, 11]. To keep the analysis simple, we here provided problem-independent bounds. A slight generalization of our technique could yield similar problem-dependent bounds.

Mas-number is the best dependency known for UCB algorithms, but action elimination and exponential weights algorithms can achieve regret bound of $\sqrt{\alpha T}$ in the bandit setting [26, 13]. Buccapatnam

et al. [27] even achieved regret scaling with $\gamma$ by using a dominating set . Unfortunately, all these techniques rely on immediate access to each feedback graph vertex which is unavailable in an MDP.

Albeit designed for different purposes, Algorithm 3 is similar to a concurrently developed algorithm [28] for exploration in absence of rewards. But there is a key technical difference: Algorithm 3 learns how to reach each element of the dominating set jointly, while the approach by Jin et al. [28] learns how to reach each state pair separately. Following Section 4, we expect that, by applying our technique to their setting, one could improve the state space dependency of their sample complexity bound from $\widetilde{O}(S^2/\epsilon^2 + S^4/\epsilon)$ to $\widetilde{O}(S^2/\epsilon^2 + S^3/\epsilon)$.

A common assumption in RL is access to a generative model that allows an algorithm to request reward and the next-state from any state-action pair [29], e.g. for sample-based planning. Our feedback graph setting is much weaker, since, in order to receive a sample transition for a state-action pair $x$, the agent needs to learn to either reach $x$ or an incoming neighbor of $x$ in the feedback graph. One can interpret additional observations through a feedback graph as an explicit way of generalizing from one state-action pair to others while using an unstructured representation of the model and value functions. This is complementary to many existing approaches that generalize by introducing structure into the model representation (for example in factored Markov decision processes) or the value function and policy representation [2, 30]

## 7  Conclusion

We studied the effect of additional observations, stipulated by a feedback graph, on the regret and sample-complexity of episodic RL in tabular MDP setting. Our results show that, when the feedback graph is undirected, optimistic model-based algorithms that just incorporate all available observations into their model update step achieve the minimax-optimal regret bound. We also proved with a new algorithm that exploiting the feedback graph structure by visiting highly informative state-action pairs (dominating set) is possible but fundamentally more difficult in the MDP setting, as compared to the well-studied bandit setting. Our work paves the way for a more extensive study of this setting. Promising directions include a regret analysis for feedback graphs in combination with function approximation motivated by impressive empirical successes [7, 8, 9]. Another question of interest is an analysis of model-free methods [31] with graph feedback which likely requires a very different analysis, as existing proofs hinge on observations arriving in trajectories.

### Broader Impact

This work is of theoretical nature and the presented insights are unlikely to have a direct impact on society at large. That said, it might guide future research with such impact.

### Acknowledgements

AS was an intern at Google Research when the work was performed. The work of MM was partly supported by NSF CCF-1535987, NSF IIS-1618662, and a Google Research Award. KS would like to acknowledge NSF CAREER Award 1750575 and Sloan Research Fellowship. KS was also a part of Google Research as a consultant when the work was performed. The work of YM received funding from the European Research Council (ERC) under the European Union's Horizon 2020 research and innovation program (grant agreement No. 882396), and by the Israel Science Foundation (grant number 993/17).

## Footnotes

[1]Note that simultaneous observations can be dependent. This allows the oracle to generate side observations from the current, possibly noisy, transition without the need for a new sample with independent noise.

[2]Note that this assumption holds in most auxiliary task learning settings and does not trivialize the problem (e.g. see Section 5).

[3]They assume undirected feedback graphs and state their results in terms of independence number.

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
