[Supplementary Material]

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

[4]Note however that the lower confidence bound in EULER is only supposed to satisfy $Q \le Q^\star$ while we here follow the ORLC approach and its analysis and require $Q$ to be a lower confidence bound on the Q-value of the computed policy $Q^\pi$.

[5]To formally satisfy an IPOC guarantee, the algorithm has to output the policy and with a certificate before each episode. We omitted outputting of policy $\pi_k$ and certificate $[\widetilde{\mathit{V}}_{k,1}(s_{k,1}), \widetilde{V}_{k,1}(s_{k,1})]$ after receiving the initial state $s_{k,1}$ in the listing of Algorithm 1 for brevity, but this can be added if readily.

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

# Contents of Appendix

# A Discussion of Feedback Graphs

## A.1 Examples for Applications with Feedback Graphs

We here provide a list of additional applications where the feedback graph model could be useful. These examples are motivation for Markov decision processes with feedback graphs and not necessarily for the specific tabular setting of Algorithm 1 or 3.

- Recommender systems: The goal of recommender systems – to maximize the long-term satisfaction of each user, by choosing which items to recommend given the user history – is a natural setting for RL. Here, RL with feedback graphs can improve the generalization across users, which is still a major challenge for RL [10]. There exist other techniques, such as collaborative filtering, that can predict that certain users would react similarly to certain recommendations, but they have other drawbacks. Incorporating these techniques into the oracle that provides side observations can leverage the complementary strengths of these methods and RL.

- Image augmentation: Image augmentation is a common practice, that has been empirically demonstrated to improve performance [7, 8, 9]. In this case, the states are images and for any state-action pair $(s, a)$, the feedback graph contains an edge to all $(\bar{s}, a)$ such that $\bar{s}$ is a random transformation of $s$.

- Personalized tutoring systems: The goal of personalized tutoring systems is to help students learn a given topic as quickly as possible by optimizing the sequence of explanations and practice problems presented to each student. Here, actions can be the problems to select from and states include descriptors of the specific student. There is a large array of expert knowledge, e.g. from education science, about which groups of students react similarly certain problems. Incorporating this knowledge as side observations and formalizing this setting through a feedback graph over students and problems is a natural fit.

- Autonomous driving: In autonomous driving the goal is to steer a car based on sensory inputs such as a camera view of the street ahead. Similar to image augmentation, an RL algorithm could be provided with additional observations by perturbing the camera view or changing the appearance of certain objects, e.g. to better account for rare appearances. A feedback graph would capture such a data augmentation scheme.

- Robotics: For a concrete example, consider a robot arm grasping different objects and putting them in different bins. In this task, the specific shape of the object is only relevant when the robot hand is close to the object but has not grasped it yet. In all other states, the agent can receive additional observations by filling in different object shapes in the state description of the actual transition. In this case, all such state-action pairs that are identical up to the object shape are connected in the feedback graph.

## A.2 Comparison of Graph Properties

In this section, we provide an extended discussion of the relevant graph properties that govern learning efficiency of RL with feedback graphs. For convenience, we repeat the definitions of the properties from Section 2.2.

- **Mas-number $\mu$**: A set of vertices $\mathcal{V} \subseteq \mathcal{X}$ form an acyclic subgraph if the subgraph $(\mathcal{V}, \{(v, w) \subseteq \mathcal{V} \times \mathcal{V} \colon v \xrightarrow{G} w\})$ of $G$ restricted to $\mathcal{V}$ is loop-free. We call the size of the maximum acyclic subgraph the *mas-number* $\mu$ of $G$.

- **Independence number $\alpha$**: A set of vertices $\mathcal{V} \subseteq \mathcal{X}$ is an independent set if there is no edge between any two nodes of that set: $\forall v, w \in \mathcal{V} \colon v \xcancel{\xrightarrow{G}} w$. The size of the largest independent set is called the *independence number* $\alpha$ of $G$.

- **Domination number $\gamma$**: A set of vertices $\mathcal{V} \subseteq \mathcal{X}$ form a dominating set if there is an edge from a vertex in $\mathcal{V}$ to any vertex in $G$: $\forall x \in \mathcal{X} \; \exists v \in \mathcal{V} \colon v \xrightarrow{G} x$. The size of the smallest dominating set is called the *domination number* $\gamma$.

- **Clique covering number $\mathcal{C}$**: A set of vertices $\mathcal{V} \subseteq \mathcal{X}$ is a clique if there it is a fully-connected subgraph, i.e., for any $x, y \in \mathcal{V} \colon x \xrightarrow{G} y$. A set of such cliques $\{\mathcal{V}_1, \dots \mathcal{V}_n\}$ is called a clique cover if every node is included in at least one of the cliques, i.e., $\mathcal{X} = \bigcup_{i=1}^{n} \mathcal{V}_i$. The size of the smallest clique cover is called the *clique covering number* $\mathcal{C}$.

In addition to the properties appearing our bounds, we here include the clique covering number $\mathcal{C}$ which has been used earlier analyses of UCB algorithms in bandits [25]. One can show that in any graph, the following relation holds

$$|\mathcal{X}| \geq \mathcal{C} \geq \mu \geq \alpha \geq \gamma.$$

For example, $\mathcal{C} \geq \mu$ follows from the fact that no two vertices that form a clique can be part of an acyclic subgraph and thus no acyclic subgraph can be larger than any clique cover. An important class of feedback graphs are symmetric feedback graphs where for each edge $x \xrightarrow{G} y$, there is a back edge $y \xrightarrow{G} x$. In fact, many analyses in the bandit settings assume undirected feedback graphs which is equivalent to symmetric directed graphs. For symmetric feedback graphs, the independence number and mas-number match, i.e., $alpha = \mu$. This is true because acyclic subgraphs of symmetric graphs cannot contain any edges, otherwise the back edge would immediately create a loop. Thus any acyclic subgraph is also an independent set and $\mu \geq \alpha$.

**Examples:**  We now discuss the value of the graph properties in feedback graphs by example (see Figure 3). The graph in Figure 3a consists of two disconnected cliques and thus the clique covering number and the domination number is 2. While the total number of nodes can be much larger – 8 in this example – all graph properties equal the number of cliques in such a graph. In practice, feedback graphs that consists of disconnected cliques occur for example in state abstractions where all $(s, a)$ pairs with matching action and where the state belongs to the same abstract state form a clique. They are examples for a simple structure that can be easily exploited by RL with feedback graphs to substantially reduce the regret.

In the feedback graph in Figure 3b, the vertices are ordered and every vertex is connected to every vertex to the left. This graph is acyclic and hence $\mu$ coincides with the number of vertices but the independence number is 1 as the graph is a clique if we ignore the direction of edges (and thus each independence set can only contain a single node). A concrete example where feedback graphs can exhibit such structure is in tutoring systems where the actions represent the number of practice problems to present to a student in a certain lesson. The oracle can fill in the outcomes (how well the performed on each problem) for all actions that are would have given fewer problems than the chosen action.

Figure 3c shows a star-shaped feedback graph. Here, the center vertex reveals information about all other vertices and thus is a dominating set with size $\gamma = 1$. At the same time, the largest independence set are the tips of the star which is much larger. This is an example where approaches such as Algorithm 3 that leverage a dominating set can learn a good policy with much fewer samples as compared to others that only rely independence sets.

The examples in Figure 3a–3c exhibit structured graphs, but it is important to realize that our results do not rely a specific structure. They can work with any feedback graph and we expect that feedback graphs in practice are not necessarily structured. Figure 3d shows a generic graph where all relevant graph properties are distinct which highlights that even in seemingly unstructured graphs, it is important which graph property governs the learning speed of RL algorithms.

**Values of graph properties in four-room environment.**  To showcase how the graph properties can differ in naturally occurring feedback graphs, we computed their values for a version of the classic four room environment [21] with additional observations. In this version, the oracle provides the agent in each time step with observations for state-action pairs $s, a$ where $s$ in its line of sight and the action matches the agent's current action. The line of sight is defined as all unobstructed states within a rectangle of size 7 in front of the agent (motivated by the observation model in the gym-minigrid suite [32]).

The feedback graph of this example has $|\mathcal{X}| = SA = 376$ vertices but the graph properties are significantly smaller: $\mu \leq 146$, $\alpha = 44$ and $\gamma = 16$.

We determined the value of each graph property by formulating it as the optimal solution to an integer program. Since the exact integer program for mas-number $\mu$ contains $|\mathcal{X}|^3 \approx 53M$ constraints (for each possible cycle of size 3) which is computationally challenging, we used a relaxation that only contains 30% of the constraints. This allows us to give the upper-bound stated above instead of the exact value.

Figure 3: Examples of feedback graphs with different vertex numbers $SA$, mas-number $\mu$, independence number $\alpha$ and domination number $\gamma$.

## B  Regret and Sample-Complexity Lower Bounds

RL in MDPs with feedback graphs is statistically easier due to side observations compared to RL without feedback graphs. Thus, existing lower bounds are not applicable. We now present a new lower-bound that shows that for any given feedback graph, the worst-case expected regret of any learning algorithm has to scale with the size of the largest independent set of at least half of the feedback graph.

**Theorem 4.** *Let $A, N, H, T \in \mathbb{N}$ and $G_1, G_2$ be two graphs with $NA$ and $(N + 1)A$ (disjoint) nodes each. If $H \geq 2 + 2 \log_A N$, then there exists a class of episodic MDPs with $2N + 1$ states, $A$ actions, horizon $H$ and feedback graph $G_1 \cup G_2 := (V(G_1) \cup V(G_2), E(G_1) \cup E(G_2))$ such that the worst-case expected regret of any algorithm after $T$ episodes is at least $\frac{1}{46}\sqrt{\alpha H^2 T}$ when $T \geq \alpha^3/\sqrt{2}$ and $\alpha \geq 2$ is the independence number of $G_1$.*

This regret lower bound shows that, up to a scaling of rewards of order $H$, the statistical difficulty is comparable to the bandit case where the regret lower-bound is $\sqrt{\alpha T}$ [6].

The situation is different when we consider lower bounds in terms of domination number. Theorem 5 below proves that there is a fundamental difference between the two settings:

**Theorem 5.** *Let $\gamma \in \mathbb{N}$ and $p_0 \in (0, 1]$ and $H, S, A \in \mathbb{N}$ with $H \geq 2 \log_A(S/4)$. There exists a family of MDPs with horizon $H$ and a feedback graph with a dominating set of size $\gamma$ and independence set of size $\alpha = \Theta(SA)$. The dominating set can be reached uniformly with probability $p_0$. Any algorithm that returns an $\epsilon$-optimal policy in this family with probability at least $1 - \delta$ has to collect the following expected number of episodes in the worst case*

$$\Omega\left(\frac{\alpha H^2}{\epsilon^2} \ln \frac{1}{\delta} \wedge \left(\frac{\gamma H^2}{p_0 \epsilon^2} \ln \frac{1}{\delta} + \frac{\alpha}{p_0}\right)\right).$$

This lower bound depends on the probability $p_0$ with which the dominating set can be reached and has a dependency $\frac{\alpha}{p_0} \approx \frac{SA}{p_0}$ on the number of states and actions. In bandits, one can easily avoid the linear dependency on number of arms by uniformly playing all actions in the given dominating set $\tilde{\Theta}(\epsilon^{-2})$ times.

We illustrate where the difficulty in MDPs comes from in Figure 5. States are arranged in a tree so that each state at the leafs can be reached by one action sequence. The lower half of state-action pairs at the leafs (red) transition to good or bad terminal states with similar probability. This mimics a bandit with $\Theta(SA)$ arms. There are no side observations available except in state-action pairs of the dominating set (shaded area). Each of them can be reached by a specific action sequence but only with probability $p_0$, otherwise the agent ends up in the bad state.

To identify which arm is optimal in the lower bandit, the agent needs to observe $\Omega(H^2/\epsilon^2)$ samples for each arm. It can either directly play all $\Theta(SA)$ arms or learn about them by visiting the dominating set uniformly. To visit the dominating set once takes $1/p_0$ attempts on average if the agent plays the right action sequence. However, the agent does not know which state-action at the leaf of the tree (blue states) can lead to the dominating set and therefore has to try each of the $\Theta(SA)$ options on average $1/p_0$ times to identify it.

Figure 4: Difficult class of MDPs with a feedback graph and small dominating set. State-action pairs in the red states (bottom) mimic a bandit with $\Theta(SA)$ arms. An agent can observe them directly or by visiting fewer nodes in the dominating set (shaded region). This dominating set is only reachable with small probability $p_0$ and the agent first has to learn which blue state (top) accesses the dominating set. Omitted transitions point to the bad state.

Figure 5: Lower bound construction depicted for $A = 2$ actions: This family of MDPs is equivalent to a Bernoulli bandit with $NA$ arms where rewards are scaled by $\bar{H} = \lfloor H - 1 - \log_A N \rfloor$.

## B.1 Proof for Lower Regret Bound with Independence Number

*Proof of Theorem 4.* We first specify a family of MDPs that are hard to learn with feedback graphs, then show that learning in hard instances of MABs with $\alpha$ arms can be reduced to learning in this family of MDPs and finally use this reduction to lower bound the regret of any agent.

**Family of hard MDPs $\mathcal{M}$:** Without loss of generality, we assume that $N = A^k$ for some $k \in \mathbb{N}$. We consider a family $\mathcal{M}$ of $\alpha$ MDPs which are illustrated in Figure 5. Each MDP in $\mathcal{M}$ has $N$ red states (and $N + 1$ white states) that form the leaves of a deterministic tree with fan-out $A$. This means that each red state is deterministically reachable by a sequence of actions of length $\lceil \log_A N \rceil$. From each red state, the agent transitions to a good absorbing state with certain probability and with remaining probability to a bad absorbing state. All rewards are 0 except in the good absorbing state where the agent accumulates reward of 1 until the end of the episode (for a total of $\bar{H} := H - 1 - \lceil \log_A N \rceil$ time steps).

Let now $G_1$ and $G_2$ be the feedback graphs for the red and white state-actions respectively. Further let $\mathcal{N}$ be an independent set of $G_1$. Each MDP $M_i \in \mathcal{M}$ is indexed by an optimal pair $i = (s^\star, a^\star) \in \mathcal{N}$ of a red state-action pair. When the agent takes $a^\star$ in $s^\star$ it transitions to the good state with probability $\delta + \epsilon$. For all other pairs in $\mathcal{N}$, it transitions to the good state with probability $\delta$. All remaining pairs of red states and actions have probability 0 of reaching the good state. The values of $\delta, \epsilon > 0$ will be specified below.

**Reduction of learning in MAB setting to RL in the MDP setting $\mathcal{M}$:** We now use a reductive argument similar to Mannor and Shamir [6, Theorem 4] to show learning in MABs with $\alpha$ actions cannot me much harder than learning in $\mathcal{M}$.

Let $\mathcal{B}$ be any MDP algorithm and denote by $R_{\mathcal{B},M_i}(T)$ its expected regret after $T$ episodes when applied to problem instance $M_i \in \mathcal{M}$. We can use $\mathcal{B}$ to construct a multi-armed bandit algorithm $\mathcal{B}'$ for a family of MABs $\mathcal{M}'$ indexed by $\mathcal{N}$. Each MAB $M_i' \in \mathcal{M}'$ has $|\mathcal{N}|$ arms, all of which have Bernoulli($\delta$) rewards except $i$ which has Bernoulli($\delta + \epsilon$) rewards. To run $\mathcal{B}'$ on $M_i' \in \mathcal{M}'$, we apply $\mathcal{B}$ to $M_i \in \mathcal{M}$. Whenever $\mathcal{B}$ chooses to execute an episode that visits a $j \in \mathcal{N}$, $\mathcal{B}'$ picks arm $j$ in $M_i'$ and passes on the observed reward as an indicator of whether the good state was reached. When $\mathcal{B}$ chooses to execute an episode that passes through a vertex $x$ of $G_1$ that is not in the independent set $\mathcal{N}$, then $\mathcal{B}'$ pulls all children $\{y \in \mathcal{N} : x \to_{G_1} y\}$ that are in the independent set in an arbitrary order. The observed rewards are again used to construct the observed feedback for $\mathcal{B}$ by interpreting them as indicators for whether the good state was reached.

**Lower bound on regret:** We denote by $T'$ the (random) number of pulls $\mathcal{B}'$ takes until $\mathcal{B}$ has executed $T$ episodes and by $U$ the expected number of times $\mathcal{B}$ plays episodes that do not visit the independent set. The regret of $\mathcal{B}'$ after $T'$ pulls can then be written as

$$R_{\mathcal{B}',M_i'}(T') \leq \frac{R_{\mathcal{B},M_i}(T)}{\bar{H}} + \epsilon\alpha U - \delta U,$$

where the first term $R_{\mathcal{B},M_i}(T)/\bar{H}$ is the regret accumulated from pulls where $\mathcal{B}$ visits the independent set and the second term from the pulls where $\mathcal{B}$ did not visit the independent set. Each such episode incurs $\delta$ regret for $\mathcal{B}$ and up to $\alpha\epsilon$ regret for $\mathcal{B}'$. We rearrange this inequality as

$$\begin{aligned}
R_{\mathcal{B},M_i}(T) \geq &\bar{H}(R_{\mathcal{B}',M_i'}(T') - \epsilon\alpha U + \delta U) \\
&\overset{①}{\geq} \bar{H}(R_{\mathcal{B}',M_i'}(T) - \epsilon\alpha U + \delta U) \\
&\overset{②}{\geq} \bar{H} R_{\mathcal{B}',M_i'}(T) + \bar{H}T[\delta - \epsilon\alpha]^-,
\end{aligned} \tag{3}$$

where ① follows from monotonicity of regret and ② from considering the best case $U \in [0, T]$ for algorithm $\mathcal{B}$. The worst-case regret of $\mathcal{B}'$ in the $\mathcal{M}'$ has been analyzed by Osband and Van Roy [19]. We build on their result and use their Lemma 3 and Proposition 1 to lower bound the regret for $\mathcal{B}'$ as follows

$$\begin{aligned}
\max_i R_{\mathcal{B}',M_i'}(T) &\geq \epsilon T \left(1 - \frac{1}{\alpha} - \epsilon\sqrt{\frac{T}{2\delta\alpha}}\right) \\
&= \frac{1}{4}\sqrt{\frac{\alpha}{2T}}T\left(1 - \frac{1}{\alpha} - \frac{1}{4}\sqrt{\frac{\alpha}{2T}}\sqrt{\frac{4T}{2\alpha}}\right) \\
&= \sqrt{\frac{\alpha T}{32}}\left(\frac{3}{4} - \frac{1}{\alpha}\right),
\end{aligned}$$

where we set $\delta = \frac{1}{4}$ and $\epsilon = \frac{1}{4}\sqrt{\frac{\alpha}{2T}}$ (which satisfy $\epsilon \leq 1 - 2\delta$ required by Proposition 1 for $T \geq \alpha/8$). Plugging this result back into (3) gives a worst-case regret bound for $\mathcal{B}$ of

$$\max_i R_{\mathcal{B},M_i}(T) \geq \sqrt{\frac{\alpha T}{32}}\left(\frac{3}{4} - \frac{1}{\alpha}\right) + \bar{H}T[\delta - \epsilon\alpha]^- \geq \bar{H}\sqrt{\frac{\alpha T}{32}}\left(\frac{3}{4} - \frac{1}{\alpha}\right) \geq \frac{H}{32}\sqrt{\frac{\alpha T}{2}},$$

where we first dropped the second term because $\delta \geq \epsilon\alpha$ for $T \geq \alpha^3/8$ and then used the assumptions $\alpha \geq 2$ and $H \geq 2 + 2\log_A N$. $\qquad\square$

### B.2 Proof for Lower Sample Complexity Bound with Domination Number

*Proof of Theorem 5.* Let $Z = \frac{S}{8}$ and $\bar{Z} = ZA$ which we assume to be integer without loss of generality. The family of MDPs consists of $\bar{Z} \times \bar{Z}$ MDPs, indexed by $(i, j) \in [\bar{Z}]^2$. All MDPs have the same structure:

**MDP Family:** We order $4Z$ states in a deterministic tree so any of the $2Z$ leaf nodes can be reached by a specific action sequence. See Figure 4 for an example with two actions. We split the state-action pairs at the leafs in two sets $\mathcal{B}_1 = \{x_1, \ldots x_{\bar{Z}}\}$ and $\mathcal{B}_2 = \{z_1, \ldots z_{\bar{Z}}\}$, each of size $\bar{Z}$. Playing $x_i$

transitions to the good absorbing state with some probability $g$ and otherwise to the bad absorbing state $b$. The reward is $0$ in all states and actions, except in the good state $g$, where agent receives a reward of $1$. The transition probabilities from $x_i$ depend on the specific MDP. Consider MDP $(j, k)$, then

$$P(g|x_1) = \frac{1}{2} + \frac{\epsilon}{2H}, \qquad \text{and,} \qquad P(g|x_i) = \frac{1}{2} + \frac{\epsilon}{H}\mathbf{1}\{i = k\}.$$

Hence, the first index of the MDP indicates which $x_i$ is optimal. Since the agent will stay in the good state for at least $H/2$ time steps (by the assumption that $H \geq 2\log_A(S/4)$ by assumption), the agent needs to identify which $x_i$ to play in order to identify an $\frac{\epsilon}{4}$-optimal policy. All pairs $z_i$ transition to the bad state deterministically, except for pair $z_j$ in MDPs $(i, j)$. This pair transitions with probability $p_0$ to another tree of states (of size at most $2Z$) which has $\gamma$ state-action pairs at the leafs, denoted by $\mathcal{D} = \{d_1, \ldots, d_\gamma\}$. All pairs in this set transition to the bad state deterministically.

The feedback graph is sparse. There are no edges, except each node $d_i$ has exactly $\frac{\bar{Z}}{\gamma}$ edges (which we assume to be integer for simplicity) to pairs in $\mathcal{B}_1$. No nodes $d_i$ and $d_j$ point to the same node. Hence, $\mathcal{D}$ forms a dominating set of the feedback graph.

**Sample Complexity:** The construction of $\mathcal{B}_1$ is equivalent to the multi-armed bandit instances in Theorem 1 by Mannor and Tsitsiklis [33]. Consider any algorithm and let $o_i$ be the number of observations an algorithm has received for $x_i$ when it terminates. By applying Theorem 1 by Mannor and Tsitsiklis [33], we know that if the algorithm indeed outputs an $\frac{\epsilon}{4}$-optimal policy with probability at least $1 - \delta$ in any instance of the family, it has to collect in instances $(1, j)$ at least the following number of samples in expectation

$$\mathbb{E}_{(1,j)}[o_i] \geq \frac{c_1 H^2}{\epsilon^2}\ln\frac{c_2}{\delta} \tag{4}$$

for some absolute constants $c_1$ and $c_2$. Let $v(x)$ be the number of times the algorithm actually visited a state-action pair $x$. Then $o_i = v(x_i) + v(d_j)$ for $j$ with $d_j \to_G x_i$ because the algorithm can only observe a sample for $x_i$ if it actually visits it or the node in the dominating set. Applying this identity to (4) yields

$$\mathbb{E}_{(1,k)}[v(d_j)] \geq \frac{c_1 H^2}{\epsilon^2}\ln\frac{c_2}{\delta} - \mathbb{E}_{(1,k)}[v(x_i)]$$

for all $d_j$ and $x_i$ with $d_j \to x_i$. Summing over $i \in \bar{Z}$ and using the fact that each $d_j$ is counted $\bar{Z}/\gamma$ times, we get

$$\frac{\bar{Z}}{\gamma}\sum_{j=1}^{\gamma}\mathbb{E}_{(1,k)}[v(d_j)] \geq \frac{c_1\bar{Z}H^2}{\epsilon^2}\ln\frac{c_2}{\delta} - \sum_{i=1}^{\bar{Z}}\mathbb{E}_{(1,k)}[v(x_i)].$$

After renormalizing, we get,

$$\sum_{j=1}^{\gamma}\mathbb{E}_{(1,k)}[v(d_j)] \geq \frac{c_1\gamma H^2}{\epsilon^2}\ln\frac{c_2}{\delta} - \frac{\gamma}{\bar{Z}}\sum_{i=1}^{\bar{Z}}\mathbb{E}_{(1,k)}[v(x_i)].$$

Next, observe that either the algorithm needs to visit nodes in $\mathcal{B}_1$ at least $\frac{c_1\bar{Z}H^2}{2\epsilon^2}\ln\frac{c_2}{\delta}$ times in expectation or nodes in the dominating set $\mathcal{D}$ at least $\frac{c_1\gamma H^2}{2\epsilon^2}\ln\frac{c_2}{\delta}$ times in expectation. The former case gives an expected number of episodes of $\Omega\left(\frac{SAH^2}{\epsilon^2}\ln\frac{1}{\delta}\right)$ which is the second term in the lower-bound to show.

It remains the case where $\sum_{j=1}^{\gamma}\mathbb{E}_{(1,k)}[v(d_j)] \geq \frac{c_1\gamma H^2}{2\epsilon^2}\ln\frac{c_2}{\delta}$. The algorithm can only reach the dominating set through $z_k$ and it can visit only one node in the dominating set per episode. Further, when the algorithm visits $z_k$, it only reaches the dominating set with probability $p_0$. Hence,

$$\mathbb{E}_{(1,k)}[v(z_k)] = p_0\sum_{j=1}^{\gamma}\mathbb{E}_{(1,k)}[v(d_j)] \geq \frac{c_1\gamma H^2}{2p_0\epsilon^2}\ln\frac{c_2}{\delta},$$

but the algorithm may also visit other pairs $z_i$ for $i \neq k$ as well. To see this, consider the expected number of visits to all $z_i$s before the algorithm visits the dominating set for the first time. By Lemma 6, this is at least $\frac{\bar{Z}}{4p_0}$ in the worst case over $k$. This shows that

$$\max_{k \in [\bar{Z}]} \sum_{i=1}^{\bar{Z}} \mathbb{E}_{(1,k)}[v(z_i)] \geq \frac{c_1 \gamma H^2}{2 p_0 \epsilon^2} \ln \frac{c_2}{\delta} - 1 + \frac{\bar{Z}}{4 p_0} = \Omega \left( \frac{\gamma H^2}{p_0 \epsilon^2} \ln \frac{c_2}{\delta} + \frac{SA}{p_0} \right).$$

$\square$

**Lemma 6.** *Consider $k$ biased coins, where all but one coin have probability $0$ of showing heads. Only one coin has probability $p$ of showing head. The identity $i$ of this coin is unknown. The expected number of coin tosses $N$ until the first head is for any strategy*

$$\mathbb{E}[N] \geq \frac{k}{4p}$$

*in the worst case over $i$.*

*Proof.* Let $N$ be the number of coin tosses when the first head occurs and let Alg be a strategy. The quantity of interest is

$$\inf_{\text{Alg}} \max_{i \in [k]} \mathbb{E}_{t(i)}[\mathbb{E}_{R(\text{Alg})}[N]],$$

where $R(\text{Alg})$ denotes the internal randomness of the strategy and $t(i)$ the random outcomes of coin tosses. We first simplify this expression to

$$\inf_{\text{Alg}} \max_{i \in [k]} \mathbb{E}_{t(i)}[\mathbb{E}_{R(\text{Alg})}[N]] \geq \inf_{\text{Alg}} \frac{1}{k} \sum_{i=1}^{k} \mathbb{E}_{t(i)}[\mathbb{E}_{R(\text{Alg})}[N]] = \inf_{\text{Alg}} \mathbb{E}_{R(\text{Alg})} \left[ \frac{1}{k} \sum_{i=1}^{k} \mathbb{E}_{t(i)}[N] \right]$$

$$= \inf_{\text{Alg}} \mathbb{E}_{R(\text{Alg})} \left[ \frac{1}{k} \sum_{i=1}^{k} \sum_{m=0}^{\infty} \mathbb{P}_{t(i)}[N > m] \right]$$

$$= \inf_{\text{Alg}} \mathbb{E}_{R(\text{Alg})} \left[ \sum_{m=0}^{\infty} \frac{1}{k} \sum_{i=1}^{k} \mathbb{P}_{t(i)}[N > m] \right], \tag{5}$$

and derive an explicit expression for $\frac{1}{k} \sum_{i=1}^{k} \mathbb{P}_{t(i)}[N > m]$. Since the strategy and its randomness is fixed, it is reduced to a deterministic sequence of coin choices. That is, for a given number of total tosses $N$, a deterministic strategy is the number of tosses of each coin $n_1, \ldots, n_k$ with $\sum_{i=1}^{k} n_i = N$. Consider any such strategy and let a $n_1, \ldots, n_k$ with $\sum_{i=1}^{k} n_i = m$ be the coins selected up to $m$. If $N > m$, then the first $n_i$ tosses of coin $i$ must be tail. Hence, using the geometric distribution, we can explicitly write the probability of this event as

$$\frac{1}{k} \sum_{i=1}^{k} \mathbb{P}_{t(i)}[N > m] \geq \frac{1}{k} \sum_{i=1}^{k} (1-p)^{n_i} \geq \inf_{n_{1:k}: \; \sum_{i=1}^{k} n_i = m} \frac{1}{k} \sum_{i=1}^{k} (1-p)^{n_i}.$$

The second inequality just considers the worst-case. The expression on the RHS is a convex program over the simplex with a symmetric objective. The optimum can therefore only be attained at an arbitrary corner of the simplex or the center. The value at the center is $(1-p)^{m/k}$ and by Young's inequality, we have

$$(1-p)^{m/k} \leq \frac{((1-p)^{m/k})^k}{k} + \frac{1^{k/k-1}}{k/(k-1)} = \frac{(1-p)^m}{k} + \frac{k-1}{k},$$

where the RHS is the value of the program at a corner. Hence $\frac{1}{k} \sum_{i=1}^{k} \mathbb{P}_{t(i)}[N > m] \geq (1-p)^{m/k}$ holds and plugging this back into (5) gives

$$\inf_{\text{Alg}} \max_{i \in [k]} \mathbb{E}_{t(i)}[\mathbb{E}_{R(\text{Alg})}[N]] \geq \sum_{m=0}^{\infty} (1-p)^{m/k} = \frac{1}{1 - (1-p)^{1/k}} \geq \frac{k}{4p},$$

where the last inequality follows from basic algebra and holds for $p < 0.5$. For $p \geq 0.5$, the worst case is $\frac{k}{4p} \leq \frac{k}{2}$ anyway because that is the expected number of trials until one can identify a coin with $p = 1$.

$\square$

# C  Additional Details on Model-Based RL with Feedback Graphs

Here, we provide additional details and extensions to Algorithm 1 in Section 3.

## C.1  Optimistic Planning and Model Update Subroutines

Algorithm 4 shows the model update subroutine called in Algorithm 2 for each observation while sampling an episode. As briefly alluded to in Section 3, the model statistics $D$ consist of

- Observation count $n(x) \in \mathbb{N}$
- Average immediate reward $\widehat{r}(x) \in [0, 1]$
- Average squared reward $\widehat{r^2}(x) \in [0, 1]$
- Transition frequencies $\widehat{P}(x)) \in [0, 1]^S$

for each state-action pair $x \in \mathcal{X}$. Each are initialized as $n(x) \leftarrow 0, \widehat{r}(x) \leftarrow 0, \widehat{r^2}(x) \leftarrow 0, \widehat{P}(x) \leftarrow [1, 0, \ldots, 0]$ at the beginning of Algorithm 1 and 3.

Algorithm 5 presents the optimistic planning subroutine called by Algorithms 1 and 3. In this procedure, the maximum value is set as $V_h^{\max} = H - h + 1$ for each time step $h$ and notation $\widehat{P}(x)f = \mathbb{E}_{s' \sim P(x)}[f(x)]$ denotes the expectation with respect to the next state distribution of any function $f \colon \mathcal{S} \to \mathbb{R}$ on states.

The `OptimistPlan` procedure computes an optimistic estimate $\widetilde{Q}$ of the optimal Q-function $Q^\star$ by dynamic programming. The policy $\pi$ is chosen greedily with respect to this upper confidence bound $\widetilde{Q}$. In addition, a pessimistic estimate $Q$ of the Q-function of this policy $Q^\pi$ is computed (lower confidence bound) analogously to $\widetilde{Q}$. The two estimates only differ in the sign of the reward bonus $\psi_h$. Up to the specific form of the reward bonus $\psi_h(x)$, this procedure is identical to the policy computation in ORLC [12] and EULER [11].[4]

---

**Algorithm 4:** Model Update Routine

1 **function** UpdateModel $(D, x, r, s')$:
2      $(n, \widehat{r}, \widehat{r^2}, \widehat{P}) \leftarrow D$;
3      $n(x) \leftarrow n(x) + 1, \qquad\qquad \widehat{P}(x) \leftarrow \frac{n(x)-1}{n(x)} \widehat{P}(x) + \frac{1}{n(x)} e_{s'},$
     $\widehat{r}(x) \leftarrow \frac{n(x)-1}{n(x)} \widehat{r}(x) + \frac{1}{n(x)} r, \qquad \widehat{r^2}(x) \leftarrow \frac{n(x)-1}{n(x)} \widehat{r^2}(x) + \frac{1}{n(x)} r^2,$
4        where $e_{s'} \in \{0, 1\}^S$ has 1 on the $s'$-th position;
     **return** : $(n, \widehat{r}, \widehat{r^2}, \widehat{P})$

---

## C.2  Runtime Analysis

Just as in learning without graph feedback, the runtime of Algorithm 1 is $O(S\widehat{S}AH)$ per episode where $\widehat{S}$ is a bound on the maximum transition probability support ($S$ in the worst case). The only difference to RL without side observations is that there are additional updates to the empirical model. However, sampling an episode and updating the empirical model requires $O(HSA)$ computation as there are $H$ time steps and each can provide at most $|\mathcal{X}| \le SA$ side observations. This is still dominated by the runtime of optimistic planning $O(S\widehat{S}AH)$. If the feedback graph is known ahead of time, one might be able to reduce the runtime, e.g., by maintaining only one model estimate for state-action pairs that form a clique in the feedback graph with no incoming edges. Then is suffices to only compute statistics of a single vertex per clique.

**Algorithm 5:** Optimistic Planning Routine

---

1 **function** OptimistPlan$(D)$**:**

2 $\quad (n, \widehat{r}, \widehat{r^2}, \widehat{P}) \leftarrow D;$

3 $\quad$ Set $\widetilde{V}_{H+1}(s) \leftarrow 0; \quad \widetilde{\mathit{V}}_{H+1}(s) \leftarrow 0 \quad \forall s \in \mathcal{S};$

4 $\quad$ **for** $h = H$ **to** $1$ **and** $s \in \mathcal{S}$ **do** $\qquad$ *// optimistic planning with upper and lower confidence bounds*

5 $\qquad$ **for** $a \in \mathcal{A}$ **do**

6 $\qquad\quad x \leftarrow (s, a);$

$\qquad\quad$ /* Compute reward bonus */

7 $\qquad\quad \eta \leftarrow \sqrt{\widehat{r^2}(x) - \widehat{r}(x)^2} + \sqrt{\widehat{P}(x)(\widetilde{V}_{h+1}^2) - (\widehat{P}(x)\widetilde{V}_{h+1})^2} \qquad$ ; *// Reward and next state variance*

8 $\qquad\quad \psi_h(x) \leftarrow O\left(\frac{1}{H}\widehat{P}(x)[\widetilde{V}_{h+1} - \widetilde{\mathit{V}}_{h+1}] + \sqrt{\frac{\eta}{n(x)}\ln\frac{|\mathcal{X}|H\ln n(x)}{\delta}} + \frac{\widehat{S}H^2}{n(x)}\ln\frac{|\mathcal{X}|H\ln n(x)}{\delta}\right)$

$\qquad\qquad$ ;

$\qquad\quad$ /* Bellman backup of upper and lower confidence bounds */

9 $\qquad\quad \widetilde{Q}_h(x) \leftarrow 0 \vee \ (\widehat{r}(x) + \widehat{P}(x)\widetilde{V}_{h+1} + \psi_h(x)) \ \wedge V_h^{\max};$ $\qquad\qquad$ *// UCB of $Q_h^\star$*

10 $\qquad\quad \widetilde{\mathit{Q}}_h(x) \leftarrow 0 \vee \ (\widehat{r}(x) + \widehat{P}(x)\widetilde{\mathit{V}}_{h+1} - \psi_h(x)) \ \wedge V_h^{\max};$ $\qquad\qquad$ *// LCB of $Q_h^\pi \geq 0$*

11 $\qquad$ **end**

$\qquad$ /* Compute greedy policy of UCB */

12 $\qquad \pi(s, h) \leftarrow \operatorname{argmax}_a \widetilde{Q}_h(s, a);$

13 $\qquad \widetilde{V}_h(s) \leftarrow \widetilde{Q}_h(s, \pi(h)); \qquad \widetilde{\mathit{V}}_h(s) \leftarrow \widetilde{\mathit{Q}}_h(s, \pi(h));$

14 $\quad$ **end**

$\quad$ **return :** $\pi, \widetilde{V}_h, \widetilde{\mathit{V}}_h$

---

## C.3 Sample Complexity

Since Algorithm 1 is a minor modification of ORLC, it follows the IPOC framework [12] for accountable reinforcement learning.[5] As a result, we can build on the results for algorithms with cumulative IPOC bounds [17, Proposition 2] and show that our algorithm satisfies a sample-complexity guarantee:

**Corollary 1** (PAC-style Bound)**.** *For any episodic MDP with state-actions $\mathcal{X}$, horizon $H$ and feedback graph $G$, with probability at least $1 - \delta$ for all $\epsilon > 0$ jointly, Algorithm 1 can output a certificate with $\widetilde{V}_{k',1}(s_{k',1}) - \widetilde{\mathit{V}}_{k',1}(s_{k',1})$ for some episode $k'$ within the first*

$$k' = O\left(\frac{MH^2}{\epsilon^2}\ln^2\frac{H|\mathcal{X}|}{\epsilon\delta} + \frac{M\widehat{S}H^2}{\epsilon}\ln^3\frac{H|\mathcal{X}|}{\epsilon\delta}\right)$$

*episodes. If the initial state is fixed, such a certificate identifies an $\epsilon$-optimal policy.*

The proof of this corollary is available in Section D.6

## C.4 Generalization to Stochastic Feedback Graphs

As presented in Section 2.2, we assumed so far that the feedback graph $G$ is fixed and identical in all episodes. We can generalize our results and consider stochastic feedback graphs where the existence of an edge in the feedback graph in each episode is drawn independently (from other episodes and edges). This means the oracle provides a side observation for another state-action pair only with a certain probability. We formalize this as the feedback graph $G_k$ in episode $k$ to be an independent sample from a fixed distribution where the probability an each edge is denoted as

$$q(x, x') := \mathbb{P}(x \xrightarrow{G_k} x').$$

This model generalizes the well-studied Erdős–Rényi model [e.g. 27] because different edges can have different probabilities. This can be used as a proxy for the strength of the user's prior. One could

for example choose the probability of states being connected to decreases with their distance. This would encode a belief that nearby states behave similarly.

Algorithm 1 can be directly applied to stochastic feedback graphs and as our analysis will show the bound in Theorem 1 still holds as long as the mas-number $\mu$ is replaced by

$$\bar{\mu} = \inf_{\epsilon \in (0,1]} \frac{\mu(G_{\geq \epsilon})}{\epsilon}$$

where $G_{\geq \nu}$ is the feedback graph that only contains an edge if its probability is at least $\nu$, i.e., $x \overset{G_{\geq \nu}}{\rightarrow} x'$ if and only if $q(x, x') \geq \nu$ for all $x, x' \in \mathcal{X}$. The quantity $\bar{\mu}$ generalizes the mas-number of deterministic feedback graphs where $q$ is binary and thus $\mu = \bar{\mu}$.

## C.5 Generalization to Side Observations with Biases

While there are often additional observations available, they might not always have the same quality as the observation of the current transition [34]. For example in environments where we know the dynamics and rewards change smoothly (e.g. are Lipschitz-continuous), we can infer additional observations from the current transition but have error that increases with the distance to the current transition. We thus also consider the case where each feedback graph sample $(x, r, s', \epsilon')$ also comes with a bias $\epsilon' \in \mathbb{R}$ and the distributions $\tilde{P}_R, \tilde{P}$ of this sample satisfy

$$|\mathbb{E}_{r \sim \tilde{P}_R}[r] - \mathbb{E}_{r \sim P_R(x)}[r]| \leq \epsilon' \quad \text{and} \quad \|\tilde{P} - P(x)\|_1 \leq \epsilon'.$$

To allow biases in side observations, we adjust the bonuses in Line 8 of Algorithm 1 to

$$\psi_h(x) + \tilde{O}\left(\sqrt{\frac{H\widehat{\epsilon}(x)}{n(x)} \ln \frac{|\mathcal{X}|H \ln n(x)}{\delta}} + H\widehat{\epsilon}(x)\right)$$

for each state-action pair $x$ where $\widehat{\epsilon}(x)$ is the average bound on bias in all observations of this $x$ so far. We defer the presentation of the full algorithm with these changes to the next section but first state the main result for learning with biased side observations here. The following theorem shows that the algorithm's performance degrades smoothly with the maximum encountered bias $\epsilon_{\max}$:

**Theorem 7** (Regret bound with biases). *In the same setting as Theorem 1 but where samples can have a bias of at most $\epsilon_{\max}$, the cumulative certificate size and regret are bounded with probability at least $1 - \delta$ for all $T$ by*

$$O\left(\sqrt{\mu H^2 T} \ln \frac{H|\mathcal{X}|T}{\delta} + \mu \hat{S} H^2 \ln^3 \frac{H|\mathcal{X}|T}{\delta} + \sqrt{\mu H^3 T \epsilon_{\max}} \ln \frac{|\mathcal{X}|HT}{\delta} + H^2 T \epsilon_{\max}\right).$$

If $T$ is known, the algorithm can be modified to ignore all observations with bias larger than $T^{-1/2}$ and still achieve order $\sqrt{T}$ regret by effectively setting $\epsilon_{\max} = O(T^{1/2})$ (at the cost of increase in $\mu$).

## C.6 Generalized Algorithm and Main Regret Theorem

We now introduce a slightly generalized version of Algorithm 1 that will be the basis for our theoretical analysis and all results for Algorithm 1 follow as special cases. This algorithm, given in Algorithm 6 contains numerical values for all quantities – as opposed to $O$-notation – and differs from Algorithm 1 in the following aspects:

1. **Allowing Biases:** While Algorithm 1 assumes that the observations provided by the feedback graph are unbiased, Algorithm 6 allows biased observations where the bias (for every sample) is bounded by some $\epsilon' \geq 0$ (see Section C.5). For the unbiased case, one can set $\epsilon'$ or the average bias $\hat{\epsilon}$ as 0 throughout.

2. **Value Bounds:** While the `OptimistPlan` subroutine of Algorithm 1 in Algorithm 5 only uses the trivial upper-bound $V_h^{\max} = H - h + 1$ to clip the value estimates, Algorithm 6 uses upper-bounds $Q_h^{\max}(x)$ and $V_{h+1}^{\max}(x)$ that can depend on the given state-action pair $x$. This is useful in situations where one has prior knowledge on the optimal value for particular states and can a smaller value bound than the worst case bound of $H - h + 1$. This is the case in Algorithm 3, where we apply an instance of Algorithm 6 to the extended MDP with different reward functions per task.

We show that Algorithm 6 enjoys the IPOC bound (see Dann et al. [12]) in the theorem below. This is the main theorem and other statements follow as a special case. The proof can be found in the next section.

**Theorem 8** (Main Regret / IPOC Theorem). *For any tabular episodic MDP with episode length $H$, state-action space $\mathcal{X} \subseteq \mathcal{S} \times \mathcal{A}$ and directed, possibly stochastic, feedback graph $G$, Algorithm 6 satisfies with probability at least $1 - \delta$ a cumulative IPOC bound for all number of episodes $T$ of*

$$
O\left( \sqrt{\bar{\mu} H \sum_{k=1}^{T} V_1^{\pi_k}(s_{k,1}) \ln \frac{|\mathcal{X}|HT}{\delta}} + \bar{\mu}\widehat{S}Q^{\max} H \ln^3 \frac{|\mathcal{X}|HT}{\delta} + \sqrt{\bar{\mu} H^3 T \epsilon_{\max}} \ln \frac{|\mathcal{X}|HT}{\delta} + H^2 T \epsilon_{\max} \right),
$$

*where $\bar{M} = \inf_\nu \frac{\mu(G_{\geq \nu})}{\nu}$ and $\mu(G_{\geq \nu})$ is the mas-number of a feedback graph that only contains edges that have probability at least $\nu$. Parameter $\widehat{S} \leq S$ denotes a bound on the number of possible successor states of each $x \in \mathcal{X}$. Further, $Q^{\max} \leq H$ is a bound on all value bounds used in the algorithm for state-action pairs that have visitation probability under any policy $\pi_k$ for all $k \in [T]$, i.e., $Q^{\max}$ satisfies*

$$
Q^{\max} \geq \max_{k \in [T], h \in [H]} \max_{x:\, w_{k,h}(x) > 0} Q_h^{\max}(x), \qquad and,
$$

$$
Q^{\max} \geq \max_{k \in [T], h \in [H]} \max_{x:\, w_{k,h}(x) > 0} V_{h+1}^{\max}(x).
$$

*The bound in this theorem is an upper-bound on the cumulative size of certificates $\sum_{k=1}^{T} \widetilde{V}_1(s_{k,1}) - \underset{\sim}{V}_1(s_{k,1})$ and on the regret $R(T)$.*

## C.7 Overview of Analysis

We now briefly discuss how the analysis of Theorem 1 differs from existing ones, with the full proof deferred to Section D. Assuming that the value functions estimated in `OptimistPlan` are valid confidence bounds, that is, $\underset{\sim}{V}_{k,h} \leq V_h^\pi \leq V_h^\star \leq \widetilde{V}_{k,h}$ for all $k \in [T]$ and $h \in [H]$, we bound regret as their differences

$$
R(T) \leq \sum_{k=1}^{T} \left[ \widetilde{V}_{k,1}(s_{k,1}) - \underset{\sim}{V}_{k,1}(s_{k,1}) \right] \lesssim \sum_{k=1}^{T} \sum_{h=1}^{H} \sum_{x \in \mathcal{X}} w_{k,h}(x) \left[ H \wedge \left[ \frac{\sigma_{k,h}(x)}{\sqrt{n_k(x)}} + \frac{\widehat{S}H^2}{n_k(x)} \right] \right], \quad (6)
$$

where $\lesssim$ and $\gtrsim$ ignore constants and log-terms and where $\wedge$ denotes the minimum operator. The second step is a bound on the value estimate differences derived through a standard recursive argument. Here, $w_{k,h}(x) = \mathbb{P}\big( (s_{k,h}, a_{k,h}) = x \mid \pi_k, s_{k,1} \big)$ is the probability that policy $\pi_k$ visits $x$ in episode $k$ at time $h$. In essence, each such expected visit incurs regret $H$ or a term that decreases with the number of observations $n_k(x)$ for $x$ so far. In the expression above, $\sigma_{k,h}^2(x) = \mathrm{Var}_{r \sim P_R(x)}(r) + \mathrm{Var}_{s' \sim P(x)}(V_{h+1}^{\pi_k}(s'))$ is the variance of immediate rewards and the policy value with respect to one transition.

In the bandit case, one would now apply a concentration argument to turn $w_{k,h}(x)$ into actual visitation indicators but this would yield a loose regret bound of $\Omega(\sqrt{H^3 T})$ here. Hence, techniques in the analysis of UCB in bandits with graph feedback [24] based on discrete pigeon-hole arguments cannot be applied here without incurring suboptimal regret in $H$. Instead, we apply a probabilistic argument to the number of observations $n_k(x)$. We show that, with high probability, $n_k(x)$ is not much smaller than the total visitation probability so far of all nodes $x' \in \mathcal{N}_G(x) := \{x\} \cup \{x' \in \mathcal{X} \colon x' \overset{G}{\to} x\}$ that yield observations for $x$:

$$
n_k(x) \gtrsim \sum_{i=1}^{k} \sum_{x' \in \mathcal{N}_G(x)} w_i(x'), \quad \text{with } w_i(x) = \sum_{h=1}^{H} w_{i,h}(x).
$$

This only holds when $\sum_{i=1}^{k} \sum_{x' \in \mathcal{N}_G(x)} w_i(x') \gtrsim H$. Hence, we split the sum over $\mathcal{X}$ in (6) in $U_k = \left\{ x \in \mathcal{X} \colon \sum_{i=1}^{k} \sum_{x' \in \mathcal{N}_G(x)} w_i(x') \gtrsim H \right\}$ and complement $U_k^c$. Ignoring fast decaying $1/n_k(x)$ terms, this yields

$$
(6) \lesssim \sum_{k=1}^{T} \left[ \sum_{x \in U_k^c} w_k(x) H + \sum_{x \in U_k} \sum_{h=1}^{H} w_{k,h}(x) \frac{\sigma_{k,h}(x)}{\sqrt{n_k(x)}} \right]
$$

**Algorithm 6:** Optimistic model-based RL algorithm for biased side observations

---

    **input** : failure tolerance $\delta \in (0, 1]$, state-action space $\mathcal{X}$, episode length $H$

    **input** : known bound on maximum transition support $\widehat{S} \leq \|P(x)\|_0 \leq S$

    **input** : known bounds on value $V_{h+1}^{\max}(x) \leq H$ and $Q_h^{\max}(x) \leq H$ with
                $V_{h+1}^{\max}(x) \geq \max_{s':P(s'|x)>0} V_{h+1}^{\star}(s')$ and $Q_h^{\max}(x) \geq Q_h^{\star}(x)$

**1**   $\phi(n) := 1 \wedge \sqrt{\frac{0.52}{n}\left(1.4 \ln\ln(e \vee 2n) + \ln\frac{5.2 \times |\mathcal{X}|(4\widehat{S}+5H+7)}{\delta}\right)} = \Theta\left(\sqrt{\frac{\ln\ln n}{n}}\right)$;

**2**   Initialize $n_1(x) \leftarrow 0$, $\widehat{\epsilon}_1(x) \leftarrow 0$, $\widehat{r}_1(x) \leftarrow 0$ $\widehat{r^2}_1(x) \leftarrow 0$, $\widehat{P}_1(x) \leftarrow e_1 \in \{0,1\}^S$ for all $x \in \mathcal{X}$;

    */\* Main loop \*/*

**3**   **for** *episode* $k = 1, 2, 3, \ldots$ **do**

**4**      $\pi_k, \widetilde{V}_{k,h}, \underset{\sim}{V}_{k,h} \leftarrow \texttt{OptimistPlan}(n_k, \widehat{r}_k, \widehat{r^2}_k, \widehat{P}_k, \widehat{\epsilon}_k)$;

**5**      Receive initial state $s_{k,1}$;

**6**      $n_{k+1}, \widehat{r}_{k+1}, \widehat{r^2}_{k+1}, \widehat{P}_{k+1}, \widehat{\epsilon}_{k+1} \leftarrow \texttt{SampleEpisode}(\pi_k, s_{k,1}, n_k, \widehat{r}_k, \widehat{r^2}_k, \widehat{P}_k, \widehat{\epsilon}_k)$;

**7**   **end**

    */\* Optimistic planning subroutine with biases \*/*

**8**   **function** $\texttt{OptimistPlan}(n, \widehat{r}, \widehat{r^2}, \widehat{P}, \widehat{\epsilon})$:

**9**      $\widetilde{V}_{H+1}(s) = 0$; $\underset{\sim}{V}_{H+1}(s) = 0$    $\forall s \in \mathcal{S}, k \in \mathbb{N}$;

**10**      **for** $h = H$ **to** $1$ **and** $s \in \mathcal{S}$ **do**

**11**          **for** $a \in \mathcal{A}$ **do**

**12**              $x \leftarrow (s, a)$;

**13**              $\eta \leftarrow \sqrt{\widehat{r^2}(x) - \widehat{r}(x)^2} + 2\sqrt{\widehat{\epsilon}(x)}H + \sigma_{\widehat{P}(x)}(\widetilde{V}_{h+1})$;

**14**              $\psi_h(x) \leftarrow$
                 $4\eta\phi(n(x)) + 53\widehat{S}HV_{h+1}^{\max}(x)\phi(n(x))^2 + \frac{1}{H}\widehat{P}(x)(\widetilde{V}_{h+1} - \underset{\sim}{V}_{h+1}) + (H+1)\widehat{\epsilon}(x)$;

**15**              $\widetilde{Q}_h(x) \leftarrow 0 \vee (\widehat{r}(x) + \widehat{P}(x)\widetilde{V}_{h+1} + \psi_h(x)) \wedge Q_h^{\max}(x)$;   *// UCB of $Q_h^{\star} \leq V_h^{\max} \leq H$*

**16**              $\underset{\sim}{Q}_h(x) \leftarrow 0 \vee (\widehat{r}(x) + \widehat{P}(x)\underset{\sim}{V}_{h+1} - \psi_h(x)) \wedge Q_h^{\max}(x)$;        *// LCB of $Q_h^{\pi} \geq 0$*

**17**          **end**

**18**          $\pi(s, h) \leftarrow \operatorname{argmax}_a \widetilde{Q}_h(s, a)$, $\widetilde{V}_h(s) \leftarrow \widetilde{Q}_h(s, \pi(h))$, $\underset{\sim}{V}_h(s) \leftarrow \underset{\sim}{Q}_h(s, \pi(h))$;

**19**      **end**

         **return** : $\pi, \widetilde{V}_h, \underset{\sim}{V}_h$

    */\* Sampling subroutine with biases \*/*

**20**   **function** $\texttt{SampleEpisode}(\pi, n, \widehat{r}, \widehat{r^2}, \widehat{P}, \widehat{\epsilon})$:

**21**      **for** $h = 1, \ldots H$ **do**

**22**          Take action $a_h = \pi(s_h, h)$ and transition to $s_{h+1}$ with reward $r_h$;

**23**          Receive transition observations $\mathcal{O}_h(G)$;

**24**          **for** *transition* $(x, r, s', \epsilon') \in \mathcal{O}_h(G)$ **do**

**25**              $n(x) \leftarrow n(x) + 1$;

**26**              $\widehat{r}(x) \leftarrow \frac{n(x)-1}{n(x)}\widehat{r}(x) + \frac{1}{n(x)}r$,    $\widehat{r^2}(x) \leftarrow \frac{n(x)-1}{n(x)}\widehat{r^2}(x) + \frac{1}{n(x)}r^2$,
             $\widehat{\epsilon}(x) \leftarrow \frac{n(x)-1}{n(x)}\widehat{\epsilon}(x) + \frac{1}{n(x)}\epsilon'$,         $\widehat{P}(x) \leftarrow \frac{n(x)-1}{n(x)}\widehat{P}(x) + \frac{1}{n(x)}e_{s'}$,

**27**              where $e_{s'} \in \{0,1\}^S$ has 1 on the $s'$-th position;

**28**          **end**

         **return** : $(n, \widehat{r}, \widehat{r^2}, \widehat{P}, \widehat{\epsilon})$

**29**      **end**

$$\lesssim \underbrace{\sum_{k=1}^{T} \sum_{x \in U_k^c} w_k(x)\, H}_{(A)} + \underbrace{\sqrt{\sum_{k=1}^{T} \sum_{x \in \mathcal{X}} \sum_{h=1}^{H} w_{k,h}(x) \sigma_{k,h}^2(x)}}_{(B)} \cdot \underbrace{\sqrt{\sum_{k=1}^{T} \sum_{x \in U_k} \frac{w_k(x)}{\sum_{i=1}^{k} \sum_{x' \in \mathcal{N}_G(x)} w_i(x')}}}_{(C)},$$

where the second step uses the Cauchy-Schwarz inequality. The law of total variance for MDPs [29] implies that $(B) \lesssim H\sqrt{T}$. It then remains to bound $(A)$ and $(C)$, which is the main technical innovation in our proof. Observe that both $(A)$ and $(C)$ are sequences of functions that map each node $x$ to a real value $w_k(x)$. While $(A)$ is a thresholded sequence that effectively stops once a node has accumulated enough weight from the in-neighbors, $(C)$ is a self-normalized sequence. We derive the following two novel results to control each term. We believe these could be of general interest.

**Lemma 9** (Bound on self-normalizing real-valued graph sequences). *Let $G = (\mathcal{X}, \mathcal{E})$ be a directed graph with the finite vertex set $\mathcal{X}$ and mas-number $\mu$, and let $(w_k)_{k \in [T]}$ be a sequence of weights $w_k : \mathcal{X} \to \mathbb{R}^+$ such that for all $k$, $\sum_{x \in \mathcal{X}} w_k(x) \leq w_{\max}$. Then, for any $w_{\min} > 0$,*

$$\sum_{k=1}^{T} \sum_{x \in \mathcal{X}} \frac{\mathbf{1}\{w_k(x) \geq w_{\min}\} w_k(x)}{\sum_{i=1}^{k} \sum_{x' \in \mathcal{N}_G(x)} w_i(x')} \leq 2\mu \ln\left(eT \cdot \frac{w_{\max}}{w_{\min}}\right),$$

*where $\mathcal{N}_G(x) = \{x\} \cup \{y \in \mathcal{X} \mid y \overset{G}{\to} x\}$ denotes all vertices with an edge to $x$ in $G$ and $x$ itself.*

**Lemma 10.** *Let $G = (\mathcal{X}, \mathcal{E})$ be a directed graph with vertex set $\mathcal{X}$ and let $w_k$ be a sequence of weights $w_k \colon \mathcal{X} \to \mathbb{R}^+$. Then, for any threshold $C \geq 0$,*

$$\sum_{x \in \mathcal{X}} \sum_{k=1}^{\infty} w_k(x) \cdot \mathbf{1}\Big\{ \sum_{i=1}^{k} \sum_{x' \in \mathcal{N}_G(x)} w_i(x') \leq C \Big\} \leq \mu C$$

*where $\mathcal{N}_G(x)$ is defined as in Lemma 9.*

We apply Lemma 9 and Lemma 10 to get the bounds $(A) \lesssim \mu H$ and $(C) \lesssim \sqrt{\mu}$ respectively. Plugging these bounds back in (6) yields the desired regret bound. Note that Lemmas 9 and 10 operate on a sequence of node weights as opposed to one set of node weights as in the analyses of exponential weights algorithms [13]. The proof of Lemma 9 uses a potential function and a pigeon-hole argument. The proof for Lemma 10 relies on a series of careful reduction steps, first to integer sequences and then to certain binary sequences and finally a pigeon-hole argument. (full proofs are deferred to Appendix E).

### C.8 On Mas-Number Dependency of Optimistic Algorithms

Our main regret bound in Theorem 8 depends on the mas-number $\mu$ which can be larger than the independence number $\alpha$ in the statistical lower-bound in Theorem 4 when the feedback graph is directed. Whether this mismatch is due to the analysis or a limitation of the algorithm itself is an open question. We can however see through a brief argument that a non-randomized algorithm such as ORLC or EULER cannot avoid a dependency on $\mu$ in the constant term:

Consider a multi-armed bandit instance with $S = 1$ states (but multiple actions) and horizon $H = 1$. The feedback graph over actions is the ordered graph in Figure 3 (b). Without breaking ties in a randomized way in the optimistic planning routine, an optimistic algorithm may try actions from right to left and thus only receive additional observations of actions that it has already visited. Thus, the algorithm needs $\mu = A$ rounds in order to receive at least one observation for each action. This indicates that such algorithms have to suffer an additive $\mu$-dependency in their regret unless tie-breaking is randomized.

## D  Analysis of Model-Based RL with Feedback Graphs

Before presenting the proof of the main Theorem 8 stated in the previous section, we show that Theorem 1 and Theorem 7 indeed follow from Theorem 8:

**Proof of Theorem 1.**

*Proof.* We will reduce from the bound in Theorem 8. We start by setting the bias in Theorem 1 to zero by plugging in $\epsilon_{\max} = 0$. Next, we set the worst-case value $Q^{\max} = H$. Next, we set the thresholded mas-number of the stochastic graph $\bar{\mu}$ to the mas-number $\mu$ of deterministic graphs (by setting $\nu = 1$ in the definition of $\bar{\mu}$). Finally, we upper-bound the initial values for all played policies by the maximum value of $H$ rewards, i.e.,

$$\sum_{k=1}^{T} V_1^{\pi_k}(s_{k,1}) \leq TH.$$

Plugging all of the above in the statement of Theorem 8, we get that Algorithm 1 satisfies the IPOC bound of

$$O\left(\sqrt{\mu H^2 T} \ln \frac{|\mathcal{X}|HT}{\delta} + \widehat{S}\mu H^2 \ln^3 \frac{|\mathcal{X}|HT}{\delta}\right).$$

$\square$

**Proof of Theorem 7.**

*Proof.* The proof follows similar to the proof of Theorem 1 (above), while setting $\epsilon_{\max} \neq 0$. Following Theorem 8, this yields additional regret / cumulative certificate size of at most

$$O\left(\sqrt{\bar{\mu} H^3 T \epsilon_{\max}} \ln \frac{|\mathcal{X}|HT}{\delta} + H^2 T \epsilon_{\max}\right).$$

$\square$

**Proof of the main theorem.** The proof of our main result, Theorem 8, is provided in parts in the following subsections:

- **Section D.1** considers the event in which the algorithm performs well. The technical lemmas therein guarantee that this event holds with high probability.
- **Section D.2** quantifies the amount of cumulative bias in the model estimates and other relevant quantities.
- **Section D.3** proves technical lemmas that establish that `OptimistPlan` always returns valid confidence bounds for the value functions.
- **Section D.4** bounds how far apart can the confidence bounds provided by `OptimistPlan` can be for each state-action pair.
- **Section E** contains general results on self-normalized sequences on nodes of graphs that only depend on the structure of the feedback graph.
- **Section D.5** connects all the results from the previous sections into the proof of Theorem 8.

### D.1 High-Probability Arguments

In the following, we establish concentration arguments for empirical MDP models computed from data collected by interacting with the corresponding MDP (with the feedback graph).

We first define additional notation. To keep the definitions uncluttered, we will use the unbiased versions of the empirical model estimates and bound the effect of unbiasing in Section D.2 below. The unbiased model estimates are defined as

$$\bar{r}_k(x) = \widehat{r}_k(x) - \frac{1}{n_k(x)} \sum_{i=1}^{n_k(x)} \bar{\epsilon}_i(x),$$

$$\bar{P}_k(s'|x) = \widehat{P}_k(s'|x) - \frac{1}{n_k(x)} \sum_{i=1}^{n_k(x)} \bar{\epsilon}_i(x, s') \tag{7}$$

where $\bar{\epsilon}_i(x)$ is the bias of the $i^{\text{th}}$ reward observation $r_i$ for $x$ and $\bar{\epsilon}_i(x, s')$ is the bias of the $i^{\text{th}}$ transition observation of $s'$ for $x$. Recall that $\bar{\epsilon}_i(x)$ and $\bar{\epsilon}_i(x, s')$ are unknown to the algorithm, which, however, receives an upper bound $\epsilon'_i$ on $|\bar{\epsilon}_i(x)|$ and $\sum_{s'\in\mathcal{S}}|\bar{\epsilon}_i(x, s')|$ for each observation $i$. Additionally, for any probability parameter $\delta' \in (0, 1)$, define the function

$$\phi(n) := 1 \wedge \sqrt{\frac{0.52}{n}\left(1.4 \ln\ln(e \vee 2n) + \ln\frac{5.2}{\delta'}\right)} = \Theta\left(\sqrt{\frac{\ln\ln n}{n}}\right). \tag{8}$$

We now define several events for which we can ensure that our algorithms exhibit good behavior with.

**Events regarding immediate rewards.**
The first two event $\mathsf{E}^{\mathrm{R}}$ and $\mathsf{E}^{\mathrm{RE}}$ are the concentration of (unbiased) empirical estimates $\bar{r}_k(x)$ of the immediate rewards around the population mean $r(x)$ using a Hoeffding and empirical Bernstein bound respectively, i.e.,

$$\mathsf{E}^{\mathrm{R}} = \left\{\forall\, k \in \mathbb{N}, x \in \mathcal{X} : |\bar{r}_k(x) - r(x)| \leq \phi(n_k(x))\right\},$$

$$\mathsf{E}^{\mathrm{RE}} = \left\{\forall k \in \mathbb{N}, x \in \mathcal{X} : |\bar{r}_k(x) - r(x)| \leq \sqrt{8\overline{\mathrm{Var}}_k(r|x)}\phi(n_k(x)) + 7.49\phi(n_k(x))^2\right\},$$

where the unbiased empirical variance is defined as $\overline{\mathrm{Var}}_k(r|x) = \frac{1}{n_k(x)}\sum_{i=1}^{n_k(x)}\left(r_i - \bar{\epsilon}_i(x) - \bar{r}_k(x)\right)^2$. The next event ensures that the unbiased empirical variance estimates concentrate around the true variance $\mathrm{Var}(r|x)$

$$\mathsf{E}^{\mathrm{Var}} = \left\{\forall k \in \mathbb{N}, x \in \mathcal{X} : \sqrt{\overline{\mathrm{Var}}_k(r|x)} \leq \sqrt{\mathrm{Var}(r|x)} + \sqrt{\frac{2\ln(\pi^2 n^2/6\delta')}{n}}\right\}.$$

**Events regarding state transitions.** The next two events concern the concentration of empirical transition estimates. We consider the unbiased estimate of the probability to encounter state $s'$ after state-action pair $x$ as defined in Equation (7). As per Bernstein bounds, they concentrate around the true transition probability $P(s'|x)$ as

$$\mathsf{E}^{\mathrm{P}} = \left\{\forall\, k \in \mathbb{N}, s' \in \mathcal{S}, x \in \mathcal{X} : |\bar{P}_k(s'|x) - P(s'|x)| \leq \sqrt{4P(s'|x)}\phi(n_k(x)) + 1.56\phi(n_k(x))^2\right\},$$

$$\mathsf{E}^{\mathrm{PE}} = \left\{\forall\, k \in \mathbb{N}, s' \in \mathcal{S}, x \in \mathcal{X} : |\bar{P}_k(s'|x) - P(s'|x)| \leq \sqrt{4\bar{P}_k(s'|x)}\phi(n_k(x)) + 4.66\phi(n_k(x))^2\right\},$$

where the first event uses the true transition probabilities to upper-bound the variance and the second event uses the empirical version. Both events above treat the probability of transitioning to each successor state $s' \in \mathcal{S}$ individually which can be loose in certain cases. We therefore also consider the concentration in total variation in the following event

$$\mathsf{E}^{\mathrm{L}_1} = \left\{\forall\, k \in \mathbb{N}, x \in \mathcal{X} : \|\bar{P}_k(x) - P(x)\|_1 \leq 2\sqrt{\widehat{S}}\phi(n_k(x))\right\},$$

where $\bar{P}_k(x) = (\bar{P}_k(s'|x))_{s'\in\mathcal{S}} \in \mathbb{R}^S$ is the vector of transition probabilities. The event $\mathsf{E}^{\mathrm{L}_1}$ has the typical $\sqrt{\widehat{S}}$ dependency in the RHS of an $\ell_1$ concentration bound. In the analysis, we will often compare the expected the empirical estimate of the expected optimal value of successor state $\bar{P}_k(x)V^\star_{h+1} = \sum_{s'\in\mathcal{S}}\bar{P}_k(s'|x)V^\star_{h+1}(s')$ to its population mean $P(x)V^\star_{h+1}$ and we would like to avoid the $\sqrt{\widehat{S}}$ factor. To this end, the next two events concern this difference explicitly

$$\mathsf{E}^{\mathrm{V}} = \left\{\forall k \in \mathbb{N}, h \in [H], x \in \mathcal{X} : |(\bar{P}_k(x) - P(x))V^\star_{h+1}| \leq \mathrm{rng}(V^\star_{h+1})\phi(n_k(x))\right\}$$

$$\mathsf{E}^{\mathrm{VE}} = \left\{\forall k \in \mathbb{N}, h \in [H], x \in \mathcal{X} :\right.$$

$$\left.\begin{array}{r}|(\bar{P}_k(x) - P(x))V_{h+1}^\star| \le 2\sqrt{\bar{P}_k(x)[(V_{h+1}^\star - P(x)V_{h+1}^\star)^2]}\phi(n_k(x)) \\[2mm] + 4.66\,\mathrm{rng}(V_{h+1}^\star)\phi(n_k(x))^2 \end{array}\right\}$$

where $\mathrm{rng}(V_{h+1}^\star) = \max_{s'\in\mathcal{S}} V_{h+1}^\star(s') - \min_{s'\in\mathcal{S}} V_{h+1}^\star(s')$ is the range of possible successor values. The first event $\mathsf{E}^V$ uses a Hoeffding bound and the second event $\mathsf{E}^{VE}$ uses empirical Bernstein bound.

**Events regarding observation counts.** All events definitions above include the number of observations $n_k(x)$ to each state-action pair $x \in \mathcal{X}$ before episode $k$. This is a random variable itself which depends on how likely it was in each episode $i < k$ to observe this state-action pair. The last events states that the actual number of observations cannot be much smaller than the total observation probabilities of all episodes so far. We denote by $w_i(x) = \sum_{h\in[H]} \mathbb{P}(s_{i,h} = s(x), a_{i,h} = a(x) \mid s_{i,1}, \mathcal{H}_{1:i-1})$ the expected number of *visits* to each state-action pair $x = (s(x), a(x)) \in \mathcal{X} \subseteq \mathcal{S} \times \mathcal{A}$ in the $i$th episode given all previous episodes $\mathcal{H}_{1:i-1}$ and the initial state $s_{i,1}$. The event is defined as

$$\mathsf{E}^{\mathrm{N}} = \left\{\forall\, k \in \mathbb{N}, x \in \mathcal{X}: n_k(x) \ge \frac{1}{2}\sum_{i<k}\sum_{\bar{x}\in\mathcal{X}} q(\bar{x}, x)w_i(\bar{x}) - H\ln\frac{1}{\delta'}\right\}.$$

The following lemma shows that each of the events above is indeed a high-probability event and that their intersection has high probability at least $1 - \delta$ for a suitable choice of the $\delta'$ in the definition of $\phi$ above.

**Lemma 11.** *Consider the data generated by sampling with a feedback graph from an MDP with arbitrary, possibly history-dependent policies. Then, for any $\delta' > 0$, the probability of each of the events, defined above, is bounded as*

*(i)* $\mathbb{P}(\mathsf{E}^{\mathrm{RE}} \cup \mathsf{E}^{\mathrm{R}}) \ge 1 - 4|\mathcal{X}|\delta'$,

*(ii)* $\mathbb{P}(\mathsf{E}^{\mathrm{RE}} \cup \mathsf{E}^{\mathrm{R}}) \ge 1 - 4|\mathcal{X}|\delta'$,

*(iii)* $\mathbb{P}(\mathsf{E}^{\mathrm{Var}}) \ge 1 - |\mathcal{X}|\delta'$,

*(iv)* $\mathbb{P}(\mathsf{E}^{\mathrm{P}}) \ge 1 - 2\widehat{S}|\mathcal{X}|\delta'$,

*(v)* $\mathbb{P}(\mathsf{E}^{\mathrm{PE}}) \ge 1 - 2\widehat{S}|\mathcal{X}|\delta'$,

*(vi)* $\mathbb{P}(\mathsf{E}^{\mathrm{L_1}}) \ge 1 - 2|\mathcal{X}|\delta'$,

*(vii)* $\mathbb{P}\left(\mathsf{E}^{\mathrm{V}}\right) \ge 1 - 2|\mathcal{X}|H\delta'$,

*(viii)* $\mathbb{P}(\mathsf{E}^{\mathrm{VE}}) \ge 1 - 2|\mathcal{X}|H\delta'$,

*(ix)* $\mathbb{P}(\mathsf{E}^{\mathrm{N}}) \le |\mathcal{X}|H\delta'$.

*Further, define the event $E$ as $E := \mathsf{E}^{\mathrm{R}} \cap \mathsf{E}^{\mathrm{RE}} \cap \mathsf{E}^{\mathrm{Var}} \cap \mathsf{E}^{P} \cap \mathsf{E}^{\mathrm{PE}} \cap \mathsf{E}^{\mathrm{L_1}} \cap \mathsf{E}^V \cap \mathsf{E}^{\mathrm{VE}} \cap \mathsf{E}^{\mathrm{N}}$. Then, the event $E$ occurs with probability at least $1 - \delta$, i.e.*

$$\mathbb{P}(E) \ge 1 - \delta,$$

*where $\delta = \delta'|\mathcal{X}|(7 + 4\widehat{S} + 5H)$.*

*Proof.* We bound the probability of occurrence of the events $\mathsf{E}^R, \mathsf{E}^P, \mathsf{E}^{\mathrm{PE}}, \mathsf{E}^{\mathrm{L_1}}, \mathsf{E}^V$ and $\mathsf{E}^{\mathrm{VE}}$ using similar techniques as in the works of Dann et al. [12], Zanette and Brunskill [11] (see for example Lemma 6 in Dann et al. [12]). However, in our setting, we work with a slightly different $\sigma$-algebra to account for the feedback graph, and explicitly leverage the bound on the number of possible successor states $\widehat{S}$. We detail this deviation from the previous works for events $\mathsf{E}^R$ and $\mathsf{E}^{\mathrm{RE}}$ in Lemma 12 (below), and the rest follow analogously.

Further, we bound the probability of occurrence of the event $\mathsf{E}^{\mathrm{N}}$ in Lemma 14. The proof significantly deviates from the prior work, as in our case, the number of observations for any state-action pair is different from the number of visits of the agent to that pair due to the feedback graph. Finally, the bound for the probability of occurrence of $\mathsf{E}^{\mathrm{Var}}$ is given in Lemma 13.

Taking a union bound for all the above failure probabilities, and setting $\delta' = \frac{\delta}{|\mathcal{X}|(7+4\widehat{S}+5H)}$, we get a bound on the probability of occurrence of the event $\mathbb{P}(E)$. $\qquad\square$

**Lemma 12.** *Let the data be generated by sampling with a feedback graph from an MDP with arbitrary, possibly history-dependent policies. Then, the event $\mathsf{E}^{\mathrm{R}} \cap \mathsf{E}^{\mathrm{RE}}$ occurs with probability at-least $1 - 4|\mathcal{X}|\delta'$, or*

$$\mathbb{P}(\mathsf{E}^{\mathrm{R}} \cap \mathsf{E}^{\mathrm{RE}}) \geq 1 - 4|\mathcal{X}|\delta'.$$

*Proof.* Let $\mathcal{F}_j$ be the natural $\sigma$-field induced by everything (all observations and visitations) up to the time when the algorithm has played a total of $j$ actions and has seen which state-action pairs will be observed but not the actual observations yet. More formally, let $k = \lceil \frac{j}{H} \rceil$ and $h = j \mod H$ be the episode and the time index when the algorithm plays the $j^{\text{th}}$ action. Then everything in episodes $1 \ldots k-1$ is $\mathcal{F}_j$-measurable as well as everything up to $s_{k,h}, a_{k,h}$ and $\bar{\mathcal{O}}_{k,h}(G)$ (which $x$ are observed at $k, h$) but not $\mathcal{O}_{k,h}(G)$ (the actual observations) or $s_{k,h+1}$.

We will use a filtration with respect to the stopping times of when a specific state-action pair is observed. To that end, consider a fixed $x \in \mathcal{X}$. Define

$$\tau_i = \inf\left\{ (k-1)H + h \colon \sum_{j=1}^{k-1}\sum_{h'=1}^{H} \mathbf{1}\{x \in \bar{\mathcal{O}}_{j,h'}(G)\} + \sum_{h'=1}^{h} \mathbf{1}\{x \in \bar{\mathcal{O}}_{k,h'}(G)\} \geq i \right\}$$

to be the index $j$ of $\mathcal{F}_j$ where $x$ was observed for the $i^{\text{th}}$ time. Note that, for all $i$, $\tau_i$ are stopping times with respect to $(\mathcal{F}_j)_{j=1}^{\infty}$. Hence, $\mathcal{F}_i^x = \mathcal{F}_{\tau_i} = \{A \in \mathcal{F}_\infty \ : \ A \cap \{\tau_i \leq t\} \in \mathcal{F}_t \ \forall t \geq 0\}$ is a $\sigma$-field. Intuitively, it captures all information available at time $\tau_i$ [35, Sec. 3.3]. Since $\tau_i \leq \tau_{i+1}$, the sequence $(\mathcal{F}_{\tau_i})_{i=1}^{\infty}$ is a filtration as well.

Consider a fixed $x \in \mathcal{X}$ and number of observations $n$. Define $X_i = \mathbf{1}\{\tau_i < \infty\}(r_i - \bar{\epsilon}_i(x) - r(x))$ where $r_i$ is the $i^{\text{th}}$ observation with bias $\bar{\epsilon}_i(x)$ of $x$. By construction $(X_i)_{i=1}^{\infty}$ is adapted to the filtration $(\mathcal{F}_i^x)_{i=1}^{\infty}$. Further, recall that $r(x) = \mathbb{E}[r|(s,a) = x] - \bar{\epsilon}_i$ is the immediate expected reward in $x$ and hence, we one can show that $(X_i)_{i=1}^{\infty}$ is a martingale with respect to this filtration. It takes values in the range $[-r, 1-r]$. We now use a Hoeffding bound and empirical Bernstein bound on $\sum_{i=1}^{n} X_i$ to show that the probability of $\mathsf{E}^{\mathrm{R}}$ and $\mathsf{E}^{\mathrm{RE}}$ is sufficiently large. We use the tools provided by Howard et al. [36] for both concentration bounds. The martingale $\sum_{i=1}^{n} X_i$ satisfies Assumption 1 in Howard et al. [36] with $V_n = n/4$ and any sub-Gaussian boundary (see Hoeffding I entry in Table 2 therein). The same is true for $-\sum_{i=1}^{n} X_i$. Using the sub-Gaussian boundary in Corollary 22 in Dann et al. [12], we get that

$$\left| \frac{1}{n}\sum_{i=1}^{n} X_i \right| \leq 1.44\sqrt{\frac{n}{4n^2}\left(1.4\ln\ln(e \vee n/2) + \ln\frac{5.2}{\delta'}\right)} \leq \phi(n)$$

holds for all $n \in \mathbb{N}$ with probability at least $1 - 2\delta'$. It therefore also holds for all random $n$ including the number of observations of $x$ after $k-1$ episodes. Hence, the condition in $\mathsf{E}^{R}$ holds for all $k$ for a fixed $x$ with probability at least $1 - 2\delta'$. An additional union bound over $x \in \mathcal{X}$ gives $\mathbb{P}(\mathsf{E}^{R}) \geq 1 - 2|\mathcal{X}|\delta'$.

We can proceed analogously for $\mathsf{E}^{\mathrm{RE}}$, except that we use the uniform empirical Bernstein bound from Theorem 4 in Howard et al. [36] with the sub-exponential uniform boundary in Corollary 22 in Dann et al. [12] which yields

$$\left| \frac{1}{n}\sum_{i=1}^{n} X_i \right| \leq 1.44\sqrt{\frac{V_n}{n^2}\left(1.4\ln\ln(e \vee 2V_n) + \ln\frac{5.2}{\delta'}\right)} + \frac{2.42}{n}\left(1.4\ln\ln(e \vee 2V_n) + \ln\frac{5.2}{\delta'}\right) \tag{9}$$

with probability at least $1 - 2\delta'$ for all $n \in \mathbb{N}$. Here, $V_n = \sum_{i=1}^{n} X_i^2 \leq n$. Using the definition of $\phi(n)$ in Equation (8), we can upper-bound the right hand side in the above equation with $2\sqrt{V_n/n}\phi(n) + 4.66\phi(n)^2$. We next bound $V_n$ in the above by the de-biased variance estimate

$$V_n = \sum_{i=1}^{n} X_i^2 = \sum_{i=1}^{n}(r_i - \bar{\epsilon}_i(x) - r(x))^2 = \sum_{i=1}^{n}(r_i - \bar{\epsilon}_i(x) - r(x))^2$$

$$\leq 2\sum_{i=1}^{n}(r_i - \bar{\epsilon}_i(x) - \bar{r}_{\tau_n}(x))^2 + 2n(r(x) - \bar{r}_{\tau_n}(x))^2$$

Applying the definition of event $\mathsf{E}^R$, we know that $|r(x) - \bar{r}_{\tau_n}(x)| \leq \phi(n)$ and thus $V_n/n \leq 2\overline{\mathrm{Var}}_{\tau_n}(r|x) + 2\phi(n)^2$. Plugging this back into (9) yields

$$|\bar{r}_{\tau_n}(x) - r(x)| = \left| \frac{1}{n} \sum_{i=1}^{n} X_i \right| \leq 2\sqrt{2\overline{\mathrm{Var}}(r) + 2\phi(n)^2}\phi(n) + 4.66\phi(n)^2$$

$$\leq \sqrt{8\overline{\mathrm{Var}}(r)}\phi(n) + 7.49\phi(n)^2$$

This is the condition of $\mathsf{E}^{RE}$ which holds for all $n$ and as such $k$ as long as $\mathsf{E}^R$ also holds. With a union bound over $\mathcal{X}$, this yields

$$\mathbb{P}(\mathsf{E}^{RE} \cup \mathsf{E}^R) \geq 1 - 4|\mathcal{X}|\delta'.$$

$\square$

**Lemma 13.** *Let the data be generated by sampling with a feedback graph from an MDP with arbitrary (and possibly history-dependent) policies. Then, the event $\mathsf{E}^{Var}$ occurs with probability at least $1 - |\mathcal{X}|\delta'$, i.e.,*

$$\mathbb{P}(\mathsf{E}^{\mathrm{Var}}) \geq 1 - |\mathcal{X}|\delta'.$$

*Proof.* Consider first a fix $x \in \mathcal{X}$ and let $K$ be the total number of observations for $x$ during the entire run of the algorithm. We denote the observations by $r_i$. Define now $X_i = r_i - \bar{\epsilon}_i(x)$ for $i \in [K]$ and $X_i \sim P_R(x)$ independently. Then by construction $X_i$ is a sequence of i.i.d. random variables in $[0, 1]$. We now apply Theorem 10, Equation 4 by Maurer and Pontil [37] which yields that for any $n$

$$\sqrt{\frac{n}{n-1}\widehat{\mathrm{Var}}(X_n)} \leq \mathrm{Var}(X) + \sqrt{\frac{2\ln(n^2\pi^2/6\delta')}{n-1}}$$

holds with probability at least $1 - \frac{6\delta'}{\pi^2 n^2}$, where $\mathrm{Var}(X)$ is the variance of $X_i$ and $\widehat{\mathrm{Var}}(X_n) = \frac{1}{n}\sum_{i=1}^{n}(X_i - \bar{X}_n)^2$ with $\bar{X}_n = \frac{1}{n}\sum_{i=1}^{n} X_i$ is the empirical variance of the first $n$ samples. By applying a union bound over $n \in \mathbb{N}$, and multiplying by $\sqrt{n/(n-1)}$ we get that

$$\sqrt{\widehat{\mathrm{Var}}(X_n)} \leq \sqrt{\frac{n-1}{n}}\mathrm{Var}(X) + \sqrt{\frac{2\ln(n^2\pi^2/6\delta)}{n}} \leq \mathrm{Var}(X) + \sqrt{\frac{2\ln(n^2\pi^2/6\delta)}{n}}$$

holds for all $n \in \mathbb{N}$ with probability at least $1 - \frac{6\delta'}{\pi^2}\sum_{n=1}^{\infty}\frac{1}{n^2} \geq 1 - \delta'$. We now note that $\mathrm{Var}(X) = \mathrm{Var}(r|x)$ and for each episode $k$, there is some $n$ so that $\overline{\mathrm{Var}}_k(r|x) = \widehat{\mathrm{Var}}(X_n)$. Hence, with another union bound over $x \in \mathcal{X}$, the statement follows. $\square$

**Lemma 14.** *Let the data be generated by sampling with a feedback graph from an MDP with arbitrarily (possibly adversarially) chosen initial states. Then, the event $\mathsf{E}^N$ occurs with probability at-least $1 - H|\mathcal{X}|\delta'$, or*

$$\mathbb{P}(\mathsf{E}^N) \geq 1 - H|\mathcal{X}|\delta'.$$

*Proof.* Consider a fixed $x \in \mathcal{X}$ and $h \in [H]$. We define $\mathcal{F}_k$ to be the sigma-field induced by the first $k-1$ episodes and $s_{k,1}$. Let $X_{k,h} = \mathbf{1}\{x \in \bar{\mathcal{O}}_{k,h}(G)\}$ be the indicator whether $x$ was observed in episode $k$ at time $h$. The probability that this indicator is true given $\mathcal{F}_k$ is simply the probability $w_{k,h}(x) = \mathbb{P}(s_{k,h} = s(x), a_{k,h} = a(x) \mid s_{k,1}, \mathcal{H}_{1:k-1})$ of visiting each $\bar{x} \in \mathcal{X}$ at time $h$ and the probability $q(\bar{x}, x)$ that $\bar{x}$ has an edge to $x$ in the feedback graph in the episode

$$\mathbb{P}(X_{k,h} = 1 \mid \mathcal{F}_k) = \sum_{\bar{x} \in \mathcal{X}_h} q(\bar{x}, x)w_k(\bar{x}).$$

We now apply Lemma F.4 by Dann et al. [38] with $W = \ln\frac{1}{\delta'}$ and obtain that

$$\sum_{i=1}^{k} X_{i,h} \geq \frac{1}{2}\sum_{i=1}^{k}\sum_{\bar{x} \in \mathcal{X}_h} q(\bar{x}, x)w_i(\bar{x}) - \ln\frac{1}{\delta'}$$

for all $k \in \mathbb{N}$ with probability at least $1 - \delta'$. We now take a union-bound over $h \in [H]$ and $x \in \mathcal{X}$ get that $\mathbb{P}(\mathsf{E}^N) \geq 1 - |\mathcal{X}|H\delta'$ after summing over $h \in [H]$ because the total number of observations after $k-1$ episodes for each $x$ is simply $n_k(x) = \sum_{i=1}^{k-1}\sum_{h \in [H]} X_{k,h}$. $\square$

## D.2 Bounds on the Difference of Biased Estimates and Unbiased Estimates

We now derive several helpful inequalities that bound the difference of biased and unbiased estimates.

$$|\bar{r}_k(x) - \widehat{r}_k(x)| = \frac{1}{n_k(x)} \sum_{i=1}^{n_k(x)} \bar{\epsilon}_i(x) \leq \widehat{\epsilon}_k(x)$$

$$\|\bar{P}_k(x) - \widehat{P}_k(x)\|_1 = 2 \max_{\mathcal{B} \subseteq \mathcal{S}} |\bar{P}_k(\mathcal{B}|x) - \widehat{P}_k(\mathcal{B}|x)| = 2 \left| \sum_{s' \in \mathcal{B}} \frac{1}{n_k(x)} \sum_{i=1}^{n_k(x)} \bar{\epsilon}_i(x, s') \right|$$

$$\leq \frac{2}{n_k(x)} \sum_{i=1}^{n_k(x)} \left| \sum_{s' \in \mathcal{B}} \bar{\epsilon}_i(x, s') \right| \leq \widehat{\epsilon}_k(x).$$

The final inequality follows from the fact that $\sum_{s' \in \mathcal{B}} \bar{\epsilon}_i(x, s') \leq \frac{1}{2} \|P(x) - P_i'(x)\|_1 \leq \frac{\epsilon_i'}{2}$ where $P_i'(x)$ denotes the true distribution of the $i^{\text{th}}$ transition observation of $x$ and $\epsilon_i'$ denotes the bias parameter for this observation. From this total variation bound, we can derive a convenient bound on the one-step variance of any "value"-function $f \colon \mathcal{S} \to [0, f_{\max}]$ over the states. In the following, we will use the notation

$$\sigma_P^2(f) := \mathbb{E}_{s' \sim P}[f(s')^2] - \mathbb{E}_{s' \sim P}[f(s')]^2.$$

Using this notation, we bound the difference of the one-step variance of the biased and unbiased state distributions as

$$|\sigma_{\bar{P}_k(x)}^2(f) - \sigma_{\widehat{P}_k(x)}^2(f)| = |\bar{P}_k(x)f^2 - (\bar{P}_k(x)f)^2 - \widehat{P}_k(x)f^2 + (\widehat{P}_k(x)f)^2|$$

$$= |(\bar{P}_k(x) - \widehat{P}_k(x))f^2 + (\bar{P}_k(x) - \widehat{P}_k(x))f(\bar{P}_k(x) + \widehat{P}_k(x))f|$$

$$\leq f_{\max}^2 \|\bar{P}_k(x) - \widehat{P}_k(x)\|_1 + 2f_{\max}^2 \|\bar{P}_k(x) - \widehat{P}_k(x)\|_1 \leq 3f_{\max}^2 \widehat{\epsilon}_k(x).$$
(10)

We also derive the following bounds on quantities related to the variance of immediate rewards. In the following, we consider any number of episodes $k$ and $x \in \mathcal{X}$. To keep notation short, we omit subscript $k$ and argument $x$ below. That is, $r = r(x)$ is the expected reward, $n = n_k(x)$ is the number of observations, which we denote by $r_1, \ldots, r_n$ each. Further $\bar{\epsilon}_i = \bar{\epsilon}_i(x)$ is the bias of the $i$th reward sample for this $x$ and $\epsilon_i \geq \bar{\epsilon}_i$ the accompanying upper-bound provided to the algorithm. We denote by $\widehat{\text{Var}}(r) = \frac{1}{n} \sum_{i=1}^{n} (r_i - \widehat{r})^2$ the empirical variance estimate and by $\overline{\text{Var}}(r) = \overline{\text{Var}}_k(r|x) = \frac{1}{n} \sum_{i=1}^{n} (r_i - \bar{\epsilon}_i - \bar{r})^2$. Thus,

$$\overline{\text{Var}}(r) = \frac{1}{n} \sum_{i=1}^{n} (r_i - \bar{\epsilon}_i - \bar{r})^2 \leq \frac{2}{n} \sum_{i=1}^{n} (r_i - \widehat{r})^2 + \frac{2}{n} \sum_{i=1}^{n} (\widehat{r} - \bar{\epsilon}_i - \bar{r})^2$$

$$= 2\widehat{\text{Var}}(r) + \frac{2}{n} \sum_{i=1}^{n} \left( \left( \frac{1}{n} \sum_{j=1}^{n} \bar{\epsilon}_j \right) - \bar{\epsilon}_i \right)^2$$

$$\leq 2\widehat{\text{Var}}(r) + \frac{2}{n} \sum_{i=1}^{n} \bar{\epsilon}_i^2 \leq 2\widehat{\text{Var}}(r) + \frac{2}{n} \sum_{i=1}^{n} \epsilon_i^2 \leq 2\widehat{\text{Var}}(r) + 2\widehat{\epsilon},$$
(11)

where the last inequality follows from the definition of $\widehat{\epsilon}$ and using the fact that $\bar{\epsilon}_i \leq 1$. The right hand side of the above chain of inequalities is empirically computable and, subsequently, used to derive the reward bonus terms.

Analogously, we can derive a reverse of this bound that upper bounds the computable variance estimate $\widehat{\text{Var}}(r)$ by the unbiased variance estimate $\overline{\text{Var}}(r)$. This is given as

$$\widehat{\text{Var}}(r) = \frac{1}{n} \sum_{i=1}^{n} (r_i - \widehat{r})^2 \leq \frac{2}{n} \sum_{i=1}^{n} (r_i - \bar{\epsilon}_i - \bar{r})^2 + \frac{2}{n} \sum_{i=1}^{n} (\bar{\epsilon}_i - \widehat{r} + \bar{r})^2$$

$$\leq \frac{2}{n} \sum_{i=1}^{n} (r_i - \bar{\epsilon}_i - \bar{r})^2 + \frac{2}{n} \sum_{i=1}^{n} \epsilon_i^2$$

$$= 2\overline{\mathrm{Var}}(r) + \frac{2}{n} \sum_{i=1}^{n} \epsilon_i^2 \cdot \leq 2\overline{\mathrm{Var}}(r) + 2\hat{\epsilon}. \tag{12}$$

### D.3 Correctness of Optimistic Planning

In this section, we provide the main technical results to guarantee that in event $E$ (defined in Lemma 11), the output of `OptimistPlan` are upper and lower confidence bounds on the value functions.

**Lemma 15** (Correctness of Optimistic Planning). *Let $\pi, \widetilde{V}, \underset{\sim}{V}$ be the policy and the value function bounds returned by* `OptimistPlan` *with inputs $n, \widehat{r}, \widehat{r^2}, \widehat{P}, \widehat{\epsilon}$ after any number of episodes $k$. Then, in event $E$ (defined in Lemma 11), the following hold.*

1. *The policy $\pi$ is greedy with respect to $\widetilde{V}$ and satisfies for all $h \in [H]$*

$$\underset{\sim}{V}_h \leq V_h^\pi \leq V_h^\star \leq \widetilde{V}_h.$$

2. *The same chain of inequalities also holds for the Q-estimates used in* `OptimistPlan`, *i.e.,*

$$\underset{\sim}{Q}_h \leq Q_h^\pi \leq Q_h^\star \leq \widetilde{Q}_h.$$

*Proof.* We show the statement by induction over $h$ from $H + 1$ to 1. For $h = H + 1$, the statement holds for the value functions $\underset{\sim}{V}_{H+1}, \widetilde{V}_{H+1}$ by definition. We now assume it holds for $h + 1$. Due to the specific values of $\psi_h$ in `OptimistPlan`, we can apply Lemmas 16 and 17 and get that $\underset{\sim}{Q}_h \leq Q_h^\pi \leq Q_h^\star \leq \widetilde{Q}_h$. Taking the maximum over actions, gives that $\underset{\sim}{V}_h \leq V_h^\pi \leq V_h^\star \leq \widetilde{V}_h$. Hence, the claim follows. The claim that the policy is greedy with respect to $\widetilde{V}$ follows from the definition $\pi(s, h) \in \mathrm{argmax}_a \widetilde{Q}_h(s.a)$. $\square$

**Lemma 16** (Lower bounds admissible). *Let $\pi, \widetilde{V}, \underset{\sim}{V}$ be the policy and the value function bounds returned by* `OptimistPlan` *with inputs $n, \widehat{r}, \widehat{r^2}, \widehat{P}, \widehat{\epsilon}$ after any number of episodes $k$. Consider $h \in [H]$ and $x \in \mathcal{X}$ and assume that $\widetilde{V}_{h+1} \geq V_{h+1}^\star \geq V_{h+1}^\pi \geq \underset{\sim}{V}_{h+1}$ and that the confidence bound width is at least*

$$\psi_h(x) \geq 4\left( \sqrt{\widehat{\mathrm{Var}}(r|x)} + \sigma_{\widehat{P}(x)}(\widetilde{V}_{h+1}) + 2\sqrt{\widehat{\epsilon}(x)}H \right)\phi(n(x)) + 53\widehat{S}HV_{h+1}^{\max}(x)\phi(n(x))^2$$

$$+ \frac{1}{H}\widehat{P}(x)(\widetilde{V}_{h+1} - \underset{\sim}{V}_{h+1}) + (H + 1)\widehat{\epsilon}(x).$$

*Then, in event $E$ (defined in Lemma 11), the lower confidence bound at time $h$ is admissible, i.e.,*

$$Q_h^\pi(x) \geq \underset{\sim}{Q}_h(x).$$

*Proof.* When $\underset{\sim}{Q}_h(x) = 0$, the statement holds trivially. Otherwise, we can decompose the difference of the lower bound and the value function of the current policy as

$$Q_h^\pi(x) - \underset{\sim}{Q}_h(x) \geq \underbrace{r(x) - \bar{r}(x) + (P(x) - \bar{P}(x))V_{h+1}^\star}_{(A)} + \underbrace{(P(x) - \bar{P}(x))(V_{h+1}^\pi - V_{h+1}^\star)}_{(B)}$$

$$+ \bar{P}(x)(V_{h+1}^\pi - \underset{\sim}{V}_{h+1}) + \underbrace{\bar{r}(x) - \widehat{r}(x) + (\bar{P}(x) - \widehat{P}(x))\underset{\sim}{V}_{h+1}}_{(C)} + \underset{\sim}{\psi}_h(x).$$

$$\tag{13}$$

Note that $\bar{P}(x)(V_{h+1}^\pi - \underset{\sim}{V}_{h+1}) \geq 0$ by assumption. We bound the terms (A), (B) and (C) separately as follows.

- **Bound on (A).** Given that the event $E$ occurs, the events $\mathsf{E}^{\mathrm{RE}}$ and $\mathsf{E}^{\mathrm{VE}}$ also hold (see definition of $E$ in Lemma 11). Thus,

$$
\begin{aligned}
|r(x) - & \bar{r}(x) + (P(x) - \bar{P}(x))V_{h+1}^{\star}| \\
&\leq \left( \sqrt{8\overline{\mathrm{Var}}(r|x)} + 2\sqrt{\bar{P}(x)[(V_{h+1}^{\star} - P(x)V_{h+1}^{\star})^2]} \right) \phi(n(x)) \\
&\quad + (4.66 V_{h+1}^{\max}(x) + 7.49)\phi(n(x))^2 \\
&\overset{(i)}{\leq} \left( \sqrt{8\overline{\mathrm{Var}}(r|x)} + \sqrt{12}\sigma_{\bar{P}(x)}(\widetilde{V}_{h+1}) \right) \phi(n(x)) \\
&\quad + (24H\sqrt{\widehat{S}}V_{h+1}^{\max}(x) + 8.13V_{h+1}^{\max}(x) + 7.49)\phi(n(x))^2 \\
&\quad + \frac{1}{2H}\bar{P}(x)(\widetilde{V}_{h+1} - \underset{\sim}{V}_{h+1}) \\
&\overset{(ii)}{\leq} \left( \sqrt{16\widehat{\mathrm{Var}}(r|x) + 2\widehat{\epsilon}(x)} + \sqrt{36H^2\widehat{\epsilon}(x) + 12\sigma_{\widehat{P}(x)}^2(\widetilde{V}_{h+1})} \right) \phi(n(x)) \\
&\quad + (24H\sqrt{\widehat{S}}V_{h+1}^{\max}(x) + 8.13V_{h+1}^{\max}(x) + 7.49)\phi(n(x))^2 \\
&\quad + \frac{1}{2H}\widehat{P}(x)(\widetilde{V}_{h+1} - \underset{\sim}{V}_{h+1}) + \frac{\bar{\epsilon}(x)}{2} \\
&\leq \left( 4\sqrt{\widehat{\mathrm{Var}}(r|x)} + \sqrt{12}\sigma_{\widehat{P}(x)}(\widetilde{V}_{h+1}) \right) \phi(n(x)) \\
&\quad + (24H\sqrt{\widehat{S}}V_{h+1}^{\max}(x) + 8.13V_{h+1}^{\max}(x) + 7.49)\phi(n(x))^2 \\
&\quad + \frac{1}{2H}\widehat{P}(x)(\widetilde{V}_{h+1} - \underset{\sim}{V}_{h+1}) + \frac{\bar{\epsilon}(x)}{2} + (6H + \sqrt{2})\sqrt{\widehat{\epsilon}(x)}\phi(n(x))
\end{aligned}
\tag{14}
$$

where the inequality $(i)$ is given by Lemma 10 in Dann et al. [12] and, the inequality $(ii)$ follows from equations (10) and (11).

- **Bound on (B).** An application of Lemma 17 in Dann et al. [12] implies that

$$
\begin{aligned}
|(P(x) - & \bar{P}(x))(V_{h+1}^{\pi} - V_{h+1}^{\star})| \\
&\leq (8H + 4.66)\widehat{S}V_{h+1}^{\max}(x)\phi(n(x))^2 + \frac{1}{2H}\bar{P}(x)(V_{h+1}^{\star} - V_{h+1}^{\pi}) \\
&\leq (8H + 4.66)\widehat{S}V_{h+1}^{\max}(x)\phi(n(x))^2 + \frac{1}{2H}\widehat{P}(x)(\widetilde{V}_{h+1} - \underset{\sim}{V}_{h+1}) + \frac{\bar{\epsilon}(x)}{2}
\end{aligned}
$$

where the last inequality uses the assumption that $\widetilde{V}_{h+1} \geq V_{h+1}^{\star} \geq V_{h+1}^{\pi} \geq \underset{\sim}{V}_{h+1}$.

- **Bound on (C).** Note that

$$
|\bar{r}(x) - \widehat{r}(x) + (\bar{P}(x) - \widehat{P}(x))\underset{\sim}{V}_{h+1}| \leq \bar{\epsilon}(x) + (H-1)\bar{\epsilon}(x) = H\bar{\epsilon}(x).
$$

Plugging the above bounds back in (13), we get

$$
\begin{aligned}
Q_h^{\pi}(x) - \underset{\sim}{Q}_h(x) \geq \; & -\frac{1}{H}\widehat{P}(x)(\widetilde{V}_{h+1} - \underset{\sim}{V}_{h+1}) - 4\left( \sqrt{\widehat{\mathrm{Var}}(r|x)} + \sigma_{\widehat{P}(x)}(\widetilde{V}_{h+1}) \right)\phi(n(x)) \\
& - 53\widehat{S}HV_{h+1}^{\max}(x)\phi(n(x))^2 - (H+1)\widehat{\epsilon}(x) - 8H\sqrt{\widehat{\epsilon}(x)}\phi(n(x)) + \psi_h(x)
\end{aligned}
$$

which is non-negative by our choice of $\psi_h(x)$. $\qquad\square$

**Lemma 17** (Upper bounds admissible). *Let $\pi, \widetilde{V}, \underset{\sim}{V}$ be the policy and the value function bounds returned by* OptimistPlan *with inputs $n, \widehat{r}, \widehat{r^2}, \widehat{P}, \widehat{\epsilon}$ after any number of episodes $k$. Consider $h \in [H]$ and $x \in \mathcal{X}$ and assume that $\widetilde{V}_{h+1} \geq V_{h+1}^{\star} \geq V_{h+1}^{\pi} \geq \underset{\sim}{V}_{h+1}$ and that the confidence bound width is at least*

$$
\psi_h(x) \geq 4\left( \widehat{\mathrm{Var}}(r|x) + 2H\sqrt{\widehat{\epsilon}(x)} + \sigma_{\widehat{P}(x)}(\widetilde{V}_{h+1}) \right)\phi(n(x)) + 40\sqrt{\widehat{S}}HV_{h+1}^{\max}(x)\phi(n(x))^2
$$

$$+ \frac{1}{2H}\widehat{P}(x)(\widetilde{V}_{h+1} - \widetilde{V}_{h+1}) + (H + 1/2)\widehat{\epsilon}(x).$$

*Then, in event $E$ (defined in Lemma 11), the upper confidence bound at time $h$ is admissible, i.e.,*

$$Q_h^\star(x) \le \widetilde{Q}_h(x).$$

*Proof.* When $\widetilde{Q}_h(x) = Q_h^{\max}(x)$, the statement holds trivially. Otherwise, we can decompose the difference of the upper bound and the optimal Q-function as

$$\widetilde{Q}_h(x) - Q_h^\star(x) \ge \underbrace{\bar{r}(x) - r(x) + (\bar{P}(x) - P(x))V_{h+1}^\star}_{(A)} + \widehat{P}(x)(\widetilde{V}_{h+1} - V_{h+1}^\star)$$

$$+ \underbrace{\widehat{r}(x) - \bar{r}(x) + (\widehat{P}(x) - \bar{P}(x))V_{h+1}^\star}_{(C)} + \psi_h(x).$$

Note that by assumption $\widehat{P}(x)(\widetilde{V}_{h+1} - V_{h+1}^\star) \ge 0$. The term, (A) is bound using Equation (14) in Lemma 15 and the bias terms (C) is bound as

$$|\bar{r}(x) - \widehat{r}(x) + (\bar{P}(x) - \widehat{P}(x))V_{h+1}^\star| \le \bar{\epsilon}(x) + (H - 1)\bar{\epsilon}(x) = H\bar{\epsilon}(x).$$

Thus,

$$\widetilde{Q}_h(x) - Q_h^\star(x) \ge -4\left(\sqrt{\widehat{\mathrm{Var}}(r|x)} + 2H\sqrt{\widehat{\epsilon}(x)} + \sigma_{\widehat{P}(x)}(\widetilde{V}_{h+1})\right)\phi(n(x)) - 40\sqrt{\widehat{S}}HV_{h+1}^{\max}(x)\phi(n(x))^2$$

$$- \frac{1}{2H}\widehat{P}(x)(\widetilde{V}_{h+1} - \widetilde{V}_{h+1}) - \frac{\bar{\epsilon}(x)}{2} - H\bar{\epsilon}(x) + \psi_h(x) = \psi_h(x) - \widetilde{\psi}_h(x),$$

which is non-negative by our choice for $\psi_h$. $\qquad\square$

### D.4 Tightness of Optimistic Planning

**Lemma 18** (Tightness of Optimistic Planning). *Let $\pi, \widetilde{V}$ and $\widetilde{V}$ be the output of* OptimistPlan *with inputs $n, \widehat{r}, \widehat{r^2}, \widehat{P}$ and $\widehat{\epsilon}$ after any number of episodes $k$. In event $E$ (defined in Lemma 11), we have for all $s \in \mathcal{S}, h \in [H]$,*

$$\widetilde{V}_h(s) - \widetilde{V}_h(s) \le \sum_{x \in \mathcal{X}}\sum_{t=h}^{H}\left(1 + \frac{3}{H}\right)^{2t} w_t(x)\left[Q_t^{\max}(x) \wedge (\gamma_t(x)\phi(n(x)) + \beta_t(x)\phi(n(x))^2 + \alpha\widehat{\epsilon}(x))\right]$$

*where $\gamma_t(x) = 8\left(\sqrt{2\overline{\mathrm{Var}}(r|x)} + 7\sqrt{\widehat{\epsilon}(x)}H + 2\sigma_{P(x)}(V_{t+1}^\pi)\right)$, $\beta_t(x) = 416\widehat{S}HV_{t+1}^{\max}(x)$, $\alpha = 3H + 4$, and the weights $w_t(x) = \mathbb{P}((s_t, a_t) = x \mid s_h = s, a_{h:H} \sim \pi)$ are the probability of visiting each state-action pair at time $t$ under policy $\pi$.*

*Proof.* We start by considering the difference of Q-estimates for $h$ at a state-action pair $x \in \mathcal{X}$

$$\widetilde{Q}_h(x) - \widetilde{Q}_h(x) \le 2\psi_h(x) + \widehat{P}(x)(\widetilde{V}_{h+1} - \widetilde{V}_{h+1})$$

$$= \left(1 + \frac{2}{H}\right)\widehat{P}(x)(\widetilde{V}_{h+1} - \widetilde{V}_{h+1}) + 106\widehat{S}HV_{h+1}^{\max}(x)\phi(n(x))^2 + (2H + 2)\widehat{\epsilon}(x)$$

$$+ 8\left(\sqrt{\widehat{\mathrm{Var}}(r|x)} + 2\sqrt{\widehat{\epsilon}(x)}H + \sigma_{\widehat{P}(x)}(\widetilde{V}_{h+1})\right)\phi(n(x))$$

$$\le \left(1 + \frac{2}{H}\right)\bar{P}(x)(\widetilde{V}_{h+1} - \widetilde{V}_{h+1}) + 106\widehat{S}HV_{h+1}^{\max}(x)\phi(n(x))^2 + (3H + 4)\widehat{\epsilon}(x)$$

$$+ 8\left(\sqrt{2\overline{\mathrm{Var}}(r|x)} + 7\sqrt{\widehat{\epsilon}(x)}H + \sigma_{\bar{P}(x)}(\widetilde{V}_{h+1})\right)\phi(n(x)),$$

where, the equality is given by the definition of $\psi_h$ and the inequality follows by using Equations (10) and (12) to remove the biases. Next, using Lemma 11 from Dann et al. [12] to convert the value variance to the variance with respect to the value function of $\pi$, we get,

$$\widetilde{Q}_h(x) - \underset{\sim}{Q}_h(x) \leq \left(1 + \frac{3}{H}\right) \bar{P}(x)(\widetilde{V}_{h+1} - \underset{\sim}{V}_{h+1}) + 410\widehat{S}HV_{h+1}^{\max}(x)\phi(n(x))^2 + (3H+4)\widehat{\epsilon}(x)$$

$$+ 8\left(\sqrt{2\overline{\mathrm{Var}}(r|x)} + 7\sqrt{\widehat{\epsilon}(x)}H + 2\sigma_{P(x)}(V_{h+1}^\pi)\right)\phi(n(x))$$

$$\leq \left(1 + \frac{3}{H}\right)^2 P(x)(\widetilde{V}_{h+1} - \underset{\sim}{V}_{h+1}) + 416\widehat{S}HV_{h+1}^{\max}(x)\phi(n(x))^2 + (3H+4)\widehat{\epsilon}(x)$$

$$+ 8\left(\sqrt{2\overline{\mathrm{Var}}(r|x)} + 7\sqrt{\widehat{\epsilon}(x)}H + 2\sigma_{P(x)}(V_{h+1}^\pi)\right)\phi(n(x)), \qquad (15)$$

where the second inequality follows by using Lemma 17 from Dann et al. [12] to substiute $\bar{P}(x)(\widetilde{V}_{h+1} - \underset{\sim}{V}_{h+1})$ by $P(x)(\widetilde{V}_{h+1} - \underset{\sim}{V}_{h+1})$. Next, recalling that

$$\widetilde{V}_h(s) - \underset{\sim}{V}_h(s) = \widetilde{Q}_h(s, \pi(s,h)) - \underset{\sim}{Q}_h(s, \pi(s,h)),$$

and rolling the recursion in equation (15) from $s$ to $h$, we get,

$$\widetilde{V}_h(s) - \underset{\sim}{V}_h(s) \leq \sum_{x\in\mathcal{X}}\sum_{t=h}^{H}\left(1+\frac{3}{H}\right)^{2t} w_t(x)[Q_t^{\max}(x) \wedge (\gamma_t(x)\phi(n(x)) + \beta_t(x)\phi(n(x))^2 + \alpha\widehat{\epsilon}(x)],$$

where, $\gamma_t(x) = 8\left(\sqrt{2\overline{\mathrm{Var}}(r|x)} + 7\sqrt{\widehat{\epsilon}(x)}H + 2\sigma_{P(x)}(V_{t+1}^\pi)\right)$, $\beta_t(x) = 416\widehat{S}HV_{t+1}^{\max}(x)$ and $\alpha = 3H + 4$. The final statement follows by observing that $(1+3/H)^{2t} \leq \exp(6)$. $\qquad\square$

### D.5  Proof of the Theorem 8

In this section, we provide the proof of the desired IPOC bound for Algorithm 6.

*Proof.* Throughout the proof, we consider only outcomes in event $E$ (defined in Lemma 11) which occurs with probability at least $1 - \delta$. Lemma 15 implies that the outputs $\pi_k, \widetilde{V}_{k,h}$ and $\underset{\sim}{V}_{k,h}$ from calls to OptimistPlan during the execution of Algorithm 6 satisfy

$$\underset{\sim}{V}_{k,h} \leq V_h^{\pi_k} \leq V_h^\star \leq \widetilde{V}_{k,h}$$

and hence, all the certificates provided by Algorithm 6 are admissible confidence bounds. Further, Lemma 18 shows that the difference between the two value functions returned by OptimistPlan is bounded as

$$\widetilde{V}_{k,1}(s_{k,1}) - \underset{\sim}{V}_{k,1}(s_{k,1}) \leq \exp(6)\sum_{x\in\mathcal{X}}\sum_{h=1}^{H} w_{k,h}(x)\left[Q_h^{\max}(x) \wedge \left(\beta_h(x)\phi(n_k(x))^2 \right.\right.$$

$$\left.\left. + \gamma_{k,h}(x)\phi(n_k(x)) + \alpha\widehat{\epsilon}_k(x)\right)\right], \qquad (16)$$

where, $w_{k,h}(x) = \mathbb{P}((s_{k,h}, a_{k,h}) = x \mid \pi_k, s_{k,1})$ denotes the probability of the agent visiting $x$ in episode $k$ at time $h$ given the policy $\pi_k$ and the initial state $s_{k,1}$, and $\alpha = 3H + 4$, $\beta_h(x) = 416\widehat{S}HV_{h+1}^{\max}(x)$ and $\gamma_{k,h}(x) = 8\left(\sqrt{2\overline{\mathrm{Var}}_k(r|x)} + 7\sqrt{\widehat{\epsilon}(x)}H + 2\sigma_{P(x)}(V_{h+1}^{\pi_k})\right)$.

We define some additional notation, which will come in handy to control Equation (16) above. Let $w_k(x) := \sum_{h=1}^{H} w_{k,h}(x)$ denote the (total) expected visits of $x$ in the $k^{\text{th}}$ episode. Next, for some $w_{\min} > 0$, to be fixed later, define the following subsets of the state action pairs:

(i) $L_k$: Set of all state-actions pairs $x$ that have low expected visitation in the $k^{\text{th}}$ episode, i.e.

$$L_k := \{x \in \mathcal{X} : w_k(x) < w_{\min}\}.$$

(ii) $U_k$: Set of all state-action pairs that had low observation probability in the past, and therefore have not been observed often enough, i.e.

$$U_k := \left\{ x \in \mathcal{X} \setminus L_k : \sum_{i<k} \sum_{\bar{x} \in \mathcal{X}} q(\bar{x}, x) w_i(\bar{x}) < 4H \ln \frac{1}{\delta'} \right\}.$$

(iii) $W_k$: Set of the remaining state-action pairs that have sufficient past probability, i.e.

$$W_k := \left\{ x \in \mathcal{X} \setminus L_k : \sum_{i<k} \sum_{\bar{x} \in \mathcal{X}} q(\bar{x}, x) w_i(\bar{x}) \geq 4H \ln \frac{1}{\delta'} \right\}.$$

Additionally, let $Q^{\max}$ denote an upper bound on the value-bounds used in the algorithm for all relevant $x$ at all times in the first $T$ episodes, i.e.,

$$Q^{\max} \geq \max_{k \in [T], h \in [H]} \max_{x : w_{k,h}(x)>0} Q_h^{\max}(x) \qquad \text{and,}$$

$$Q^{\max} \geq \max_{k \in [T], h \in [H]} \max_{x : w_{k,h}(x)>0} V_{h+1}^{\max}(x).$$

Next, we bound Equation (16) (above) by controlling the right hand side separately for each of the above classes. For $L_k$ and $U_k$, we will use the upper bound $Q^{\max}$ and for the set $W_k$, we will use the bound $\beta_h(x)\phi(n_k(x))^2 + \gamma_{k,h}(x)\phi(n_k(x))) + \alpha\widehat{\epsilon}_k(x))$. Thus,

$$\sum_{k=1}^{T} \widetilde{V}_{k,1}(s_{k,1}) - \underaccent{\tilde}{V}_{k,1}(s_{k,1}) \leq \exp(6)\left( \underbrace{\sum_{k=1}^{T} \sum_{x \in L_k} w_k(x) Q^{\max}}_{(A)} + \underbrace{\sum_{k=1}^{T} \sum_{x \in U_k} w_k(x) Q^{\max}}_{(B)} \right.$$

$$\left. + \underbrace{\sum_{k=1}^{T} \sum_{x \in W_k} \sum_{h=1}^{H} w_{k,h}(x)(\beta_h(x)\phi(n_k(x))^2 + \gamma_{k,h}(x)\phi(n_k(x)) + \alpha\widehat{\epsilon}_k(x))}_{(C)} \right).$$

$$\tag{17}$$

We bound the terms (A), (B) and (C) separately as follows:

1. **Bound on (A).** Since, for any $x \in L_k$, $w_k(x) < w_{\min}$ (by definition), we have

$$Q^{\max} \sum_{k=1}^{T} \sum_{x \in L_k} w_k(x) \leq Q^{\max} T |\mathcal{X}| w_{\min}.$$

2. **Bound on (B).** By the definition of the set $U_k$,

$$\sum_{k=1}^{T} \sum_{x \in U_k} w_k(x) Q^{\max}$$

$$= Q^{\max} \sum_{k=1}^{T} \sum_{x \in \mathcal{X}} w_k(x) \mathbf{1}\left\{ \sum_{i<k} \sum_{\bar{x} \in \mathcal{X}} q(\bar{x}, x) w_i(\bar{x}) < 4H \ln \frac{1}{\delta'} \right\}. \tag{18}$$

Observe that, for any constant $\nu \in (0, 1]$, to be fixed later,

$$\sum_{i<k} \sum_{\bar{x} \in \mathcal{X}} q(\bar{x}, x) w_i(\bar{x}) \geq \sum_{i<k} \sum_{\bar{x} \in \mathcal{X}} q(\bar{x}, x) w_i(\bar{x}) \mathbf{1}\{q(\bar{x}, x) \geq \nu\}$$

$$\geq \sum_{i<k} \sum_{\bar{x} \in \mathcal{N}^-_{\geq \nu}(x)} w_i(\bar{x})\nu, \tag{19}$$

where, $\mathcal{N}^-_{\geq \nu}(x)$ denotes the of incoming neighbors of $x$ (and $x$ itself) in the truncated feedback graph $G_{\geq \nu}$. Plugging the above in Equation (18), we get,

$$\sum_{k=1}^T \sum_{x \in U_k} w_k(x) Q^{\max} \leq Q^{\max} \sum_{k=1}^T \sum_x w_k(x) \mathbf{1} \left\{ \sum_{i<k} \sum_{\bar{x} \in \mathcal{N}^-_{\geq \nu}(x)} w_i(\bar{x}) < \frac{4H}{\nu} \ln \frac{1}{\delta'} \right\}.$$

Next, using a pigeon hole argument from Lemma 23 in the above expression, we get,

$$\sum_{k=1}^T \sum_{x \in U_k} w_k(x) Q^{\max} \leq 4HQ^{\max} \frac{\mu(G_{\geq \nu})}{\nu} \left( 1 + \ln \frac{1}{\delta'} \right).$$

Since the above holds for any $\nu \in (0, 1]$, taking the the infimum over $\nu$, we get

$$\sum_{k=1}^T \sum_{x \in U_k} w_k(x) Q^{\max} \leq 4HQ^{\max} \bar{\mu} \left( 1 + \ln \frac{1}{\delta'} \right),$$

where, $\bar{\mu} := \inf_\nu \frac{\mu(G_{\geq \nu})}{\nu}$.

3. **Bound on (C).** Setting $\beta = 410 \widehat{S} Q^{\max} H$, we get,

$$(C) \leq \beta \sum_{k=1}^T \sum_{x \in W_k} w_k(x) \phi(n_k(x))^2 + \sum_{k=1}^T \sum_{x \in W_k} \sum_{h=1}^H w_{k,h}(x) \gamma_{k,h}(x) \phi(n_k(x))$$

$$+ \alpha \sum_{k=1}^T \sum_{x \in W_k} w_k(x) \widehat{\epsilon}_k(x)$$

$$\overset{\text{①}}{\lesssim} \beta \sqrt{\ln(HT)} \sum_{k=1}^T \sum_{x \in W_k} w_k(x) \phi(n_k(x))^2$$

$$+ \sum_{k=1}^T \sum_{x \in W_k} \sum_{h=1}^H w_{k,h}(x) \tilde{\gamma}_{k,h}(x) \phi(n_k(x)) + \epsilon_{\max} H^2 T$$

$$\overset{\text{②}}{\lesssim} \beta \sqrt{\ln(HT)} \underbrace{\sum_{k=1}^T \sum_{x \in W_k} w_k(x) \phi(n_k(x))^2}_{\text{(D)}}$$

$$+ \sqrt{\underbrace{\sum_{k=1}^T \sum_{x \in W_k} \sum_{h=1}^H w_{k,h}(x) \tilde{\gamma}_{k,h}(x)^2}_{\text{(E)}}} \sqrt{\underbrace{\sum_{k=1}^T \sum_{x \in W_k} w_k(x) \phi(n_k(x))^2}_{\text{(D)}} + \epsilon_{\max} H^2 T}.$$

$$\tag{20}$$

Where, we use the symbol $\lesssim$ to denote $\leq$ up to multiplicative constants, and the inequality (1) follows by bounded $\widehat{\epsilon}_k(x)$ by the largest occurring bias $\epsilon_{\max}$ and using the definition of event $\mathsf{E}^{\mathrm{Var}}$ from Lemma 13 to replace $\gamma_{k,h}(x)$ by $\tilde{\gamma}_{k,h}(x) = 8\sqrt{2 \operatorname{Var}_k(r|x)} + 56\sqrt{\widehat{\epsilon}(x)} H + 16 \sigma_{P(x)}(V^{\pi_k}_{h+1})$ while paying for an additional term of order $\sqrt{\ln(n^2/\delta')/n} \leq \sqrt{\ln(HT)} \phi(n)$. Since this additional term is multiplied by an additional $\phi(n)$, it only appears in the first term of (20). The inequality (2) is given by the Cauchy-Schwarz inequality.

We bound the terms (D) and (E) separately in the following.

(a) **Bound on (D).** The term (A) essentially has the form $\sum_{k=1}^{T}\sum_{x\in W_k} w_k(x)\frac{\ln\ln n_k(x)}{n_k(x)}$. To make our life easier, we first replace the $\ln\ln n_k(x)$ dependency by a constant. Specifically, we upper-bound $\phi(n_k(x))^2$ by a slightly simpler expression $\frac{J}{n_k(x)}$ where $J = 0.75\ln\frac{5.2\ln(2HT)}{\delta'} \geq 0.52\times 1.4\ln\frac{5.2\ln(e\vee 2n_k(x))}{\delta'} \geq 0.52(1.4\ln\ln(e\vee 2n_k(x))+\ln(5.2/\delta'))$ which replaces the dependency on the number of observations $n_k(x)$ in the log term by the total number of time steps $HT \geq Hk \geq n_k(x)$. This gives

$$(A) \leq J\sum_{k=1}^{T}\mathbf{1}\{x\in W_k\}\frac{w_k(x)}{n_k(x)}. \tag{21}$$

By the definition of $W_k$, we know that for all $x \in W_k$ the following chain of inequalities holds

$$\sum_{i<k}\sum_{\bar{x}\in\mathcal{X}} q(\bar{x},x)w_i(\bar{x}) \geq 4H\ln\frac{1}{\delta'} \geq 8H \geq 8\sum_{\bar{x}\in\mathcal{X}} q(\bar{x},x)w_k(\bar{x}).$$

The second inequality is true because of the definition of $\delta'$ gives $\frac{1}{\delta'} = \frac{|\mathcal{X}|(4\widehat{S}+5H+7)}{\delta}$ which is lower bounded by $13 \geq \exp(2)$ because $\delta \leq 1$ and $|\mathcal{X}| \geq 2$. Leveraging this chain of inequalities in combination with the definition of event $\mathsf{E}^{\mathsf{N}}$, we can obtain a lower bound on $n_k(x)$ for $x \in W_k$ as

$$\begin{aligned}
n_k(x) \geq &\frac{1}{2}\sum_{i<k}\sum_{\bar{x}\in\mathcal{X}} q(\bar{x},x)w_i(\bar{x}) - H\ln\frac{1}{\delta'} \geq \frac{1}{4}\sum_{i<k}\sum_{\bar{x}\in\mathcal{X}} q(\bar{x},x)w_i(\bar{x})\\
\geq &\frac{2}{9}\sum_{i\leq k}\sum_{\bar{x}\in\mathcal{X}} q(\bar{x},x)w_i(\bar{x})\\
\geq &\frac{2\nu}{9}\sum_{i<k}\sum_{\bar{x}\in\mathcal{N}_{\geq\nu}^{-}(x)} w_i(\bar{x})
\end{aligned}$$

where the last inequality follows from (19). Plugging this back into (21) and applying Lemma 22 gives

$$(A) \leq \frac{9J}{2\nu}\sum_{k=1}^{T}\sum_{x\in W_k}\frac{w_k(x)}{\sum_{i<k}\sum_{\bar{x}\in\mathcal{N}_{\geq\nu}^{-}(x)}w_i(\bar{x})} \leq \frac{18eJ}{\nu}\mathrm{mas}(G_{\geq\nu})\ln\left(\frac{eHT}{w_{\min}}\right).$$

Since this holds for any $\nu$, we get

$$(A) \leq 18eJ\bar{\mu}\ln\left(\frac{eHT}{w_{\min}}\right).$$

(b) **Bound on (E).** Using the law of total variance for value functions in MDPs (see Lemma 4 in Dann and Brunskill [18] or see Azar et al. [29], Lattimore and Hutter [39] for the discounted setting), we get,

$$\begin{aligned}
&\sum_{k=1}^{T}\sum_{x\in\mathcal{X}}\sum_{h=1}^{H} w_{k,h}(x)\tilde{\gamma}_{k,h}(x)^2\\
&\lesssim\sum_{k=1}^{T}\sum_{h=1}^{H}\sum_{x\in\mathcal{X}} w_{k,h}(x)(\mathrm{Var}(r|x)+H^2\epsilon(x)+\sigma_{P(x)}^2(V_{h+1}^{\pi_k}))\\
&\leq\sum_{k=1}^{T}\sum_{h=1}^{H}\sum_{x\in\mathcal{X}} w_{k,h}(x)(\mathrm{Var}(r|x)+\sigma_{P(x)}^2(V_{h+1}^{\pi_k}))+\epsilon_{\max}H^3 T\\
&\leq\sum_{k=1}^{T}\left(\sum_{x\in\mathcal{X}} w_k(x)r(x)+\mathrm{Var}\left(\sum_{h=1}^{H} r_h\;\middle|\;a_{1:H}\sim\pi_k, s_{k,1}\right)\right)\\
&\quad+\epsilon_{\max}H^3 T
\end{aligned}$$

$$\leq \sum_{k=1}^{T}(H+1)\mathbb{E}\left(\sum_{h=1}^{H} r_h \;\Big|\; a_{1:H} \sim \pi_k, s_{k,1}\right) + \epsilon_{\max}H^3 T$$

$$\leq (H+1)\sum_{k=1}^{T} V_1^{\pi_k}(s_{k,1}) + TH^3\epsilon_{\max},$$

where, the above inequalities use the fact that for any random variable $X \leq X_{\max}$ a.s., we have $\mathrm{Var}(X) \leq \mathbb{E}[X^2] \leq \mathbb{E}[X]X_{\max}$.

Plugging the above developed bounds for the terms (A), (B) and (C) in (17), we get,

$$\sum_{k=1}^{T}\widetilde{V}_{k,1}(s_{k,1}) - \underset{\sim}{V}_{k,1}(s_{k,1}) \lesssim |\mathcal{X}|Q^{\max}Tw_{\min} + \bar{\mu}Q^{\max}H\left(1 + \ln\frac{1}{\delta'}\right) + \beta\sqrt{\ln(HT)}J\bar{\mu}\ln\left(\frac{eHT}{w_{\min}}\right)$$

$$+ \sqrt{J\left(H\sum_{k=1}^{T} V_1^{\pi_k}(s_{k,1}) + H^3\epsilon_{\max}T\right)\bar{\mu}\ln\left(\frac{eHT}{w_{\min}}\right)} + H^2 T\epsilon_{\max}.$$

Setting $w_{\min} = \frac{1}{Q^{\max}|\mathcal{X}|T}$ gives

$$\sum_{k=1}^{T}\widetilde{V}_{k,1}(s_{k,1}) - \underset{\sim}{V}_{k,1}(s_{k,1}) = O\left(\sqrt{\bar{\mu}H\sum_{k=1}^{T} V_1^{\pi_k}(s_{k,1})\ln\frac{|\mathcal{X}|HT}{\delta}} + \bar{\mu}\widehat{S}Q^{\max}H\ln^3\frac{|\mathcal{X}|HT}{\delta}\right)$$

$$+ O\left(\sqrt{\bar{\mu}H^3 T\epsilon_{\max}}\ln\frac{|\mathcal{X}|HT}{\delta} + H^2 T\epsilon_{\max}\right).$$

$\square$

### D.6 Sample Complexity Bound for Algorithm 1 and Algorithm 6

For convenience, we here restate the sample-complexity bound of Algorithm 1 from Section C.3.

**Corollary 1** (PAC-style Bound)**.** *For any episodic MDP with state-actions $\mathcal{X}$, horizon $H$ and feedback graph $G$, with probability at least $1 - \delta$ for all $\epsilon > 0$ jointly, Algorithm 1 can output a certificate with $\widetilde{V}_{k',1}(s_{k',1}) - \underset{\sim}{V}_{k',1}(s_{k',1})$ for some episode $k'$ within the first*

$$k' = O\left(\frac{MH^2}{\epsilon^2}\ln^2\frac{H|\mathcal{X}|}{\epsilon\delta} + \frac{M\widehat{S}H^2}{\epsilon}\ln^3\frac{H|\mathcal{X}|}{\epsilon\delta}\right)$$

*episodes. If the initial state is fixed, such a certificate identifies an $\epsilon$-optimal policy.*

*Proof.* This Corollary is a special case of Proposition 19 below. We simply set $\gamma = 1$ and the quantities $\bar{V}(\bar{T}) = H$ and $Q^{\max} = H$ to their worst-case values. Note also that $\mu = \bar{\mu}$ in deterministic feedback graphs. Then $\bar{T}$ in Proposition 19 evaluates to

$$\bar{T} = O\left(\frac{\mu H^2}{\epsilon^2}\ln^2\frac{|\mathcal{X}|H}{\epsilon\delta} + \frac{\mu\widehat{S}H^2}{\epsilon}\ln^3\frac{|\mathcal{X}|H}{\epsilon\delta}\right)$$

which is the desired sample-complexity. $\square$

**Proposition 19** (Sample-Complexity of Algorithm 6)**.** *Consider any tabular episodic MDP with state-action pairs $\mathcal{X}$, episode length $H$ and stochastic independent directed feedback graph $G$ that provides unbiased observations ($\epsilon_{\max} = 0$). Then, with probability at least $1 - \delta$, for all $\epsilon > 0$ and $\gamma \in \mathbb{N}$ jointly, Algorithm 6 outputs $\gamma$ certificates that are smaller than $\epsilon$ after at most*

$$\bar{T} = O\left(\frac{\bar{\mu}\bar{V}(\bar{T})H}{\epsilon^2}\ln^2\frac{|\mathcal{X}|H}{\epsilon\delta} + \frac{\bar{\mu}\widehat{S}HQ^{\max}}{\epsilon}\ln^3\frac{|\mathcal{X}|H}{\epsilon\delta} + \gamma\right)$$

*episodes where $\bar{V}(T) \geq \frac{1}{T}\sum_{k=1}^{T} V_1^{\pi_k}(s_{k,1}) \leq \frac{1}{T}\sum_{k=1}^{T} V_1^{\star}(s_{k,1}) \leq H$ is a bound on the average expected return achieved by the algorithm during those episodes and can be set to $H$.*

*Proof.* Let $\epsilon_k = \widetilde{V}_{k,1}(s_{k,1}) - V_{k,1}(s_{k,1})$ be the size of the certificate output by Algorithm 6 in episode $k$. By Theorem 8, the cumulative size after $T$ episodes is with high probability $1 - \delta$ bounded by

$$\sum_{k=1}^{T} \epsilon_k \leq O\left( \sqrt{\bar{\mu} H \bar{V}(T) T} \ln \frac{|\mathcal{X}| H T}{\delta} + \bar{\mu} \widehat{S} Q^{\max} H \ln^3 \frac{|\mathcal{X}| H T}{\delta} \right).$$

Here, $\bar{V}(T) \geq \frac{1}{T} \sum_{k=1}^{T} V_1^{\pi_k}(s_{k,1})$ is any non-increasing bound that holds in the high-probability event on the average initial values of all policies played. We can always set $\bar{V}(T) = H = O(1)$ but there may be smaller values appropriate if we have further knowledge of the MDP (such as the value of the optimal policy).

If the algorithm has not returned $\gamma$ certificates of size at most $\epsilon$ yet, then $\sum_{k=1}^{T} \epsilon_k > (T - \gamma)\epsilon$. Combining this with the upper bound above gives

$$\epsilon < \frac{\sqrt{T}}{T - \gamma} \sqrt{c \bar{\mu} H \bar{V}(T)} \ln \frac{|\mathcal{X}| H T}{\delta} + \frac{c \bar{\mu} \widehat{S} Q^{\max} H}{T - \gamma} \ln^3 \frac{|\mathcal{X}| H T}{\delta}$$

for some absolute constant $c$. Since the expression on the RHS is monotonically decreasing, it is sufficient to find a $\bar{T}$ such that

$$\frac{\sqrt{\bar{T}}}{\bar{T} - \gamma} \sqrt{c \bar{\mu} H \bar{V}(\bar{T})} \ln \frac{|\mathcal{X}| H \bar{T}}{\delta} \leq \frac{\epsilon}{2} \quad \text{and} \quad \frac{c \bar{\mu} \widehat{S} Q^{\max} H}{\bar{T} - \gamma} \ln^3 \frac{|\mathcal{X}| H \bar{T}}{\delta} \leq \frac{\epsilon}{2}.$$

to guarantee that the algorithm has returned $\gamma$ certificates of size at most $\gamma$ after $\bar{T}$ episodes. Consider the first condition for $\bar{T}$ that satisfies

$$2\gamma \vee \bar{c} \frac{\bar{\mu} V(\bar{T}) H}{\epsilon^2} \ln^2 \frac{\bar{c} |\mathcal{X}| H}{\epsilon \delta} \leq \bar{T} \leq \left[ \frac{\bar{c} |\mathcal{X}| H}{\epsilon \delta} \right]^5 \tag{22}$$

for some constant $\bar{c}$ large enough ($\bar{c} \geq 3456c$ suffices). A slightly tedious computation gives

$$\frac{\sqrt{\bar{T}}}{\bar{T} - \gamma} \sqrt{c \bar{\mu} H \bar{V}(\bar{T})} \ln \frac{|\mathcal{X}| H \bar{T}}{\delta} \leq 2 \sqrt{\frac{c \bar{\mu} H \bar{V}(\bar{T})}{\bar{T}} \ln^2 \frac{|\mathcal{X}| H \bar{T}}{\delta}}$$

$$\leq \sqrt{\frac{\epsilon^2}{4 \cdot 6^2} \frac{\ln^2 \frac{|\mathcal{X}| H \bar{T}}{\delta}}{\ln^2 \frac{\bar{c} |\mathcal{X}| H}{\epsilon \delta}}} = \frac{\epsilon}{2} \cdot \frac{\ln \frac{|\mathcal{X}| H}{\delta} + \ln \bar{T}}{\ln \frac{|\mathcal{X}| H}{\delta} + \ln \frac{\bar{c}^6 |\mathcal{X}|^5 H^5}{\epsilon^6 \delta^5}}$$

and by the upper-bound condition in (22), the RHS cannot exceed $\frac{\epsilon}{2}$. Consider now the second condition for $\bar{T}$ that satisfies

$$2\gamma \vee \bar{c} \frac{\bar{\mu} \widehat{S} H O^{\max}}{\epsilon} \ln^3 \frac{\bar{c} |\mathcal{X}| H}{\epsilon \delta} \leq \bar{T} \leq \left[ \frac{\bar{c} |\mathcal{X}| H}{\epsilon \delta} \right]^5 \tag{23}$$

which yields

$$\frac{c \bar{\mu} \widehat{S} Q^{\max} H}{\bar{T} - \gamma} \ln^3 \frac{|\mathcal{X}| H \bar{T}}{\delta} \leq \frac{2 c \bar{\mu} \widehat{S} Q^{\max} H}{\bar{T}} \ln^3 \frac{|\mathcal{X}| H \bar{T}}{\delta}$$

$$\leq \frac{\epsilon}{2} \cdot \frac{\ln^3 \frac{|\mathcal{X}| H \bar{T}}{\delta}}{4 \cdot 6^3 \ln^3 \frac{\bar{c} |\mathcal{X}| H}{\epsilon \delta}} = \frac{\epsilon}{2} \cdot \left[ \frac{\ln \frac{|\mathcal{X}| H}{\delta} + \ln \bar{T}}{\ln \frac{|\mathcal{X}| H}{\delta} + \ln \frac{\bar{c}^6 |\mathcal{X}|^5 H^5}{\epsilon^6 \delta^5}} \right]^3.$$

Hence, we have shown that if $\bar{T}$ satisfies the conditions in (22) and (23), then the algorithm must have produced at least $\gamma$ certificates of size at most $\epsilon$ within $\bar{T}$ episodes. By realizing that we can pick

$$\bar{T} = 2\gamma + \bar{c} \frac{\bar{\mu} V(\bar{T}) H}{\epsilon^2} \ln^2 \frac{\bar{c} |\mathcal{X}| H}{\epsilon \delta} + \bar{c} \frac{\bar{\mu} \widehat{S} H O^{\max}}{\epsilon} \ln^3 \frac{\bar{c} |\mathcal{X}| H}{\epsilon \delta} \leq \left[ \frac{\bar{c} |\mathcal{X}| H}{\epsilon \delta} \right]^5,$$

as long as $\gamma$ is not significantly larger than the following quantities, the statement to show follows. $\square$

# E   Technical Lemmas on Sequences on Vertices of a Graph

In this section, we present several technical results that form the foundation for our performance bounds in terms of feedback graph properties. We begin with bounds on self-normalizing sequences on vertices. Lemma 20 provides a bound for vertex-values sequences, which we then generalize to integer-valued vector sequences in Lemma 21 and to real-values vector sequences in Lemma 22. Finally, Lemma 23 gives a bound on a cumulative thresholded process defined over vertices. These results may be of interest beyond the analysis of our specific algorithms and are therefore provided separately.

**Lemma 20** (Bound on self-normalizing vertex sequences). *Let $G = (\mathcal{X}, \mathcal{E})$ be a directed graph and $x \in \mathcal{X}^T$ be a vector of length $T$ taking values in $\mathcal{X}$. Then*

$$\sum_{k=1}^{T} \frac{1}{\sum_{i \in [k]} \sum_{x' \in \mathcal{N}_G(x_k)} \mathbf{1}\{x_i = x'\}} \leq \mu(G) \ln(eT), \tag{24}$$

*where $\mathcal{N}_G(x) = \{x\} \cup \{x' \in \mathcal{X} : (x', x) \in \mathcal{E}\}$ are all incoming neighbors of $x$ and $x$ itself.*

*Proof.* The proof of this lemma essentially follows the layering argument by Lykouris et al. [24]. It works by re-ordering the sum over $T$ in groups based on the graph structure. Consider any mapping $\ell$ of indices to groups that satisfies $\ell(k) = \min\{l \in [T] \colon \forall i < k, \ell(i) = l \Rightarrow x_i \notin \mathcal{N}_G(x_k)\}$ which can be constructed inductively. It assigns each index to the smallest group that does not already contain an earlier incoming neighbor. This assignment has two convenient properties:

- There can be at most $\mu(G)$ indices be assigned to a group because otherwise the subgraph of the associated vertices contains a cycle. If there were a cycle then there would be an index in that cycle that is the child of an earlier index. This violates the definition of $\ell$.

- For all occurrences it holds that $\sum_{i \leq k} \mathbf{1}\{x_i \in \mathcal{N}_G(x_k\} \geq \ell(k)$. This is true because in all layers $l < \ell(k)$ there must be at least one earlier index that is a parent. Otherwise $\ell(k)$ would be $l$ instead.

We now leverage both properties to bound the left hand side of Equation (24) as

$$\text{(LHS of 24)} = \sum_{l=1}^{T} \sum_{k=1}^{T} \frac{\mathbf{1}\{\ell(k) = l\}}{\sum_{i=1}^{k} \mathbf{1}\{x_i \in \mathcal{N}_G(x_k)\}} \leq \sum_{l=1}^{T} \sum_{k=1}^{T} \frac{\mathbf{1}\{\ell(k) = l\}}{l} \leq \sum_{l=1}^{T} \frac{\mu(G)}{l} \leq \mu(G) \ln(eT),$$

where the last inequality comes from a bound on the harmonic number $\sum_{i=1}^{T} 1/i \leq \ln(T) + 1 = \ln(eT)$. This grouping argument bears resemblance with the argument by Lykouris et al. [24]. □

**Lemma 21** (Bound on self-normalizing integer-valued sequences). *Let $G = (\mathcal{X}, \mathcal{E})$ be a directed graph defined on a finite vertex set $\mathcal{X}$ with a maximum acyclic subgraph of size $\mu(G)$ and let $(w_k)_{k \in [T]}$ be a sequence of bounded integer weight functions $w_k \colon \mathcal{X} \to \{0\} \cup [W]$. The following quantity is bounded from above as*

$$\sum_{k=1}^{T} \sum_{x \in \mathcal{X}} \frac{w_k(x)}{\sum_{i=1}^{k} \sum_{x' \in \mathcal{N}_G(x)} w_i(x')} \leq \mu(G) \ln \left( e \sum_{x} \sum_{k=1}^{T} w_k(x) \right)$$

*where $\mathcal{N}_G(x) = \{x\} \cup \{y \in \mathcal{X} \ : \ (y, x) \in \mathcal{E}\}$ is the set of all neighbors pointing to $x$ (and $x$ itself) in $G$.*

*Proof.* We will first reduce this statement to the case where all weights are binary by extending the length of the sequence by a factor of at most $W$. For each index $k$ and value $m \in [W]$ define the weights $\bar{w}_{W(k-1)+m}(x) = \mathbf{1}\{w_k(x) \geq m\}$. Each original index $k$ corresponds now to a block of $W$ indices of which the first $w_k(x)$ are set to 1. Then we rewrite the quantity of interest in terms of these binary weights as

$$\text{(LHS of 21)} = \sum_{k=1}^{T} \sum_{m=1}^{W} \sum_{x \in \mathcal{X}} \frac{\bar{w}_{(k-1)W+m}(x)}{\sum_{i=1}^{k} \sum_{x' \in \mathcal{N}_G(x)} \sum_{m=1}^{W} \bar{w}_{(i-1)W+m}(x')}$$

$$\leq \sum_{k=1}^{WT} \sum_{x \in \mathcal{X}} \frac{\bar{w}_k(x)}{\sum_{i=1}^{k} \sum_{x' \in \mathcal{N}_G(x)} \bar{w}_i(x')}. \tag{25}$$

The inequality holds because we have only changed the indexing but both sides are identical except that the right-hand side potentially contains up to $W$ fewer terms in the denominator per $x \in \mathcal{X}$.

Let now $\mathcal{O}$ be the set of all occurrences of $\bar{w}_k(x) > 0$ and with slight abuse of notation denote by $k(o)$ and $x(o)$ the index and vertice of the occurrence. Note that the total number of occurrences is bounded $|\mathcal{O}| = \bar{T} := \sum_x \sum_{k=1}^{T} w_k(x) \leq |\mathcal{X}|WT$. Further, consider any total order of this set that satisfies $o \leq o'$ implies $k(o) \leq k(o')$ for any $o, o' \in \mathcal{O}$ (i.e., order respects index order but occurrences at the same index can be put in any order). We then rewrite (25) in terms of occurrences

$$(25) \leq \sum_{o \in \mathcal{O}} \frac{1}{\sum_{o' \leq o} \mathbf{1}\{x(o') \in \mathcal{N}_G(x(o))\}}. \tag{26}$$

The inequality holds because the denominator on the right-hand side includes all occurrences of all incoming neighbors at previous indices (but might not count occurrences of neighbors at the current index). Let $X \in \mathcal{X}^{\bar{T}}$ be the vertex-valued sequence of these ordered occurrences, that is, $X = [x(o_1), \dots, x(o_{\bar{T}})]$ for $o_1 < \cdots < o_{\bar{T}}$ and apply Lemma 20. This gives the desired bound

$$(26) \leq \mu(G) \ln(e\bar{T}) = \mu(G) \ln \left( e \sum_x \sum_{k=1}^{T} w_k(x) \right).$$

$\square$

**Lemma 22** (Bound on self-normalizing real-valued sequences, Restatement of Lemma 9). *Let $G = (\mathcal{X}, \mathcal{E})$ be a directed graph defined on a finite vertex set $\mathcal{X}$ with a maximum acyclic subgraph of size $\mu(G)$ and let $(w_k)_{k \in [T]}$ be a sequence of non-negative weight functions $w_k : \mathcal{X} \to \mathbb{R}^+$ which satisfy for all $k$ that $\sum_{x \in \mathcal{X}} w_k(x) \leq w_{\max}$. For any $w_{\min} > 0$, the following quantity is bounded from above as*

$$\sum_{k=1}^{T} \sum_{x \in \mathcal{X}} \frac{\mathbf{1}\{w_k(x) \geq w_{\min}\} w_k(x)}{\sum_{i=1}^{k} \sum_{x' \in \mathcal{N}_G(x)} w_i(x')} \leq 2\mu(G) \ln \left( \frac{eT w_{\max}}{w_{\min}} \right)$$

*where $\mathcal{N}_G(x) = \{x\} \cup \{y \in \mathcal{X} : (y, x) \in \mathcal{E}\}$ is the set of all neighbors pointing to $x$ (and $x$ itself) in $G$.*

*Proof.* Without loss of generality, we can assume that all weights take values in $\{0\} \cup [w_{\min}, w_{\max}]$ and ignore the indicator in the numerator. This is because

$$\sum_{k=1}^{T} \sum_{x \in \mathcal{X}} \frac{\mathbf{1}\{w_k(x) \geq w_{\min}\} w_k(x)}{\sum_{i=1}^{k} \sum_{x' \in \mathcal{N}_G(x)} w_i(x')} \leq \sum_{k=1}^{T} \sum_{x \in \mathcal{X}} \frac{\mathbf{1}\{w_k(x) \geq w_{\min}\} w_k(x)}{\sum_{i=1}^{k} \sum_{x' \in \mathcal{N}_G(x)} \mathbf{1}\{w_i(x') \geq w_{\min}\} w_i(x')}.$$

We define a new set of integer-values weights $\hat{w}_k(x) = \left\lfloor \frac{w_k(x)}{w_{\min}} \right\rfloor$. These new weights have several convenient properties. First, $\hat{w}_k(x)$ are integers bounded by $\frac{w_{\max}}{w_{\min}}$. Second, their total sum is nicely bounded as $\sum_{k=1}^{T} \sum_{x \in \mathcal{X}} \hat{w}_k(x) \leq \frac{T w_{\max}}{w_{\min}}$. Third, from the assumption that $w_k(x) \in \{0\} \cup [w_{\min}, w_{\max}]$, it follows that $\hat{w}_k(x) \in \{0\} \cup \left[1, \frac{w_{\max}}{w_{\min}}\right]$. This implies that

$$\frac{w_k(x)}{2 w_{\min}} \leq \hat{w}_k(x) \leq \frac{w_k(x)}{w_{\min}}$$

as the flooring has the largest relative effect when $\frac{w_k(x)}{w_{\min}} \nearrow 2$. Rearranging terms, we get $w_{\min} \hat{w}_k(x) \leq w_k(x) \leq 2 w_{\min} \hat{w}_k(x)$. We now use this relationship to exchange the original weights with the discretized weights and only pay a factor of 2. Specifically,

$$\sum_{k=1}^{T} \sum_{x \in \mathcal{X}} \frac{w_k(x)}{\sum_{i=1}^{k} \sum_{x' \in \mathcal{N}_G(x)} w_i(x')} \leq \sum_{k=1}^{T} \sum_{x \in \mathcal{X}} \frac{2 w_{\min} \hat{w}_k(x)}{\sum_{i=1}^{k} \sum_{x' \in \mathcal{N}_G(x)} w_{\min} \hat{w}_i(x')}$$

$$\leq 2\mu(G) \ln \left( \frac{eT w_{\max}}{w_{\min}} \right).$$

The final inequality is an application of Lemma 21. $\square$

**Lemma 23** (Restatement of Lemma 10). *Let $G = (\mathcal{X}, \mathcal{E})$ be a graph with finite vertex set $\mathcal{X}$ and let $w_k$ be a sequence of weights $w_k \colon \mathcal{X} \to \mathbb{R}^+$. For any threshold $C \geq 0$,*

$$\sum_{x \in \mathcal{X}} \sum_{k=1}^{\infty} w_k(x) \mathbf{1}\Big\{ \sum_{i=1}^{k} \sum_{x' \in \mathcal{N}_G(x)} w_i(x') \leq C \Big\} \leq \mu(G) C$$

*where $\mathcal{N}_G(x) = \{x\} \cup \{y \in \mathcal{X} \colon (y, x) \in \mathcal{E}\}$ is the set of $x$ and all in-neighbors in $G$.*

*Proof.* We proceed with an inductive argument that modifies the weight function sequence. To that end, we define $w_k^{(0)} = w_k$ for all $k$ as the first element in this sequence (over sequences of weight functions). We then give the value of interest with respect to $(w_k^{(t)})_{k \in \mathbb{N}}$ an explicit name

$$F^{(t)} = \sum_{x \in \mathcal{X}} \sum_{k=1}^{\infty} w_k^{(t)}(x) \mathbf{1}\Big\{ \sum_{i \leq k} \sum_{x' \in \mathcal{N}_G(x)} w_i^{(t)}(x') \leq C \Big\}.$$

Let $y^{(t)}(x) = \sum_{k=1}^{\infty} \mathbf{1}\Big\{ \sum_{i \leq k} \sum_{x' \in \mathcal{N}_G(x)} w_i^{(t)}(x') \leq C \Big\}$ be the largest index for each $x$ that can have positive weight in the sum. Note that $y^{(t)}(x)$ can be infinity. Let $\hat{y}^{(t)} = \max_{x \in \mathcal{X}} y^{(t)}$ be the largest index and $x^{(t)} \in \operatorname{argmax}_x y^{(t)}(x)$ a vertex that hits the threshold last (if at all). We now effectively remove it and its parents from the graph by setting their weights to 0. Specifically, define

$$w_k^{(t+1)}(x) = w_k^{(t)}(x) \mathbf{1}\{x \notin \mathcal{N}_G(x^{(t)})\} \mathbf{1}\{k \leq \hat{y}^{(t)}\} \qquad \text{for all } k \in \mathbb{N}$$

as the weight function of the next inductive step. First note that all weights after $\hat{y}^{(t)}$ can be set to 0 without affecting $F^{(t)}$ because of how we picked $\hat{y}^{(t)}$. Second, by the condition in the first indicator, $x \notin \mathcal{N}_G(x^{(t)})$ the total sum of zeroed weights before $\hat{y}^{(t)}$ is

$$\sum_{i=1}^{\hat{y}^{(t)}} \sum_{x' \in \mathcal{N}_G(x^{(t)})} w_i(x')$$

which can be at most $C$ because $\hat{y}^{(t)}$ was picked as exactly the index where this bound holds. Hence, $F^{(t+1)}$ can decrease at most by $C + w_{\max}$, i.e., $F^{(t+1)} \geq F^{(t)} - C$. We now claim that all weights are 0 after at most $\mu(G)$ steps. This is true because in each step we zero out the weights of at least one vertex that must have at least one positive weight as well as all its parents. We can do this at most the size of the largest acyclic subgraph. Hence $F^{(\mu(G))} = 0$ and therefore

$$F^{(0)} \leq F^{(1)} + C \leq \cdots \leq F^{(\mu(G))} + \sum_{t=1}^{\mu(G)} C = \mu(G) C$$

which completes the proof. $\qquad \square$

**Corollary 2.** *Let $G = (\mathcal{X}, \mathcal{E})$ be a graph defined on a finite vertex set $\mathcal{X}$ and let $w_k$ be a sequence of non-negative bounded weight functions $w_k \colon \mathcal{X} \to [0, w_{\max}]$. For any threshold $C \geq 0$, the following bound holds*

$$\sum_{x \in \mathcal{X}} \sum_{k=1}^{\infty} w_k(x) \mathbf{1}\Big\{ \sum_{i < k} \sum_{x' \in \mathcal{N}_G(x)} w_i(x') \leq C \Big\} \leq \mu(G)(C + w_{\max})$$

*where $\mathcal{N}_G(x) = \{x\} \cup \{y \in \mathcal{X} \colon (y, x) \in \mathcal{E}\}$ is the set of $x$ and all its parents in $G$*

*Proof.* We match the index ranges in front of and within the indicator by increasing the threshold $C$ by the maximum value $w_{\max}$ that the weight can take when the indicator condition is met for the last time

$$(\text{LHS of } 2) \leq \sum_{x \in \mathcal{X}} \sum_{k=1}^{\infty} w_k(x) \mathbf{1}\Big\{ \sum_{i \leq k} \sum_{x' \in \mathcal{N}_G(x)} w_i(x') \leq C + w_{\max} \Big\}.$$

We can now apply Lemma 23. $\qquad \square$

# F    Details and Analysis of Domination Set Algorithm

## F.1    Extension to Unknown Dominating Sets

Since we pay only a logarithmic price for the number of tasks attempted to be learned in the first phase, we can modify the algorithm to attempt to learn policies to reach all $S$ states (and thus all $\mathcal{X}$) and stop as soon as a small dominating set is found.

Assume that we know the domination number (or an upper-bound to it) $\gamma$ but do not know the identity of a dominating set of this size. In this case, we modify Algorithm 3 to

- initialize the active set to $\mathcal{I} = \mathcal{X}$ to all state-action pairs;
- maintain a partial model of the feedback graph $\widetilde{G}$ by initializing it to the empty graph $(\mathcal{X}, \varnothing)$ over state-action pairs and, at every time step, adding all side observations as edges if not already present;
- check after every elimination of a vertex from the active set whether $\widetilde{G}$ has a dominating set of size at most $\gamma$ that only consists of vertices $\mathcal{X} \setminus \mathcal{I}$. If that is the case, we $\mathcal{X}_D$ to this set and move on to the second phase of the algorithm.

We can analyze this version of the algorithm in the same fashion as the case where $\mathcal{X}_D$ is known in advance. As long as there is a dominating set of size $\gamma$ where each vertex of this set is reachable by some policy at least $p_0$ times per episode in expectation, the algorithm has moves on to the second phase after at most $\widetilde{O}\left(\frac{\mu \widetilde{S} H^2}{p_0}\right)$ episodes. The total sample-complexity price for not knowing the dominating set at most a logarithmic $\log |\mathcal{X}|$ factor.

Further, if no good bound on the dominating number $\gamma$ is known, one can still test if the set of inactive vertices $\mathcal{X} \setminus \mathcal{I}$ contains a dominating set and move on to the second phase if that is the case. However, the algorithm may then settle for a larger but easier reachable dominating set (which would be found earlier) compared to the smallest dominating set that may be harder to reach.

## F.2    Proof of Sample-Complexity Bound with Domination Number

In this section, we will prove the main sample-complexity bound for Algorithm 3 in Theorem 3. We will do this in two steps:

1. We show an intermediate, looser bound with an additional additive $\frac{\mu H^2}{p_0^2}$ term stated in Theorem 24 in Section F.3.
2. We prove the final bound in Theorem 3 based on the intermediate bound in Section F.4.

## F.3    Proof of Intermediate Sample-Complexity Bound

**Theorem 24** (Sample-Complexity of Algorithm 3, Loose Bound)**.** *For any tabular episodic MDP with state-actions $\mathcal{X}$, horizon $H$, feedback graph with mas-number $\mu$ and given dominating set $\mathcal{X}_D$ with $|\mathcal{X}_D| = \gamma$ and accuracy parameter $\epsilon > 0$, Algorithm 3 returns with probability at least $1 - \delta$ an $\epsilon$-optimal policy after*

$$O\Big(\Big(\frac{\gamma H^3}{p_0 \epsilon^2} + \frac{\gamma \widehat{S} H^3}{p_0 \epsilon} + \frac{\mu \widehat{S} H^2}{p_0} + \frac{\mu H^2}{p_0^2}\Big) \ln^3 \frac{|\mathcal{X}| H}{\epsilon \delta}\Big)$$

*episodes. Here, $p_0 = \min_{i \in [\gamma]} p^{(i)}$ is the expected number of visits to the vertex in the dominating set that is hardest to reach.*

*Proof.* Algorithm 3 can be considered an instance of Algorithm 1 executed on the extended MDP with two differences:

- We choose $\delta/2$ as failure probability parameter in `OptimistPlan`. The remaining $\frac{\delta}{2}$ will be used later.

- We choose the initial states per episode adaptively. This does not impact any of the analysis of Algorithm 1 as it allows potentially adversarially chosen initial states.

- In the second phase, we do not collect samples with the policy proposed by the OptimistPlan routine but with previous policies.

We therefore can consider the same event $E$ as in the analysis of Algorithm 1 which still has probability at least $1 - \frac{\delta}{2}$ by Lemmas 11. In this event, by Lemma 15 it holds that $\underline{V}_h \leq V_h^\pi \leq V_h^\star \leq \widetilde{V}_h$ for $\underline{V}_h, \widetilde{V}_h, \pi$ returned by all executions of OptimistPlan. As a result, the correctness of the algorithm follows immediately as $\widehat{\pi}$ is guaranteed to be $\epsilon$-optimal in the considered event. It remains to bound the number of episodes collected by the algorithm before returning.

While the regret bound of Algorithm 1 in Theorem 1 does not apply to the second phase, it still holds in the first phase. We can therefore use it directly to bound the number of episodes collected in the first phase.

**Length of first phase:** We first claim that the first phase must end when the algorithm encounters a certificate for the chosen task that has size at most $\frac{p_0}{2}$. This is true from the stopping condition in Line 6. The algorithm removes $i$ from $\mathcal{I}$ as soon as $\widetilde{V}_1((s_1, i)) \leq 2\underline{V}_1((s_1, i))$. This implies that when the stopping condition is met

$$\underline{V}_1((s_1, i)) \geq \frac{\widetilde{V}_1((s_1, i))}{2} \geq \frac{V_1^\star((s_1, i))}{2} = \frac{p^{(i)}}{2}, \tag{27}$$

where the second inequality follows from the fact that $\widetilde{V}_1 \geq V_1^\star$ in event $E$. That means that policy $\pi^{(i)}$ visits node $X_i$ indeed at least $\widehat{p}^{(i)} \geq \frac{p^{(i)}}{2}$ times per episode in expectation.

When the stopping condition is not met, then $\widetilde{V}_1((s_1, i)) > 2\underline{V}_1((s_1, i))$ and hence $\widetilde{V}_1((s_1, i)) - \underline{V}_1((s_1, i)) > \underline{V}_1((s_1, i))$. Note also that $\widetilde{V}_1((s_1, i)) - \underline{V}_1((s_1, i)) \geq V_1^\star((s_1, i)) - \underline{V}_1((s_1, i))$ at all times in event $E$. Combining both lower bounds gives

$$\widetilde{V}_1((s_1, i)) - \underline{V}_1((s_1, i)) \geq (V_1^\star((s_1, i)) - \underline{V}_1((s_1, i))) \vee \underline{V}_1((s_1, i))$$
$$\geq \frac{V_1^\star((s_1, i))}{2} = \frac{p^{(i)}}{2}. \tag{28}$$

Assume the algorithm encounters a certificate that satisfies

$$\widetilde{V}_1((s_1, i)) - \underline{V}_1((s_1, i)) \leq \frac{p_0}{4},$$

where $i$ is the task which is about to be executed. By the task choice of the algorithm, this implies for any $j \in \mathcal{I}$

$$\widetilde{V}_1((s_1, j)) - \underline{V}_1((s_1, j)) \leq \widetilde{V}_1((s_1, i)) - \underline{V}_1((s_1, i)) \leq \frac{p_0}{4} < \frac{p^{(j)}}{2},$$

where the last inequality follows the definition of $p_0$. As a result, by contradiction with (28), all remaining tasks would be removed from $\mathcal{I}$. Hence, the first phase ends when or before the algorithm has produced a certificate for the chosen task of size $\frac{p_0}{4}$. By Proposition 19, this can take at most

$$O\Big(\frac{\mu H^2}{p_0^2} \ln^2 \frac{|\mathcal{X}|H}{p_0 \delta} + \frac{\mu \widehat{S} H^2}{p_0} \ln^3 \frac{|\mathcal{X}|H}{p_0 \delta}\Big)$$

episodes. Note that even though the algorithm operates in the extended MDP, the size of the maximum acyclic subgraph $\mu$ is identical to that of the original feedback graph since all copies of a state-action pair form a clique in the extended feedback graph $\bar{G}$. Further note that even though the number of states $\bar{S}$ in the extended MDP is larger than in the original MDP by a factor of $(\gamma + 1)$, this factor does not appear in the lower-order term as the number of possible successor states (which can have positive transition probability) are still bounded by $\widehat{S}$ in each state-action pair of the extended MDP. It only enters the logarithmic term due to the increased state-action space.

**Length of second phase:** We now determine a minimum number of samples per state-action pair that ensures that the algorithm terminates. By Lemma 18, the difference $\widetilde{V}_1((s_1, 0)) - \underline{V}_1((s_1, 0))$ can be bounded for the case where $\epsilon_{\max} = 0$ by

$$\exp(6) \sum_{x \in \mathcal{X}} \sum_{h=1}^{H} w_{\pi,h}(x)(H \wedge (\beta \phi(n(x))^2 + \gamma_h(x)\phi(n(x))))$$

with $\beta = 416\widehat{S}H^2$ and $\gamma_h(x) = 16\sigma_{P(x)}(V_{h+1}^\pi) + 16$ (where we use $Q^{\max} = H$ and 1 as an upper-bound to $\overline{\text{Var}}(r|x)$). The weights $w_{\pi,h}(x) = \mathbb{E}_\pi \left[ \mathbf{1}\{(s_h, a_h) = x\} \right]$ are the probability of $\pi$ visiting each state-action pair at a certain time step $h$. This can be upper-bounded by

$$\exp(6)\Big(\beta \sum_{x \in \mathcal{X}} w_\pi(x)\phi(n(x))^2 + \sum_{x \in \mathcal{X}} \sum_{h=1}^{H} \gamma_h(x) w_{\pi,h}(x)\phi(n(x))\Big)$$

$$\leq \exp(6)\Big(\beta \sum_{x \in \mathcal{X}} w_\pi(x)\phi(n(x))^2 + \sqrt{\sum_{x \in \mathcal{X}} \sum_{h=1}^{H} \gamma_h^2(x) w_{\pi,h}(x)}\sqrt{\sum_{x \in \mathcal{X}} w_\pi(x)\phi(n(x))^2}\Big), \tag{29}$$

where we used the shorthand notation $w_\pi(x) = \sum_{h=1}^{H} w_{\pi,h}(x)$ and applied Cauchy-Schwarz in the second step. Assume now that we had at least $\bar{n} \in \mathbb{N}$ samples per state-action pair. Then (29) is again upper-bounded by

$$\exp(6)\beta H \phi(\bar{n})^2 + \exp(6)\sqrt{H}\phi(\bar{n})\sqrt{\sum_{x \in \mathcal{X}} \sum_{h=1}^{H} \gamma_h^2(x) w_{\pi,h}(x)}. \tag{30}$$

For the remaining term under the square-root, we use the law of total variance for value functions in MDPs [22, 18] and bound

$$\sum_{x} \sum_{h=1}^{H} w_{\pi,h}(x)\gamma_h(x)^2 \leq 2 \times 16^2 \sum_{x} \sum_{h=1}^{H} w_{\pi,h}(x) + 2 \times 16^2 \sum_{x} \sum_{h=1}^{H} w_{\pi,h}(x)\sigma_{P(x)}^2(V_{h+1}^\pi)$$

$$\leq 2 \times 16^2(H + H^2) \leq 4^5 H^2.$$

Plugging this back into (30) gives

$$416\exp(6)\widehat{S}H^3\phi(\bar{n})^2 + 4^{5/2}\exp(6)H^{3/2}\phi(\bar{n}) \leq \frac{c\widehat{S}H^3 \ln\ln\bar{n}}{\bar{n}}\ln\frac{|\mathcal{X}|H}{\delta} + \sqrt{\frac{cH^3\ln\ln\bar{n}}{\bar{n}}\ln\frac{|\mathcal{X}|H}{\delta}}$$

for some absolute constant $c$ where we bounded $\phi(\bar{n})^2 \lesssim \frac{\ln\ln\bar{n}}{\bar{n}}\ln\frac{|\mathcal{X}|H}{\delta}$. Then there is an absolute constant $\bar{c}$ so that this expression is smaller than $\epsilon$ for

$$\bar{n} = \frac{\bar{c}H^3}{\epsilon^2}\ln^2\frac{|\mathcal{X}|H}{\epsilon\delta} + \frac{\bar{c}\widehat{S}H^3}{\epsilon}\ln^2\frac{|\mathcal{X}|H}{\epsilon\delta}.$$

Hence, the algorithm must stop after collecting $\bar{n}$ samples for each state-action pair. By the property of the dominating set, it is sufficient to collected $\bar{n}$ samples for each element of the dominating set. Analogously to event $\mathsf{E}^\mathsf{N}$ in Lemma 11, we can show that with probability at least $1 - \delta/2$, for all $k$ and $i$, the number of visits to any element of the dominating set $X_i$ are lower-bounded by the total visitation probability so far as

$$v(X_i) \geq \frac{1}{2}\sum_{j \leq k} w_j(X_i) - H\ln\frac{2\gamma}{\delta}, \tag{31}$$

where $k$ is the total number of episodes collected so far and $w_j(X_i)$ is the expected number of visits to $X_i$ of the policy played in the $j$th episode of the algorithm. Further, the stopping condition in the first phase was designed so that $\pi^{(i)}$ visits $X_i$ at least $\widehat{p}^{(i)} \geq \frac{p^{(i)}}{2}$ times per episode in expectation

(see Equation (27)). This follows from the definition of the reward in the extended MDP and the fact that certificates are valid upper and lower confidence bounds on the value function, that is

$$\widehat{p}^{(i)} = \underset{\sim}{V_1}((s_1, i)) \geq \frac{\widetilde{V}_1((s_1, i))}{2} \geq \frac{V_1^\star((s_1, i))}{2} = \frac{p^{(i)}}{2}.$$

Hence, if $\pi^{(i)}$ is executed for $m_i$ episodes in the second phase, the total observation probability for $X_i$ is at least $\frac{m_i p^{(i)}}{2}$. Plugging this back in (31) gives

$$v(X_i) \geq \frac{1}{4} m_i p^{(i)} - H \ln \frac{2\gamma}{\delta}.$$

Hence, to ensure that the algorithm has visited each vertex of the dominating set sufficiently often, i.e., $\min_{i \in [\gamma]} v(X_i) \geq \bar{n}$, it is sufficient to play

$$m_i = O\left( \frac{H^3}{p^{(i)}\epsilon^2} \ln^2 \frac{|\mathcal{X}|H}{\epsilon\delta} + \frac{\widehat{S}H^3}{p^{(i)}\epsilon} \ln^2 \frac{|\mathcal{X}|H}{\epsilon\delta} \right)$$

episodes with each policy $\pi^{(i)}$ in the second phase. Hence, we get a bound on the total number of episodes in the second phase by summing over $\gamma$, which completes the proof. $\qquad\square$

### F.4 Proof of Tighter Sample Complexity Bound Avoiding $1/p_0^2$

The sample complexity proof of Algorithm 3 in Theorem 24 follows with relative ease from the guarantees of Algorithm 1. It does however have a $\tilde{O}\left( \frac{\mu H^2}{p_0^2} \right)$ dependency which is absent in the lower-bound in Theorem 5. We now show how to remove this additive $\tilde{O}\left( \frac{\mu H^2}{p_0^2} \right)$ term and prove the main result for Algorithm 3 which we restate here:

**Theorem 3** (Sample complexity of Algorithm 3). *For any tabular episodic MDP with state-actions $\mathcal{X}$, horizon $H$, feedback graph with mas-number $\mu$ and given dominating set $\mathcal{X}_D$ with $|\mathcal{X}_D| = \gamma$ and accuracy $\epsilon > 0$, Algorithm 3 returns with probability at least $1 - \delta$ an $\epsilon$-optimal policy after*

$$\widetilde{O}\left( \frac{\gamma H^3}{p_0\epsilon^2} + \frac{\gamma \widehat{S}H^3}{p_0\epsilon} + \frac{\mu \widehat{S}H^2}{p_0} \right) \tag{2}$$

*episodes. Here, $p_0 = \min_{i \in [\gamma]} p^{(i)}$ is possible expected number of visits to the node in the dominating set that is hardest to reach.*

Before presenting the formal proof, we sketch the main argument. The proof of the intermediate result in Theorem 24 relies on Corollary 1 for Algorithm 1 to bound the length of the first episode. Yet, Proposition 19 shows that the dominant term of the sample-complexity of Algorithm 1 only scales with $\frac{1}{\epsilon^2}\mu H \frac{1}{T}\sum_{k=1}^T V_1^\star(s_{k,1})$ for some $T$ instead of the looser $\frac{\mu H^2}{\epsilon^2}$ in Corollary 1. We can upper-bound each summand $V_1^\star(s_{k,1})$ by the optimal value of the task of the episode, e.g., $p^{(i)}$ for task $i$. If all vertices in the dominating set are equally easy to reach, that is, $p^{(1)} = p^{(2)} = \ldots = p^{(\gamma)} = p_0$, this yields $V_1^\star(s_{k,1}) = p_0$ and $\epsilon \approx p_0$. In this case, this term in the sample-complexity evaluates to

$$\frac{\mu H p_0}{p_0^2} \approx \frac{\mu H}{p_0},$$

and gets absorbed into the last term $\frac{\mu \widehat{S}H^2}{p_0}$ of the sample-complexity in Theorem 3. However, there is a technical challenge when $p^{(i)}$s vary significantly across tasks $i$, i.e., some vertices in the dominating set can be reached easily while others can only be reached with low probability. A straightforward bound only yields

$$\frac{\mu H \max_{i \in [\gamma]} p^{(i)}}{p_0^2},$$

which can be much larger when $\max_i p^{(i)} \gg \min_i p^{(i)} = p_0$. To avoid this issue, we will apply a careful argument that avoids a linear factor of the number of policies learned $\gamma$ (which a separate

analysis of every task would give us, see Section 4) while at the same time still only having a $1/p_0$ dependency instead of the $1/p_0^2$.

The key is an inductive argument that bounds the number of episodes for the $j$ vertices of the dominating set that are the easiest to reach for any $j \in [\gamma]$. Thus, assume without loss of generality that vertices are ordered with decreasing reachability, i.e., $p^{(1)} \geq p^{(2)} \geq \cdots \geq p^{(\gamma)}$. We will show that the algorithm plays tasks $1, \ldots, j$ in at most

$$O\Big(j + \frac{\mu \widehat{S} H^2}{p^{(j)}} \ln^3 \frac{|\mathcal{X}|H}{\delta p_0}\Big) \tag{32}$$

episodes. For $j = \gamma$, this gives the total length of the first phase and yields the desired reduction in sample complexity for Theorem 3. Assuming that this bound holds for 1 to $j - 1$, we consider the subset of episodes $\mathcal{K}_j$ in which the algorithm plays tasks $[j]$ and show the average optimal value in these episodes is not much larger than $p^{(j)}$

$$\frac{1}{|\mathcal{K}_j|} \sum_{k \in \mathcal{K}_j} V_1^\star(s_{k,1}) \lesssim p^{(j)} \ln \frac{e p^{(1)}}{p^{(j)}}.$$

This insight is the key to prove (32) for $j$.

**Full proof:**

*Proof of Theorem 3.* The proof of Theorem 24 can be directly applied here. It yields that with probability at least $1 - \delta$, Algorithm 3 returns an $\epsilon$-optimal policy and event $E$ from Lemma 11 holds. We further know that the algorithm collects at most $T_1$ and $T_2$ episodes in the first and second phase respectively, where

$$T_1 = O\Big(\Big(\frac{\mu H^2}{p_0^2} + \frac{\mu \widehat{S} H^2}{p_0}\Big) \ln^3 \frac{|\mathcal{X}|H}{\epsilon\delta}\Big), \quad \text{and,} \quad T_2 = O\Big(\frac{\gamma \widehat{S} H^3}{p_0 \epsilon} \ln^2 \frac{|\mathcal{X}|H}{\epsilon\delta}\Big).$$

It is left to provide a tighter bound for the length of the first phase. As mentioned above, assume without loss of generality that the nodes of the dominating set are ordered with decreasing reachability, i.e., $p^{(1)} \geq p^{(2)} \geq \cdots \geq p^{(\gamma)}$. For any $j \in [\gamma]$, let $\mathcal{K}_j \subseteq [T_1]$ be the set of episodes where the algorithm played task $1, \ldots, j$. To reason how large this set can be, we need slightly refined versions of the IPOC bound of Algorithm 6 in Theorem 8 and the corresponding sample-complexity result in Proposition 19. We state them below as Lemmas 25 and 26. They allow us to reason over arbitrary subset of episodes instead of consecutive episodes. Their proof is virtually identical to those of Theorem 8 and Proposition 19.

As we know from the proof of Theorem 24, the algorithm cannot play task $i$ anymore once it has encountered a certificate $\widetilde{V}_1((s_1, i)) - \underline{V}_1((s_1, i)) \leq \frac{p^{(i)}}{4}$. Hence, it can only encounter at most $j$ episodes in $\mathcal{K}_j$ where the certificate was at most $\frac{p^{(j)}}{4}$. Thus by Lemma 26 below

$$|\mathcal{K}_j| \leq O\Big(j + 1 + \mu H \frac{\sum_{k \in \mathcal{K}_j} V_1^{\pi_k}(s_{k,1})}{|\mathcal{K}_j| \cdot (p^{(j)})^2} \ln^2 \frac{|\mathcal{X}|HT_1}{\delta} + \frac{\mu \widehat{S} H^2}{p^{(j)}} \ln^3 \frac{|\mathcal{X}|HT_1}{\delta}\Big). \tag{33}$$

Since $j \leq \gamma$ and we can assume that the provided dominating set is of sufficient quality, i.e., $\gamma \leq \frac{\mu \widehat{S} H^2}{p^{(j)}} \ln^3 \frac{|\mathcal{X}|HT_1}{\delta}$, the $j + 1$ term is dominated by the later terms in this bound. We now claim that

$$|\mathcal{K}_j| = O\Big(\frac{\mu \widehat{S} H^2}{p^{(j)}} \ln^3 \frac{|\mathcal{X}|H}{p_0\delta}\Big) \tag{34}$$

which we will show inductively. Assume that (34) holds for all $1, \ldots j - 1$ and consider the sum of policy values in $\mathcal{K}_j$ from (33)

$$\sum_{k \in \mathcal{K}_j} V_1^{\pi_k}(s_{k,1}) \leq \sum_{k \in \mathcal{K}_j} V_1^\star(s_{k,1}) = \sum_{i=1}^j \sum_{k \in \mathcal{K}_j \setminus \mathcal{K}_{j-1}} p^{(i)} = \sum_{i=1}^j p^{(i)}(|\mathcal{K}_i| - |\mathcal{K}_{i-1}|)$$

where we define $\mathcal{K}_0 = \varnothing$ for convenience. Consider $C = c\mu\widehat{S}H^2 \ln^3 \frac{|\mathcal{X}|H}{p_0\delta}$ with a large enough numerical constant $c$ so that induction assumption implies $|\mathcal{K}_i| \leq C/p^{(i)}$ for $i = 1, \ldots, j-1$. Assume further that $|\mathcal{K}_j| \geq C/p^{(j)}$. Then with $1/p^{(0)} := 0$

$$\frac{1}{|\mathcal{K}_j|} \sum_{k \in \mathcal{K}_j} V_1^{\pi_k}(s_{k,1}) \leq p^{(j)} \sum_{i=1}^{j} p^{(i)}\Big(\frac{1}{p^{(i)}} - \frac{1}{p^{(i-1)}}\Big).$$

Define now $w_i = \frac{1}{p^{(i)}} - \frac{1}{p^{(i-1)}}$, which allows us to write $p^{(i)} = \frac{1}{\sum_{l=1}^{i} w_l}$ because $\sum_{l=1}^{i} w_l = \frac{1}{p^{(i)}} - \frac{1}{p^{(0)}} = \frac{1}{p^{(i)}}$. Writing the expression above in terms of $w_i$ yields

$$\frac{1}{|\mathcal{K}_j|} \sum_{k \in \mathcal{K}_j} V_1^{\pi_k}(s_{k,1}) \leq p^{(j)} \sum_{i=1}^{j} \frac{w_i}{\sum_{l=1}^{i} w_l} \overset{(i)}{=} p^{(j)}\Big(1 + \ln\Big(\sum_{i=1}^{j} w_i\Big) - \ln w_1\Big)$$

$$= p^{(j)}\Big(1 + \ln \frac{1}{p^{(j)}} - \ln \frac{1}{p^{(1)}}\Big)$$

$$= p^{(j)} \ln \frac{ep^{(1)}}{p^{(j)}} \leq p^{(j)} \ln \frac{eH}{p_0},$$

where $(i)$ follows from the fundamental theorem of calculus (see e.g. Lemma E.5 by Dann et al. [38]). We just showed that if $|\mathcal{K}_j| \geq C/p^{(j)}$, the average policy value $\frac{1}{|\mathcal{K}_j|} \sum_{k \in \mathcal{K}_j} V_1^{\pi_k}(s_{k,1})$ cannot be much larger than $1/p^{(j)}$. Plugging this back into (33) gives that

$$|\mathcal{K}_j| = O\Big(\frac{\mu H p^{(j)}}{(p^{(j)})^2} \ln \frac{eH}{p_0} \ln^2 \frac{|\mathcal{X}|HT_1}{\delta} + \frac{\mu\widehat{S}H^2}{p^{(j)}} \ln^3 \frac{|\mathcal{X}|HT_1}{\delta}\Big) = O\Big(\frac{\mu\widehat{S}H^2}{p^{(j)}} \ln^3 \frac{|\mathcal{X}|H}{p_0\delta}\Big),$$

where the equality follows since $\ln(T_1) \lesssim \ln \frac{|\mathcal{X}|H}{p_0\delta}$. We have just shown that (34) also holds for $j$ which completes the inductive argument. Evaluating (34) for $j = \gamma$ shows that the length of the first phase is indeed $O\Big(\frac{\mu\widehat{S}H^2}{p_0} \ln^3 \frac{|\mathcal{X}|H}{p_0\delta}\Big)$ which completes the proof. $\qquad\square$

**Lemma 25.** *For any tabular episodic MDP with episode length $H$, state-action space $\mathcal{X}$ and a directed feedback graph $G$, the total size of certificates of Algorithm 1 on any (possibly random) set of episodes indices $\mathcal{K}$ as is bounded in event $E$ (defined in Lemma 11) as*

$$\sum_{k \in \mathcal{K}} \widetilde{V}_{k,1}(s_{k,1}) - \underset{\sim}{V}_{k,1}(s_{k,1}) = O\Big(\sqrt{\mu H \sum_{k \in \mathcal{K}} V_1^{\pi_k}(s_{k,1}) \ln \frac{|\mathcal{X}|HT}{\delta}} + \mu\widehat{S}H^2 \ln^3 \frac{|\mathcal{X}|HT}{\delta}\Big),$$

*where $T = \max\{k \colon k \in \mathcal{K}\}$ is the largest episode index in $\mathcal{K}$.*

*Proof.* The proof of this lemma is in complete analogy to the proof of Theorem 8, except that we take the sum $\sum_{k \in \mathcal{K}}$ instead of $\sum_{k=1}^{T}$. In the decomposition in Equation (20), we replace in term (D) the sum over $\mathcal{K}$ with $[T]$ and proceed normally (which yields the $\ln T$ terms). But in term (E) we keep the sum over $\mathcal{K}$ which yields the $\sum_{k \in \mathcal{K}} V_1^{\pi_k}(s_{k,1})$ term in the bound above. $\qquad\square$

**Lemma 26.** *Consider any tabular episodic MDP with state-action space $\mathcal{X}$, episode length $H$ and directed feedback graph $G$ with mas-number $\mu$. For any $\epsilon > 0$, $m \in \mathbb{N}$ and (possibly random) subset of episodes $\mathcal{K} \subseteq [T]$ with*

$$|\mathcal{K}| = O\Big(m + \frac{\mu H \frac{1}{|\mathcal{K}|} \sum_{k \in \mathcal{K}} V_1^{\pi_k}(s_{k,1})}{\epsilon^2} \ln^2 \frac{|\mathcal{X}|HT}{\delta} + \frac{\mu\widehat{S}H^2}{\epsilon} \ln^3 \frac{|\mathcal{X}|HT}{\delta}\Big).$$

*Algorithm 6 produces in event $E$ (defined in Lemma 11 at least $m$ certificates with size $\widetilde{V}_{k,1}(s_{k,1}) - \underset{\sim}{V}_{k,1}(s_{k,1}) \leq \epsilon$ with $k \in \mathcal{K}$.*

*Proof.* The proof of this lemma is in complete analogy to the proof of Proposition 19, except that we take the sum $\sum_{k \in \mathcal{K}}$ instead of $\sum_{k=1}^{T}$ when we consider the cumulative certificate size and apply Lemma 25 instead of Theorem 8. $\qquad\square$

## F.5 Comparison to Lower Bound

In general MDPs where we do not have a good idea about how reachable the dominating set is and whether the MDP has sparse transitions, the sample-complexity of Algorithm 3 is

$$\widetilde{O}\Big(\frac{\mu S H^2}{p_0} + \frac{\gamma H^3}{p_0 \epsilon^2} + \frac{\gamma S H^3}{p_0 \epsilon}\Big),$$

while the lower bound is

$$\widetilde{\Omega}\Big(\frac{\alpha H^2}{\epsilon^2} \wedge \Big(\frac{\alpha}{p_0} + \frac{\gamma H^2}{p_0 \epsilon^2}\Big)\Big).$$

When $\epsilon$ is small enough and the dominating set is of good quality, i.e., $\gamma < \alpha$, the second term dominates the first in the lower bound. We see that the $1/p_0$ dependency in our sample-complexity upper bound is tight up to log factors. Nonetheless, there is a gap of $H^2$ and $SH$ between our upper- and lower-bound even when the feedback graph is symmetric (where $\mu = \alpha$). It should be noted that the explicit $S$ dependency in the $1/\epsilon$-term is typical for model-based algorithms and it is still an open problem whether it can be removed without increase in $H$ for model-based algorithms in MDPs with dense transitions.

However, the lower bound in Theorem 5 relies on a class of MDPs that in fact have sparse transitions. If we know that the true MDP belongs to this class, then we can run Algorithm 3 with the planning routine of Algorithm 6 that supports state-action-dependent upper-bounds and set

$$Q_h^{\max}(x) = 1, \quad V_{h+1}^{\max}(x) = 1 \quad \text{for } x \text{ in tasks } \{1, \dots, \gamma\} \text{ and } \widehat{S} = 2,$$

because each dominating node can only be reached once per episode and each state-action pair can only transition to one of two states. With these modifications, one can show that Algorithm 3 terminates within

$$\widetilde{O}\Big(\frac{\mu H}{p_0} + \frac{\gamma H^3}{p_0 \epsilon^2}\Big)$$

episodes matches the lower-bound up to one factor of $H$ and log-terms in symmetric feedback graphs for small enough $\epsilon$.