[Reviews · NeurIPS 2020]

Review 1

Summary and Contributions: The paper presents improved bounds for learning in fully-observable, tabular MDPs with side observations using a feedback graph.

Strengths: The regret and sample-complexity bounds are tighter than previous best in a similar setting. The feedback graph has potential as a new analysis tool both for defining bounds and developing new algorithms.

Weaknesses: Tabular MDPs, accurate side information. I have no objection in principle to analysis that starts with the simplest case but the paper is careless in postulating that it will generalize to a realistic setting. Would have liked to see more intuition developed around the feedback graph. Particularly because I misunderstood the first time around and thought it depended on the side information provided with the currents state being transitions from the state. -=- Misunderstanding kept for posterity -=- It is essentially an expanded MDP diagram, where each vertex is a {s, a} pair and each edge indicates a valid transition. This is an easy construct to think about. For example, Mas-number is then the longest possible history that doesn't repeat itself, independence number is the maximum number of state-action pairs that don't connect directly to each other, and domination number is the minimal set of {s, a} pairs whose neighbours cover the space. It would be vastly easier to follow the implications if these kinds of common-sense ties were explicitly provided in the paper rather than left as an exercise to the reader. Appendix F? Perhaps this work would be better served as a journal submission, if 48 pages are required to present it clearly.

Correctness: Empirical methodology irrelevant as not necessary for the value of this work. Claims are sound and well supported. -=- Kept for posterity -=- Self-loops being implicit and thus ignored seemed incorrect, given the analysis and bounds depend on mas-number. If self-loops exist for all {s, a} pairs then there are no acyclic subgraphs. There is no empirical analysis, although I don't believe that is strictly necessary for this kind of work.

Clarity: I found the paper incredibly difficult to follow with too much reliance on appendices. Update: However, given the authors' response I do think the readability will be improved for the unfortunates like myself who do not have the relevant background information easy to hand. Material in the appendix helps greatly and if some can be moved into the body of the paper that could resolve this issue.

Relation to Prior Work: The paper rigorously compares this approach to related bandit analysis. I would have liked to see this related to Dyna-style approaches that use side information drawn from trajectories, and a tighter exploration of how generated/estimated side information would impact the analysis.

Reproducibility: Yes

Additional Feedback: I was quite intrigued by the core idea and would like to see it explored in either a forum that allows for the level of detail required or split into more easily-digestible pieces.


Review 2

Summary and Contributions: This paper addresses the setting where an RL agent receives additional observations, after executing every action, which provide it with information about possible transitions that it could have experienced. These side observations may be generated, for instance, by auxiliary sensors. The authors formalize this setting by constructing a feedback graph based on additional observations. Feedback graphs may be used by model-based RL algorithms to learn more efficiently. In particular, the authors show that the regret of the resulting model-based algorithm is bounded by certain properties of the graph, instead of depending on the number of states and actions that exist in the original problem (without side observations).

Strengths: This is a well-written theoretical paper. Although it does not demonstrate empirically that feedback graphs work in real-life problems, the underlying theory seems sound and the formal results that are presented are certainly not trivial.

Weaknesses: The main weakness of this paper is that it lacks an empirical evaluation section demonstrating that the theory is actually applicable to interesting problems. Also, the results depend on a few assumptions that may not always be realistic: tabular model-based algorithms, or multitask settings where the rewards of all but one task are known, and where the dynamics are equal across tasks.

Correctness: The theory underlying the results seems correct. The paper has no empirical results.

Clarity: Yes.

Relation to Prior Work: Related work is briefly but appropriately discussed.

Reproducibility: Yes

Additional Feedback: This paper addresses the problem of an RL agent that receives additional observations, after executing every action, which provide it with information about possible transitions that it could have experienced. These side observations might be generated, for instance, by auxiliary sensors. The authors formalize this setting by can defining a feedback graph based on the additional observations. Feedback graphs may be used by model-based RL algorithms to learn more efficiently. In particular, the authors show that the regret of the resulting model-based algorithm is bounded by certain properties of the graph, instead of depending on the number of states and actions that exist in the original problem (without side observations). This is a well-written theoretical paper. Although it does not demonstrate empirically that feedback graphs work in real-life problems, the underlying theory seems sound and the formal results that are presented are not trivial. I have just a few questions and comments: 1) Is it possible to formally characterize the relation between the results that you present and possible bounds on the performance achievable by off-policy learning algorithms? There might exist a relation, given that at each step, these algorithms may also run updates over (past) experiences other than the current experience. 2) Could your analyses be used to formally characterize the regret of algorithms that operate over a replay buffer? I believe that it may be possible to interpret many modern algorithms that use a buffer of past experiences (and that run updates over them) as algorithms that, besides that current experience, are also given, at each step, a set of additional observations. I wonder if your results could be extended to show regret bounds on off-policy algorithms of this type. 3) Intuitively, what are the main challenges involved in adapting your findings to model-free algorithms? 4) Could you please better motivate the key assumption made when analyzing extended MDPs? In particular, in what type of multitask settings would it be realistic to assume that the rewards of all but one task are known, and that the dynamics are equal across tasks? This does not seem to fit into the usual multitask setting, where the dynamics may differ from task to task, and where the rewards are typically assumed to be unknown. ************************ POST-REBUTTAL COMMENTS I have read the authors' rebuttal and thank them for addressing my questions and suggestions. I believe that they have satisfactorily addressed the main points that I brought up.


Review 3

Summary and Contributions: This paper considers reinforcement learning with feedback graphs --- there is a directed graph with state-action pairs as nodes, and whenever the agent is in state s and takes action a, it observes not only the next state and reward, but also samples of transitions from the state-action pairs that have an edge from (s,a). The main contributions of this paper are to give lower and upper bounds on the regret and sample complexity of the problem, as a function of various properties of the feedback graph, namely the size of the maximal acyclic subgraph, the size of the largest independent set, and the size of the smallest dominating set. The first result shows that for standard model based algorithms, the dependence of the regret on the number of states and actions can be replaced by the m.a.s. number μ. In the lower bounds, SA can be replaced by γ, the size of the maximal independent set. For undirected graphs, μ = γ, so the bounds are tight. The improvement comes purely from the extra state transitions available to the learner. The other major contribution is to improve the dependence on γ to α, the minimal dominating set. This is known to be possible for bandit settings, since taking dominating actions is enough to learn about all the other actions. In MDPs, however, there is an additional factor depending on how frequently it is possible to visit dominating states. Furthermore, the authors have a lower bound showing that dependence on α cannot be completely eliminated in the lower-order terms. Furthermore, the authors show that the algorithm has a negligible penalty when the dominating set is not known a priori, since the algorithm can visit states and thereby recover the full feedback graph.

Strengths: This paper is a natural extension of the feedback graph idea from bandits to Markov decision processes. In bandits, it gives us a better understanding of the interplay between taking informative vs. rewarding actions, as a step towards the more general partial monitoring setting. It appears that some of these insights are valuable in the MDP setting as well, particularly the role of the m.a.s. vs independence numbers and the improvement to the dominating sets. The proof techniques used may also be interesting to others. This paper is also extremely comprehensive, with useful discussion in the appendix and many generalizations considered, including feedback that is only available sometimes and/or biased.

Weaknesses: While the paper is motivated based on extending feedback graphs from bandits to MDPs, it would be useful to give examples of applications where such structured feedback is likely to be available. In practice, MDPs and bandits are applied in different contexts, so a problem formulation that is convincing for one is not necessarily so for the latter. While I appreciated the theoretical contributions in this paper, I was left wondering how they could be applied.

Correctness: I did not have time to check the correctness of all the proofs, but I did skim through the major ones and checked that the results were reasonable. It is quite possible that I have missed errors.

Clarity: The paper is, on the whole, very clearly written. It suffers a little from being forced into a conference format, because much of the discussion in the (extremely long) appendix would have been valuable in the main text. In particular, I appreciated the illustrative examples of feedback graphs with their associated properties and characteristic numbers. Some minor comments are in the "additional feedback."

Relation to Prior Work: To my knowledge, the paper is correct in claiming to be the first exploration of MDPs with side-information in the form of feedback graphs. While this has been explored a fair bit in the stochastic bandits literature, there are additional non-obvious insights to be gained in the MDP settings. I think this paper is a useful and novel advance over prior work.

Reproducibility: Yes

Additional Feedback: * Not being familiar with the "IPOC" framework, I found Remark 2 out-of-place and hard to parse. It either needs some explanatory text, or to be moved to the appropriate part of the appendix. * In the caption of Table 1, please include the definition of Ŝ. Also, while the definition of p₀ is deferred to section 5 where it naturally belongs, it would be helpful to have a brief note on what it means earlier in the paper. * Footnote 3 is important and should be in the main text, since a natural question is how to deal with not knowing the dominating set. Questions: * The paper comments on the gap between the upper and lower bounds in the case of directed graphs, but it would be useful to have a qualitative discussion of whence this gap arises and whether it's likely to be "real". I do not have much intuition for what directed feedback graphs would mean for minimizing regret, so it would be useful for the authors to share theirs. Such a discussion is provided in the appendix for the necessity of the independence number in the lower bound, and is very useful. * Loosely speaking, in other RL work some sort of state-action representation scheme is used to generalize learned values between states. Then one may consider the hardness of planning with a given representation, vs. the hardness of learning a good representation (which is still largely unknown). The idea of a feedback graph blurs that line. I would like to see a broader comparison of the feedback-graph approach to structured MDPs vs. other more common assumptions in the literature. *** Post-rebuttal update Thanks to the authors for their response. I'm looking forward to seeing the suggested changes made in the paper, including more illustrative examples/applications and moving the intuitive explanations from the appendix to the main paper.


Review 4

Summary and Contributions: Thanks for the reply and the authors answered my question in the author feedback. The paper has a major structural issue. A significant portion of the material is presented in the appendix. This not only makes reviewing process difficult, but also bypass the page limit. I believe this paper is more suited for a journal submission. This paper considers a tabular MDP problem with feedback graphs. Authors prove that existing model-based RL algorithms achieve regret and sample complexity bounds do not scale with SA. Lower bounds of the regret and the sample complexity are also presented to show the optimality of their analysis. Authors also present an algorithm that achieves a sample complexity bound that scales with the more favorable domination number.

Strengths: The problem of online MDPs with feedback graph is interesting and promising. The theoretical analysis is rigorous and valid.

Weaknesses: 1. In algorithm 3, you assume the dominating set is known to the agent. I am not sure this is true is most reinforcement learning applications. In the robot moving example, the transition model is unknown which means the agent does not know which states are in the same line. Then you don’t know the dominating set in this case. 2. In algorithm 3, authors use the extended MDP to simultaneously learn policies to reach the goal and the points in the dominating set. However, the sample complexity analysis does not show the benefit of the ‘multi-task’ learning process.

Correctness: Yes. Through regret analysis, this paper shows that the proposed algorithm utilizes the feedback graph information and achieves a better regret bound.

Clarity: Yes. The problem is well explained. The proofs are easy to read and follow.

Relation to Prior Work: Yes. Related works are well discussed and

Reproducibility: Yes

Additional Feedback:

[Author Response · NeurIPS 2020]

**Scope:** We thank all reviewers for their useful comments. Our goal was to theoretically investigate how side observations, formalized by feedback graphs and available in several RL applications, can be used to learn faster in MDPs. For a first study of this problem, a tabular assumption is a natural starting point that already posed several new challenges compared to the bandit setting, which we addressed in a comprehensive way. While direct applications of any tabular approach are limited, we are excited about extensions to non-tabular settings that would capture many real-world applications and believe that the insights in our paper provide the basis for such extensions. A careful empirical evaluation would also be a nice complement indeed, but our theoretical study has already led to a rich and dense content. Thus, we have chosen to devote a separate study to the applications of our theory and experiments in the future.

**Reviewer 1:**

• **Tabular MDP, accurate side info:** We cover biased (inaccurate) side observations in the appendix. Regarding the tabular assumption, please see comments about the scope above. We will clarify that Lines 45-54 are motivation for feedback graphs in MDPs generally and not necessarily for the tabular case. We believe that our assumptions and claims are carefully stated but we will be happy to rectify any part of the paper that the reviewer views as "careless".

• **More intuition around feedback graph:** There appears to be a misunderstanding about the role of a feedback graph. An edge in the feedback graph does *not* indicate a valid transition. An edge between $(s, a)$ and $(\bar{s}, \bar{a})$ indicates that when the agent is in state $s$ and takes action $a$ it also observes some transition observation $(\bar{s}, \bar{a}, \bar{r}, \bar{s}')$ from $(\bar{s}, \bar{a})$. However, the edge does not indicate anything about the successor states from $(s, a)$ or $(\bar{s}, \bar{a})$. We will provide more intuition about the graph properties and move examples from Appendix A to the main body (see item 2 of R3).

• **Correctness of implicit self loops:** The analysis is correct and consistent with our definitions. We defined feedback graphs to not contain any self-loops and to only stipulate what observations are available *in addition* to the transition performed by the agent ($\mathcal{O}_h(G)$ on pg 3). This is in line with prior work in bandits [24]. If we wanted to make self-loops explicit, then the definitions of the graph properties would need to be changed to ignore such self loops.

**Reviewer 2:**

• **Replay buffer and off-policy RL:** These are interesting directions and we believe that future work with regret bounds for model-free algorithms is a good next step (as replay buffers and explicit off-policy RL matters most here).

• **Challenges for model-free RL:** Model-free methods use the Q-value of the successor state to update the current state. In the feedback graph setting, one might use side observations to update a state $s$ many times using Q-values of a successor state that has never been observed. Then the Q-value estimate of $s$ is still bad, even though the number of observations is large. Without feedback graphs, where all observations come in full trajectories, this cannot happen.

• **Multi-task settings:** It is correct that our assumptions only hold in certain multi-task settings. The first phase of Alg 3 is one example. Other examples are multiple-destination shortest path problems where the agent needs to learn how to reach different goal states in the same environment. Each goal corresponds to one reward function that can often be assumed to be known.

**Reviewer 3:**

• **Example applications:** Besides *recommender systems* and *image augmentation* discussed in the introduction, we expect that such structured side observations to be available in *certain robotics applications* where partial knowledge of the environment is often available. Other examples include problems in *personalized tutoring systems*, *autonomous driving* and *personalized medicine*. We will include a list of tasks with more details in the paper.

• **Feedback graph examples (App A):** Thank you for highlighting their helpfulness. We will move some to Sec 2.

• **Gap between upper- and lower-bound:** We will provide a discussion in the appendix showing that non-randomized UCB algorithms like Alg. 1 cannot avoid an mas-number dependency in a lower-order term. But whether any algorithm can achieve scaling with independence number in the main-order $\sqrt{T}$ term is an interesting open question.

• **Feedback graphs vs. other approaches to structured MDPs:** Good suggestion, We will add a brief discussion.

**Reviewer 4:**

• **Alg 3 assumes dominating set known:** The assumption of a known dominating set was made to simplify the presentation. As we briefly allude to in Appendix F.1, one can apply a sightly modified version of Alg. 3 to problems where a dominating set is unknown. One then has S tasks in the first phase (one to reach each state) and move on to the second phase as soon as a suitable dominating set was discovered (which we can test at run-time). The sample-complexity of this modified algorithm is identical up to log-terms. We will expand on this in Appendix F.1.

• **Benefit of multi-task learning process:** If Algorithm 3 did not use the multi-task learning process, this would yield a sample-complexity bound of order $\frac{\gamma\mu\widehat{S}H^2}{p_0}$ when the dominating set is known and $\frac{S\mu\widehat{S}H^2}{p_0}$ when it is unknown (since we then pay an additional linear factor in the number of states we want to learn to reach). This is substantially worse than the bound enabled by our analysis. We will expand on the sketch of why this is true in the paper.

[Meta-Review · NeurIPS 2020]

All reviewers have positive opinions of this paper, with scores of (7, 7, 6, 6). I also agree that the paper is quite novel and intriguing. The main issue appears to be readability, particularly regarding the extremely lengthy supplemental material. Quoting from the reviewers: This is a well-written theoretical paper. Although it does not demonstrate empirically that feedback graphs work in real-life problems, the underlying theory seems sound and the formal results that are presented are certainly not trivial. It appears to be the first exploration of MDPs with side-information in the form of feedback graphs.